# A Study of Sea Ice Topography in the Weddell and Ross Seas Using Dual Polarimetric TanDEM-X Imagery

Lanqing Huang[1,3] and Irena Hajnsek[1,2]

[1]Institute of Environmental Engineering, Swiss Federal Institute of Technology in Zürich (ETH), Zürich, 8093, Switzerland
[2]Microwaves and Radar Institute, German Aerospace Center (DLR), Wessling, 82234, Germany
[3]Center for Polar Observation and Modelling, University College London, WC1E6BS, London

**Correspondence:** Lanqing Huang (huang@ifu.baug.ethz.ch)

**Abstract.**

The total freeboard, which is the ice layer above water level and includes snow thickness, is needed to retrieve ice thickness and ice surface topography. Single-pass interferometric synthetic aperture radar (InSAR) allows for the generation of digital elevation models (DEMs) over the drifting sea ice. However, accurate sea ice DEMs (i.e., total freeboard) derived from InSAR

are impeded due to the radar signals penetrating the snow and ice layers. This research introduces a novel methodology for retrieving sea ice DEMs using dual-polarization interferometric SAR images, considering the variation in radar penetration bias across multiple ice types. The accuracy of the method is verified through photogrammetric measurements, demonstrating the derived DEM with a root-mean-square error of $0.26\,\mathrm{m}$ over a $200 \times 19\,\mathrm{km}$ area. The method is further applied to broader regions in the Weddell and the Ross Sea, offering new insights into the regional variations of sea ice topography in the Antarc-

tic. We also characterize the non-Gaussian statistical behavior of total freeboard using log-normal and exponential-normal distributions. The results suggest that the exponential-normal distribution is superior in the thicker sea ice region (average total freeboard $> 0.5\,\mathrm{m}$), whereas the two distributions exhibit similar performance in the thinner ice region (average total freeboard $< 0.5\,\mathrm{m}$). These findings offer an in-depth representation of total freeboard and roughness in the Weddell and Ross Seas, which can be conducted in time series data to comprehend sea ice dynamics, including its growth and deformation.

**1   Introduction**

The Sea ice topography refers to the ice shape, height, and large-scale roughness at the meter scale. It encompasses a variety of ice features, including rafted ice, ridges, rubble fields, and hummocks, all of which contribute to the intricate nature of sea ice topography (Weeks and Ackley, 1986). The presence of snow cover atop the ice surface further influences the topographic characteristics, adding another layer of complexity to the overall sea ice topography (Massom et al., 2001).

The sea ice surface topography plays a crucial role in understanding sea ice dynamics and interactions within the air-ocean-ice system. It determines the spatial distribution of distinct surface features such as snow dunes (Trujillo et al., 2016; Iacozza and Barber, 1999) and deformed ice (Haas et al., 1999; Petty et al., 2016), which are impacted by the forces from winds and currents. Moreover, the atmospheric drag coefficient over sea ice, which is topography-dependent, is an important parameter for understanding interactions at the ice-atmosphere boundary (Garbrecht et al., 2002; Castellani et al., 2014).

Sea ice topography can be described through digital elevation models (DEM), which refers to the total freeboard (snow+ice) above the local sea surface. The DEM (i.e., total freeboard) can be converted to thickness with the knowledge of snow depth and the assumed values of snow, ice, and seawater densities (Kwok and Kacimi, 2018). Estimating sea ice thickness over time offers valuable insights into the overall stability of sea ice in the changing climate. Furthermore, mapping sea ice topography is paramount for safe navigation in polar oceans. By providing information on ice deformation and identifying safe routes, accurate sea ice topography maps contribute to ensuring the safety and efficiency of ship navigation in challenging environments (Dammann et al., 2017).

Sea ice DEMs can be obtained using laser altimeter mounted on different platforms, including helicopters (Dierking, 1995), aircraft such as IceBridge (Petty et al., 2016), and satellites like ICESat-1 (Zwally et al., 2008) and ICESat-2 (Kacimi and Kwok, 2020). These laser altimeters provide high-spatial resolution ($< 1\,\mathrm{m}$) in measuring total freeboard. However, limited spatial coverage and long revisit times (e.g., 91 days for ICESat-2) restrict their capacity for consistent and comprehensive sea ice monitoring. In recent decades, synthetic aperture radar (SAR) has been of significant importance for Earth observation, offering a balance between spatial resolution (meters to tens of meters) and swath coverage (tens to hundreds of kilometers). SAR is unaffected by weather conditions or daylight limitations, enabling consistent data acquisition with a revisit time of around ten days. Notably, the single-pass interferometric SAR (InSAR) sensor, exemplified by TanDEM-X, presents an unprecedented opportunity to generate sea ice DEMs over landfast sea ice (Dierking et al., 2017; Yitayew et al., 2018). For drifting ice, the accuracy of InSAR-derived DEMs can be affected by additional phase shifts induced by ice motion. Dierking et al. (2017) calculated and theoretically discussed the sensitivity of InSAR-derived DEMs concerning ice-drifting velocity, InSAR frequency, and baseline configuration.

Nevertheless, the InSAR-derived DEM can be affected by the microwave penetration into the snow and ice layers. Dry snow can have penetration depths up to hundreds of wavelengths (Guneriussen et al., 2001). For X-band SAR, the penetration into younger ice, such as new and first-year ice, is minimal due to the high salinity of the ice surface (Hallikainen and Winebrenner, 1992). On the other hand, for older and desalinated ice, such as multi-year ice, the penetration depth varies from $0 - 1\,\mathrm{m}$ depending on the temperature and salinity (Hallikainen and Winebrenner, 1992; Huang et al., 2021). To account for the scattering mechanism from the volumes (snow and ice) and layers (snow-ice-water interfaces), a two-layer-plus-volume (TLPV) model (Huang et al., 2021) has been developed to determine the penetration depth over snow-covered old ice in the Antarctic. The model improves the precision of sea ice topographic mapping by offsetting the InSAR phase center to the top surface.

SAR polarimetry complements interferometry by providing valuable insights into scattering processes and has proven useful for characterizing sea ice properties (Winebrenner et al., 1995; Ressel et al., 2016; Singha et al., 2018). For old and deformed ice, a radar theory has been developed to examine the relationship between scattering mechanisms and sea ice DEM (Nghiem et al., 2022), resulting in a geophysical model function based on co-polarimetric coherence for retrieving sea ice DEM (Huang et al., 2022). These findings emphasize the significance of integrating polarimetric and interferometric information for accurate sea ice topography mapping using SAR imagery.

Given the variations in the microwaves' penetration depth into snow and ice, deriving sea ice DEM from SAR imagery over a broad spatial scale encompassing diverse ice types is still constrained. In this study, we develop an innovative two-step method

to generate sea ice DEM across multiple ice types using machine learning and polarimetric-interferometry SAR techniques. The initial step involves the development of a random forest classifier using specific SAR features to categorize sea ice into two groups: small-penetration condition ice (SPI) and large-penetration condition ice (LPI), based on the penetration depth of microwaves into the snow and ice. Subsequently, a sea ice DEM is created for each ice type. In the case of SPI, standard InSAR processing is applied to determine the total freeboard. For LPI, a novel inversion algorithm is proposed to estimate the parameters of the developed TLPV model (Huang et al., 2021). This model allows for correcting penetration bias in the InSAR signal over LPI, resulting in an accurate retrieval of the total freeboard. We validate the proposed method against the photogrammetric DEM from the IceBridge aircraft. A root-mean-square error (RMSE) of $0.26\,\mathrm{m}$ between the derived DEM and reference data indicates an improved accuracy in total freeboard retrieval.

We further implement the proposed two-step approach to 162 SAR images covering 12 segments (each covering an area of $\sim 500 \times 20\,\mathrm{km}$) in the Weddell and Ross Seas. This allows a broad mapping of sea ice DEM and roughness, offering new insights into the topographic patterns of sea ice at a large spatial scale. Note that the roughness in this study refers to the macroscale roughness, which is defined as the standard deviation of total freeboard within $50 \times 50\,\mathrm{m}$ window. We analyze the variation in sea ice DEM and roughness along the southwards direction and associate it with the variation in sea ice classes obtained from an operational product from the U.S. National Ice Center. The statistics of total freeboard over various regions are modeled using the log-normal and exponential-modified normal distributions. The findings enhance our understanding of sea ice formation and dynamics and can be used to interpret geophysical parameters associated with sea ice topography.

The paper is structured as follows. Section 2 describes the data sets and data processing procedures. The two-step approach for sea ice DEM retrieval is introduced in Sect. 3. The retrieval results and interpretation of topographic characteristics are given in Sect. 4 and further discussed in Sect. 5. Finally, Section 6 summarizes the study.

## 2 Data sets and processing

### 2.1 Study area

The region of interest includes both the Weddell Sea and the Ross Sea, as shown in Fig. 1. The SAR footprints over the two seas are zoomed-in in boxes A and B, respectively. The footprints consist of 12 segments, each corresponding to a sequence of SAR images within the same orbit that were acquired with only a few seconds difference. The segments will be referred to as W1-U, W1-L, W2-U, W2-L, W3-U, W3-L, W4, W5-U, W5-L, R1-U, R1-L, and R5 in the following sections for conciseness.

### 2.2 SAR Imagery

The TanDEM-X is a SAR interferometer that operates as a bistatic single-pass system, capable of acquiring two images simultaneously (Krieger et al., 2007). The two images are co-registered single-look complex products, which can be processed to derive sea ice DEM through interferometry.

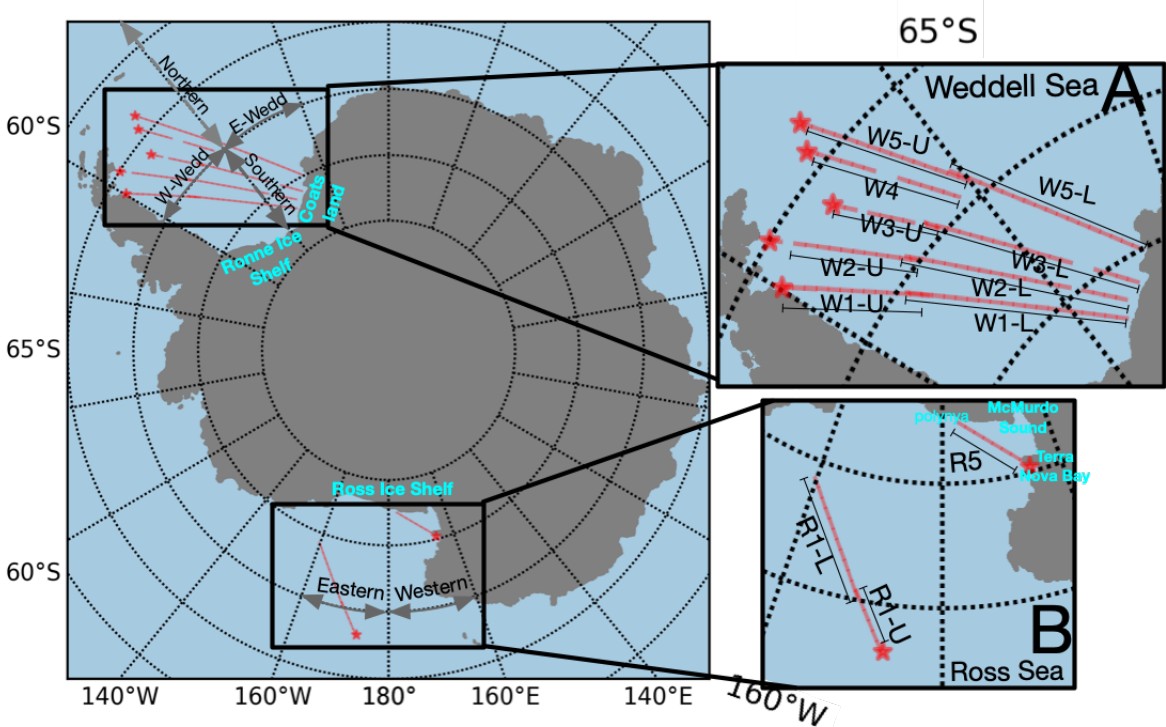

**Figure 1.** Geolocation of the study area. The northernmost positions in each segment are marked with star symbols and serve as reference points for calculating the relative distance in Sec. 4.2.

In the study, we collected 162 SAR images over the twelve segments in StripMap mode in dual-pol channels (HH and VV). The pixel spacing is around $0.9\,\mathrm{m} \times 2.7\,\mathrm{m}$ in slant range and azimuth. The acquisition time and the number of images for each segment are listed in Table 1. The incidence angle (InA) is measured at the center of the scene, and the height of ambiguity (HoA) corresponds to an interferometric phase change of $2\pi$. Note that for R5, the larger HoA leads to relatively higher average uncertainty in the derived InSAR height ($h_{\mathrm{InSAR}}$) compared to other InSAR configurations with smaller HoA. More details can be found in the appendix A1.

The multilooking processing was conducted using a $4 \times 12$ window, resulting in a $\sim 10 \times 10\,\mathrm{m}$ pixel spacing in azimuth and ground range. This resolution ($\sim 10 \times 10\,\mathrm{m}$) was subsequently utilized for the sea ice classification and DEM retrieval detailed in Section 3. The backscattering intensity $\sigma_{\mathrm{measure}}$ of the images includes additive thermal noise, which can be described by the noise equivalent sigma zero (NESZ) and assumed to be uncorrelated with the signal (Nghiem et al., 1995). Removing the thermal noise allows for a better representation of sea ice features, which is crucial for ice classification. We denoised backscattering intensities for the different polarizations (i.e., HH, VV, Pauli-1 (HH+VV), and Pauli-2 (HH-VV)) by subtracting

**Table 1.** Summary of SAR acquisitions and Ice Charts over the study area.

| Segment | Number of SAR images | SAR acquisition time | HoA(m) | InA(∘) | Weekly average Ice Charts (ending date) |
|---------|---------------------|----------------------|--------|--------|-----------------------------------------|
| W1-L | 20 | 2017-10-24T23:30 | $30 - 35$ | 29 | 2017-10-26 |
| W1-U | 13 | 2017-10-29T23:41 | $33 - 35$ | 35 | 2017-11-02 |
| W2-L | 19 | 2017-10-25T23:13 | $30 - 35$ | 29 | 2017-10-26 |
| W2-U | 12 | 2017-10-30T23:23 | $32 - 34$ | 35 | 2017-11-02 |
| W3-L | 18 | 2017-10-26T22:56 | $30 - 35$ | 29 | 2017-11-02 |
| W3-U | 8 | 2017-11-22T23:05 | $36 - 37$ | 35 | 2017-11-23 |
| W4 | 12 | 2017-11-01T22:49 | $32 - 34$ | 35 | 2017-11-02 |
| W5-L | 18 | 2017-11-02T22:30 | $30 - 34$ | 29 | 2017-11-09 |
| W5-U | 15 | 2017-10-27T22:41 | $31 - 34$ | 35 | 2017-11-02 |
| R1-L | 12 | 2017-11-11T07:16 | $33 - 35$ | 31 | 2017-11-16 |
| R1-U | 6 | 2017-10-25T07:25 | $34 - 35$ | 36 | 2017-10-26 |
| R5 | 9 | 2017-11-07T09:58 | $40 - 42$ | 35 | 2017-11-09 |

the noise equivalent sigma zero (NESZ) from the $\sigma_{\mathrm{measure}}$ (Huang et al., 2022). More details about the thermal noise removal can be found in the appendix A2. The denoised backscattering intensities are used in the following sections.

## 2.3 Optical Digital Mapping System (DMS) data

With an objective to investigate Antarctic sea ice topography, Operation IceBridge (OIB) and TanDEM-X Antarctic Science Campaign (OTASC) (Nghiem et al., 2018) was successfully carried out along a portion of the W1, shown in Fig. 2a. Equipped with a digital mapping system (DMS), the OIB aircraft captured optical images (Dominguez, 2010, updated 2018) and generated DEM using photogrammetric techniques at a spatial resolution of approximately $40\,\mathrm{cm} \times 40\,\mathrm{cm}$ with a vertical accuracy of $0.2\,\mathrm{m}$ (Dotson and Arvesen., 2012, updated 2014). The DMS acquisitions occurred between 17:45 and 18:44 UTC on October

29, 2017. Figure 2b and c showcase DMS optical images over specific areas, highlighting a diverse range of sea ice features, including ridges, deformed ice, smooth ice with snow cover, and snow-free ice.

In this study, we geocoded the DMS DEM to match the same coordinates and resolution as the multilooked SAR image, which is approximately $10 \times 10\,\mathrm{m}$ in both range and azimuth. Note that DMS DEM gives height values relative to the WGS-84 ellipsoid. To obtain the total freeboard, we calibrated the DMS DEM to the local sea level through a manual selection of the

115 water surface from DMS images (Huang et al., 2021). The calibrated DMS DEM, henceforth is referred to as DMS DEM for brevity.

As the sea ice is constantly moving, co-registration is crucial to compensate for the time lag ($\sim 6$ hours) between the DMS sensor and TanDEM-X. To achieve this, we carefully aligned the two data by identifying distinctive sea ice features in both

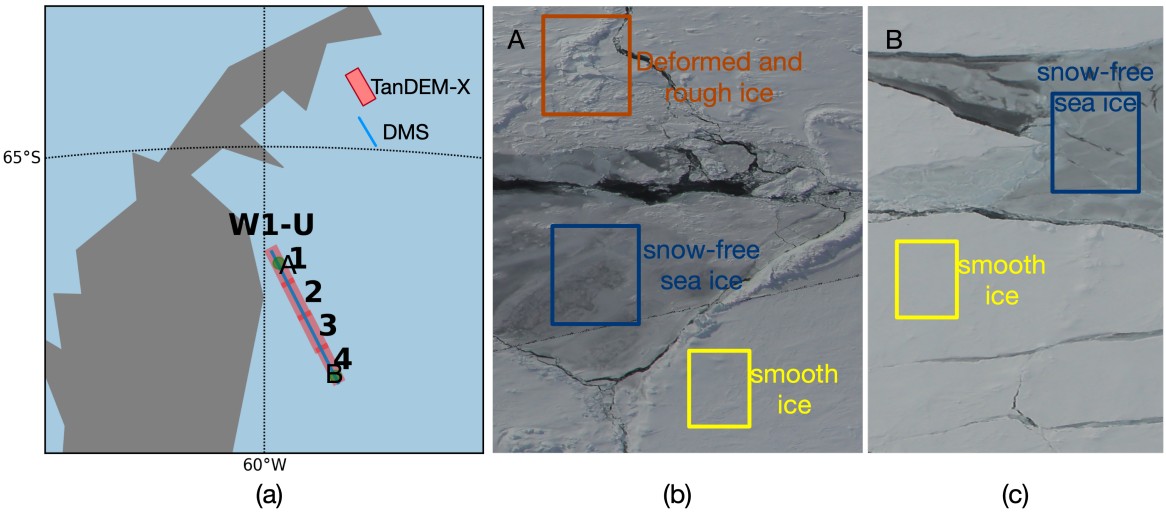

**Figure 2.** (a) Geolocation of DMS measurements superimposed on four SAR footprints in segment W1-U. Zoomed-in views of DMS digital images at points A and B (green dots) are displayed in (b) and (c), respectively.

optical and SAR images (Huang et al., 2021, 2022). Note that the segments lacking distinctive sea ice features in optical and
SAR images were eliminated to ensure co-registration quality. The co-registered DMS DEM is used as reference data in this study.

## 2.4 Ice Charts

The U.S. National Ice Center's Antarctic sea ice charts (referred to as Ice Charts hereafter) offer weekly products detailing total sea ice concentration, partial concentration, and stage of development (U.S. National Ice Center., 2022). The Ice Charts
covering the date of SAR acquisitions are listed in Table 1.

The Ice Charts are provided in Shapefile format as grids with a spatial resolution of $10 \times 10 \, \text{km}$. For each specified latitude and longitude, three ice concentration values are given, each corresponding to a different stage of ice development. Details of these stages and their corresponding thicknesses can be found in the first and second columns of Table 2, respectively. The postprocessing of the Ice Charts consists of two steps. First, we categorized the three stages of ice into thin ice (TI), first-year
ice (FYI), and multiyear ice (MYI) types according to the third column of Table 2. Next, we extracted the ice concentration values for TI, FYI, and MYI, respectively. An example of the ice chart showing the dominant ice type is provided in Fig. 3. Note that the dominant ice type refers to the ice type with the highest concentration values.

**Table 2.** Stage of develops for ice type categories (U.S. National Ice Center., 2022).

| Ice Stage of development | Thickness (cm) | Ice type |
|---|---|---|
| New ice | $< 10$ | |
| Nilas, ice rind | $< 10$ | |
| Young ice | $10- < 30$ | Thin ice (TI) |
| Gray ice | $10- < 15$ | |
| Gray-white ice | $15- < 30$ | |
| FYI | $\geq 30 - 200$ | |
| Thin FYI | $30- < 70$ | |
| Medium FYI | $70- < 120$ | First-year ice (FYI) |
| Thick FYI | $\geq 120$ | |
| Old ice | | |
| 2nd year ice | N/A | Multiyear ice (MYI) |
| multiyear ice | | |

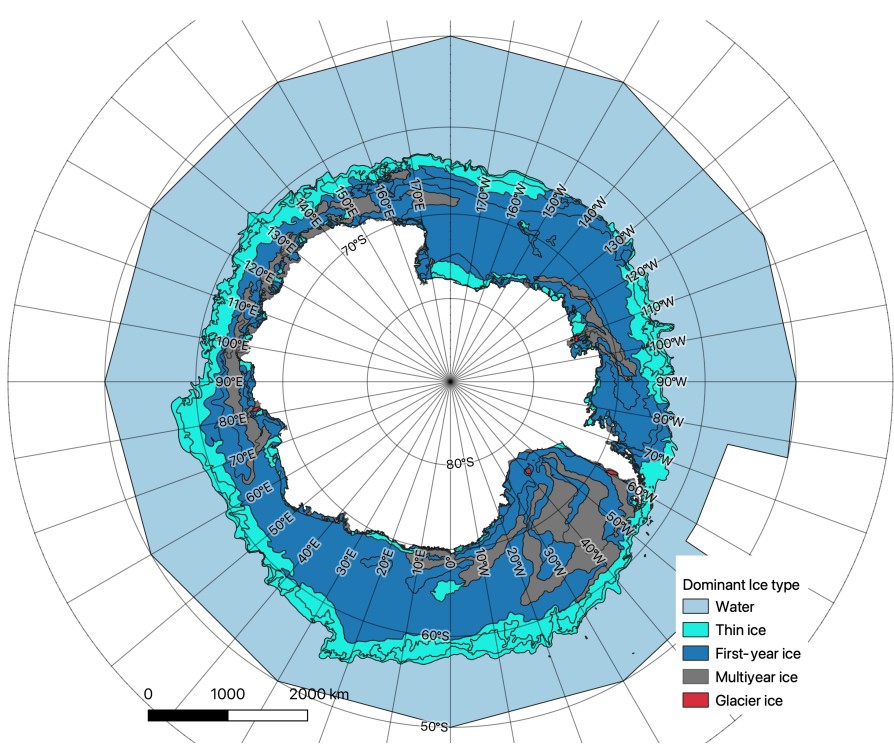

**Figure 3.** The dominant ice type from the U.S. National Ice Center's Antarctica weekly sea ice chart (October 26th, 2017).

## 2.5 SAR interferometry

Single-pass interferometer acquires two simultaneous observations, denoted as $s_1$ and $s_2$. The complex interferogram $\gamma$ and the interferometric phase $\phi_\gamma$ can be described as (Cloude, 2010)

$$\gamma = s_1 s_2^* \tag{1}$$

$$\phi_\gamma = \arg\{s_1 s_2^*\} \tag{2}$$

The further processing of $\phi_\gamma$ includes flat earth removal, interferogram filtering, low-coherence area mask, and phase unwrapping (Huang and Hajnsek, 2021). The resulting $\phi_\gamma'$ can be converted to height by

$$h_{\text{InSAR}} = h_a \frac{\phi_\gamma'}{2\pi} \tag{3}$$

where $h_{\text{InSAR}}$ is the height of InSAR phase center and $h_a$ is the HoA related to the InSAR baseline configuration provided in Table 1.

The complex interferometric coherence $\tilde{\gamma}_{\text{InSAR}}$ between the two images can be estimated by (Cloude, 2010)

$$\tilde{\gamma}_{\text{InSAR}} = \gamma_{\text{InSAR}} \cdot e^{i\phi_\gamma} = \frac{<s_1 s_2^*>}{\sqrt{<s_1 s_1^*><s_2 s_2^*>}} \tag{4}$$

where the symbol $< . >$ denotes an ensemble average within a $4 \times 12$ multilooking window. Pixels with $\gamma_{\text{InSAR}} < 0.3$ were designated as water areas and excluded from further processing. The above interferometric processing was carried out using the GAMMA software.

The $h_{\text{InSAR}}$ obtained from Eq. 3 was further calibrated to the average water surface. Instead of identifying water pixels that were masked out due to the low InSAR coherence (less than 0.3), we selected smooth and new ice regions, assuming they are thin enough and their elevation (i.e., radar freeboard) is negligible and approximately equal to the water surface. The smooth and thin ice regions typically exhibit very low backscattering intensities in SAR image (Dierking et al., 2017). Therefore, we selected pixels with backscattering intensities within the range of $-19\,\text{dB}$ to $-18\,\text{dB}$, slightly above TanDEM-X's noise level ($-19\,\text{dB}$), and generated a histogram of $h_{\text{InSAR}}$ values for these pixels. We determined the 3rd percentile of the height of these pixels as the water surface elevation. The threshold value, i.e., the 3rd percentile, was chosen based on applying the method to four SAR scenarios overlaid with DMS DEM. By choosing the 3rd percentile as the water surface level, we ensured alignment between the water surface levels derived from InSAR and those from the DMS data, thus validating the threshold value. Note that we estimate a single value representing the water surface for each SAR scene. However, it is important to note that this method may introduce inaccuracies due to the centimeter-level radar freeboard of the selected thin and new ice, as well as the fluctuating water surface within each SAR scenario.

## 2.6 SAR polarimetry

SAR Polarimetry reflects scattering mechanisms and has been proven as a proxy for characterizing sea ice properties (Wakabayashi et al., 2004; Ressel et al., 2016; Huang and Hajnsek, 2021; Singha et al., 2018; Nghiem et al., 2022).

### 2.6.1 Co-polarization ratio

The co-polarization (coPol) ratio ($R_{\mathrm{coPol}}$) measures the backscattering intensity ratio between the dual-pol channels and can be calculated as follows

$$R_{\mathrm{coPol}} = \frac{\sigma_{\mathrm{HH}}}{\sigma_{\mathrm{VV}}} \tag{5}$$

where $\sigma_{\mathrm{HH}}$ and $\sigma_{\mathrm{VV}}$ are denoised SAR backscattering intensity in dual-pol channels in linear scale. $R_{\mathrm{coPol}}$ extracted from L-band SAR images is associated with the dielectric constant and has therefore been used as an indicator of ice thickness

(Wakabayashi et al., 2004). Further investigation is required to determine if $R_{\mathrm{coPol}}$ from the X-band can also serve as a proxy for ice thickness. Additionally, $R_{\mathrm{coPol}}$ has been identified as an important feature for discriminating thicker ice and water and is an effective tool for classifying sea ice in X-band SAR imagery (Ressel et al., 2016).

### 2.6.2 Pauli-polarization ratio

Similarly, we can obtain the Pauli-polarization ratio ($R_{\mathrm{pauli}}$) by

$$R_{\mathrm{Pauli}} = \frac{\sigma_{\mathrm{P1}}}{\sigma_{\mathrm{P2}}} = \frac{|s_{\mathrm{HH}} + s_{\mathrm{VV}}|^2}{|s_{\mathrm{HH}} - s_{\mathrm{VV}}|^2} \tag{6}$$

where $\sigma_{\mathrm{P1}}$ and $\sigma_{\mathrm{P2}}$ are denoised SAR backscattering intensity in Pauli-1 and Pauli-2 polarizations in linear scale, respectively. $s_{\mathrm{HH}}$ and $s_{\mathrm{VV}}$ are single-look complex images in dual-pol channels, respectively.

### 2.6.3 Complex coPol coherence

The complex coPol correlation $\tilde{\gamma}_{\mathrm{coPol}}$ is calculated as (Lee and Pottier, 2009)

$$\tilde{\gamma}_{\mathrm{coPol}} = \gamma_{\mathrm{coPol}} \cdot e^{i\phi_{\mathrm{coPol}}} = \frac{< s_{\mathrm{VV}} s_{\mathrm{HH}}^* >}{\sqrt{< s_{\mathrm{VV}} s_{\mathrm{VV}}^* >< s_{\mathrm{HH}} s_{\mathrm{HH}}^* >}} \tag{7}$$

where $\gamma_{\mathrm{coPol}}$ is the coPol coherence magnitude and $\phi_{\mathrm{coPol}}$ is the coPol phase.

$\gamma_{\mathrm{coPol}}$ measures the degree of electromagnetic wave depolarization caused by the surface roughness and the volume scattering. This parameter has been shown to be associated with sea ice DEM (Huang and Hajnsek, 2021) and thickness (Kim et al., 2011).

$\phi_{\mathrm{coPol}}$ is sensitive to the anisotropic structure of the medium and deviates from $0°$ when the signal delay becomes polarization dependent (Leinss et al., 2014). $\phi_{\mathrm{coPol}}$ has been utilized in retrieving fresh-snow anisotropy over ground (Leinss et al., 2016) and characterizing the topography of snow layer (Huang and Hajnsek, 2021).

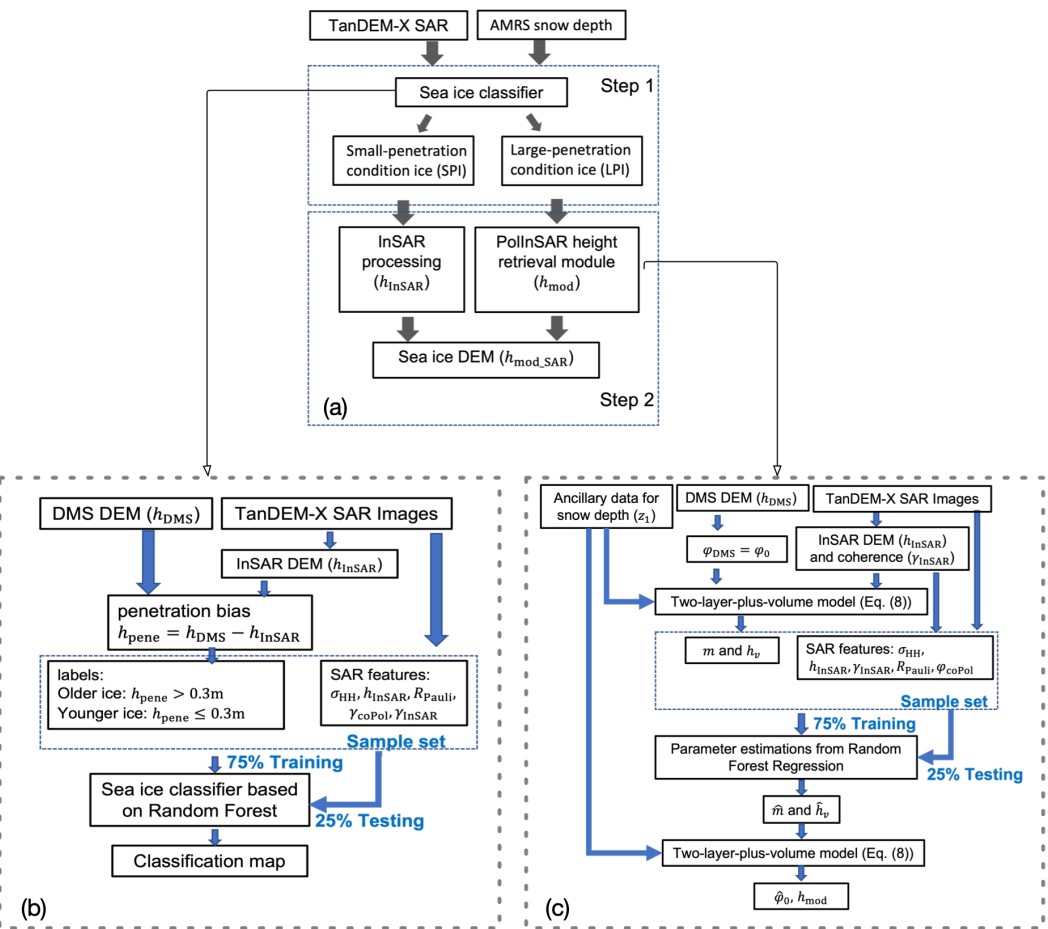

**Figure 4.** (a) The proposed two-step approach for sea ice DEM retrieval. (b) The details (training and validation) of the sea ice classifier and (c) the PolInSAR height retrieval module.

## 3 Methodology

This section introduces an innovative two-step approach for retrieving sea ice DEM across various ice conditions. The initial
step is categorizing sea ice into LPI and SPI types based on radar penetration depths. The second step involves generating the sea ice DEM using different methods for the two ice categories. The two-step approach is presented in Fig. 4(a) and are detailed in Sect. 3.1 and Sect. 3.2, respectively. The method is developed and validated using the four SAR images (Fig. 2) overlapped with DMS DEM.

### 3.1 Sea ice classification

As shown in Fig. 4(a), in Step 1, the sea ice is classified into SPI and LPI using a random forest (RF) classifier (Breiman, 2001).
A detailed description of the training and validating process for the classifier is given in Fig. 4(b), where DMS DEM ($h_{\mathrm{DMS}}$)

are utilized as reference data. The penetration depth $h_{\mathrm{pene}} = h_{\mathrm{DMS}} - h_{\mathrm{InSAR}}$, where $h_{\mathrm{DMS}}$ measures the total freeboard, i.e., elevation from the snow-air surface relative to the water level. InSAR DEM ($h_{\mathrm{InSAR}}$) measures the radar freeboard, i.e., the elevation of the InSAR phase center relative to the water level, which can be somewhere inside of the snow or ice, depending on the snow and ice condition. $h_{\mathrm{InSAR}}$ is generated from TanDEM-X InSAR pair following the principles in Sect. 2.5. In general, microwaves can penetrate much shallower into the younger and more saline compared to the older and less saline sea ice. According to Hallikainen and Winebrenner (1992) the penetration depth into multi-year ice varies between $0.3$ and $1\,\mathrm{m}$ at X-band, depending on salinity and temperature. Desalination within ice ridges increases the effective penetration depth compared to level ice (Dierking et al., 2017). Considering these findings and given the study area's snow cover and the presence of deformed ice formations such as ridges, we chose a penetration depth of $0.3\,\mathrm{m}$ as the threshold for distinguishing the two ice types. Hence, pixels with $h_{\mathrm{pene}} < 0.3\,\mathrm{m}$ are labeled as SPI, whereas those with $h_{\mathrm{pene}} \geq 0.3\,\mathrm{m}$ are LPI.

We investigate a range of features for classification, including denoised backscattering intensity in HH polarization ($\sigma_{\mathrm{HH}}$), polarimetric features such as coPol ratio ($R_{\mathrm{coPol}}$), Pauli-polarization ratio ($R_{\mathrm{Pauli}}$), coPol coherence magnitude ($\gamma_{\mathrm{coPol}}$), and coPol phase ($\phi_{\mathrm{coPol}}$), as well as interferometric features including InSAR coherence magnitude ($\gamma_{\mathrm{InSAhR}}$) and height of interferometric phase center ($h_{\mathrm{InSAR}}$). To improve computational performance, we rank features based on Gini Importance (i.e., Mean Decrease in Impurity), which measures the average gain of purity by splits of a given variable. The top five features, i.e., $R_{\mathrm{Pauli}}$, $\sigma_{\mathrm{HH}}$, $h_{\mathrm{InSAR}}$, $\gamma_{\mathrm{coPol}}$, and $\gamma_{\mathrm{InSAR}}$ are selected as effective predictors for the RF classifier. The ranking of all the SAR features is given in the appendix Fig. A3. Note that the computed Gini importance is not inherently specific to a particular class or ice type. Instead, it represents the relative importance of features in making overall classification decisions within the context of the entire dataset. Therefore, the importance level determined by Gini importance is not specific to individual ice types but reflects the significance of features for the classifier's overall predictive performance across all classes.

The selected features together with the ice labels (i.e., LPI and SPI) form the sample set. 75% is used for training the RF classifier, implemented in Python using default hyperparameters. Since sample numbers for the SPI and LPI classes are well-balanced (48% and 52%, respectively), no balanced training strategy is particularly implemented. The validation of ice classification over the testing subset (25%) will be given in Sect. 4.1.

## 3.2   DEM generation

As shown in Fig. 4(a), in Step 2, we separately retrieve the sea ice DEMs for the two categories of ice based on the classification map. For SPI, the conventional InSAR processing (Section 2.5) is conducted, given the minimal penetration depth attributed to the saline ice. On the other hand, for LPI which is subject to radar signal penetration, we apply the TLPV model developed in (Huang et al., 2021), which incorporates InSAR processing and corrects for the radar penetration bias into the snow-covered old ice.

The TLPV model includes surface scattering from the top and bottom interfaces and volumes scattering from the snow and ice, shown in Fig. 5. The model was further simplified by merging the contributions of the snow volume, the ice volume, and

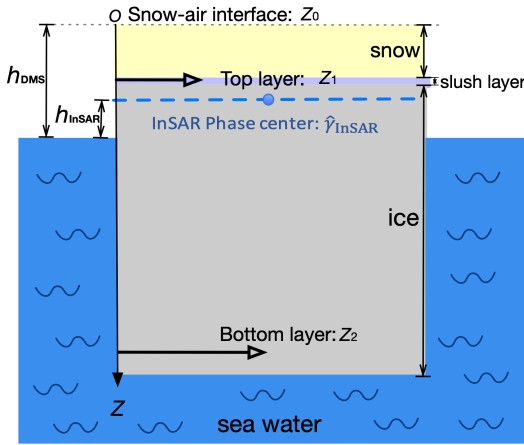

**Figure 5.** The schematic of the proposed TLPV model for sea ice (Huang et al., 2021).

the top layer into one Dirac delta (Huang et al., 2021):

$$
\begin{aligned}
&\tilde{\gamma}_{\text{InSAR}} \\
&\approx e^{i\phi_0} \frac{1 \cdot e^{i\phi_1} + m \cdot e^{i\phi_2}}{1+m} \\
&= e^{i\phi_0} \tilde{\gamma}_{\text{mod}}(m, z_1, z_2) \\
&= e^{i\phi_0} \tilde{\gamma}_{\text{mod}}(m, z_1, h_v)
\end{aligned}
\tag{8}
$$

where $\phi_0$ is the topographic phase at the snow-air interface, $\phi_1 = \kappa_{z\_\text{vol}} z_1$, $\phi_2 = \kappa_{z\_\text{vol}} z_2$, $z_1$ and $z_2$ are the locations of the layers, respectively. $h_v = z_1 - z_2$ refers to the depth between the top and bottom layers. It is worth noting that the bottom layer may not always be at the ice-water interface. In certain situations, there might be a lower basal saline ice layer, inducing strong surface scattering from the ice-basal layer interface (Nghiem et al., 2022). This basal saline layer contains brine inclusions with higher salinity, transitioning towards the ice-seawater interface (Tison et al., 2008). $\kappa_{z\_\text{vol}}$ is the vertical wavenumber in the volume which depends on the InSAR configuration such as HoA and the incidence angle, and the dielectric constant of the volume (Dall, 2007; Sharma et al., 2012; Huang et al., 2021). $m$ refers to the layer-to-layer scattering ratio, which is the backscattering power ratio between the top and bottom layers:

$$
m = \frac{\sigma_{bottom}(\boldsymbol{\omega})}{\sigma_{top}(\boldsymbol{\omega})}
\tag{9}
$$

where $\sigma_{top}\boldsymbol{\omega})$ and $\sigma_{bottom}(\boldsymbol{\omega})$ denotes the backscattering power from the top and bottom interface, respectively, at a given polarization $\boldsymbol{\omega}$. $m$ potentially reveals the relative importance of scattering from these interfaces, depending on factors like interface roughness, dielectric constant, and radar polarization. A larger value of $m$ signifies that surface scattering from the bottom layer predominates, while a smaller $m$ indicates that surface scattering from the top layer is more significant.

The aim is to estimate $\phi_0$ and convert it into height ($h_{\text{mod}}$), which is the total freeboard of LPI. When fixing the origin at the air-snow interface, $z_1$ is equivalent to snow depth, which can be obtained from the AMSR Level-3 data (Meier et al., 2018).

However, Eq. (8) still contains two unknown variables, $m$ and $h_v$, preventing direct estimation of $\phi_0$. Therefore, we develop a PolInSAR height retrieval module to invert the TLPV model and estimate $m$ and $h_v$, shown in Fig. 4(c). We first establish an empirical relation (RF regression) between SAR features and the true values of $m$ and $h_v$ which can be derived using DMS DEM as a priori information. Specifically, we simulate the interferometric phase ($\phi_{\mathrm{DMS}} = \phi_0$) from the height ($h_{\mathrm{DMS}}$) using Eq. (3) tailored to the specific InSAR configuration with $h_a$ given in Table 1. With snow depth $z_1$ and $\tilde{\gamma}_{\mathrm{InSAR}}$ from AMSR data and InSAR observations, respectively, $m$ and $h_v$ values are derived by inverting Eq. (8) and used as true values for training the RF regressor.

We use Gini importance to rank the seven features for regression, selecting the top five predictors for estimating $\hat{m}$ and $\hat{h}_v$: $\sigma_{\mathrm{HH}}$, $h_{\mathrm{InSAR}}$, $\gamma_{\mathrm{InSAR}}$, $R_{\mathrm{Pauli}}$, and $\phi_{\mathrm{coPol}}$. The ranking of all the SAR features is given in the appendix Fig. A4. The selected features, together with the true $m$ and $h_v$, form the sample set. 75% is used for training the RF regressor. Note that the RF regressor is trained using the same sample set as the sea ice classification.

The well-trained RF regressor is subsequently utilized to estimate $\hat{m}$ and $\hat{h}_v$ for SAR scenes that do not overlap with DMS measurements. Selected features from SAR images, along with $z_1$ from ancillary data, serve as inputs to the RF regression model for estimating $\hat{m}$ and $\hat{h}_v$. Subsequently, the topographic phase $\hat{\phi}_0$ can be derived by solving Eq. (8), and transformed into total freeboard $h_{\mathrm{mod}}$ using Eq. (3). For SAR scenes overlaid with DMS measurements, validation of height retrieval accuracy over the testing subset (25%) will be given in Sect. 4.1.

## 4 Results

Following the two-step approach developed in Sect. 3, this section obtains the SAR-derived DEM from 162 dual-pol InSAR pairs that cover the sea ice in the Weddell and Ross Seas. We verify the accuracy of the SAR-derived DEM. We further analyze the variation of total freeboard and roughness along the southward direction and examine the statistical characteristics of sea ice DEM across various geographic regions.

### 4.1 Sea ice topography retrieval and validation

The proposed two-step approach for sea ice DEM retrieval is visually and quantitatively validated based on the four scenes overlapped with DMS measurements. The SAR backscattering intensities over the four scenes are displayed in the left column in Fig. 6. In the first step, the proposed classification scheme demonstrates good performance on the testing set, with an accuracy of 0.84 and a confusion matrix presented in Fig. 7(a). The classifier is then applied to the entire SAR images, including the region not overlapped by DMS DEM, and the classified maps are shown in the middle column of Fig. 6.

In the second step, the sea ice DEM ($h_{\mathrm{mod\_SAR}}$) is obtained by merging $h_{\mathrm{mod}}$ and $h_{\mathrm{InSAR}}$ over LPI and SPI. Note that $h_{\mathrm{mod\_SAR}}$ represents the total freeboard relative to the water surface retrieved from the pixel at $10 \times 10\,\mathrm{m}$ spacing size. The retrieved sea ice DEMs are compared with $h_{\mathrm{DMS}}$ over the testing set, shown in Fig. 7(b). The RMSE between $h_{\mathrm{mod\_SAR}}$ and $h_{\mathrm{DMS}}$ is $0.26\,\mathrm{m}$. This result is promising as Dierking et al. (2017) suggested the satisfactory accuracy for a sea ice DEM being

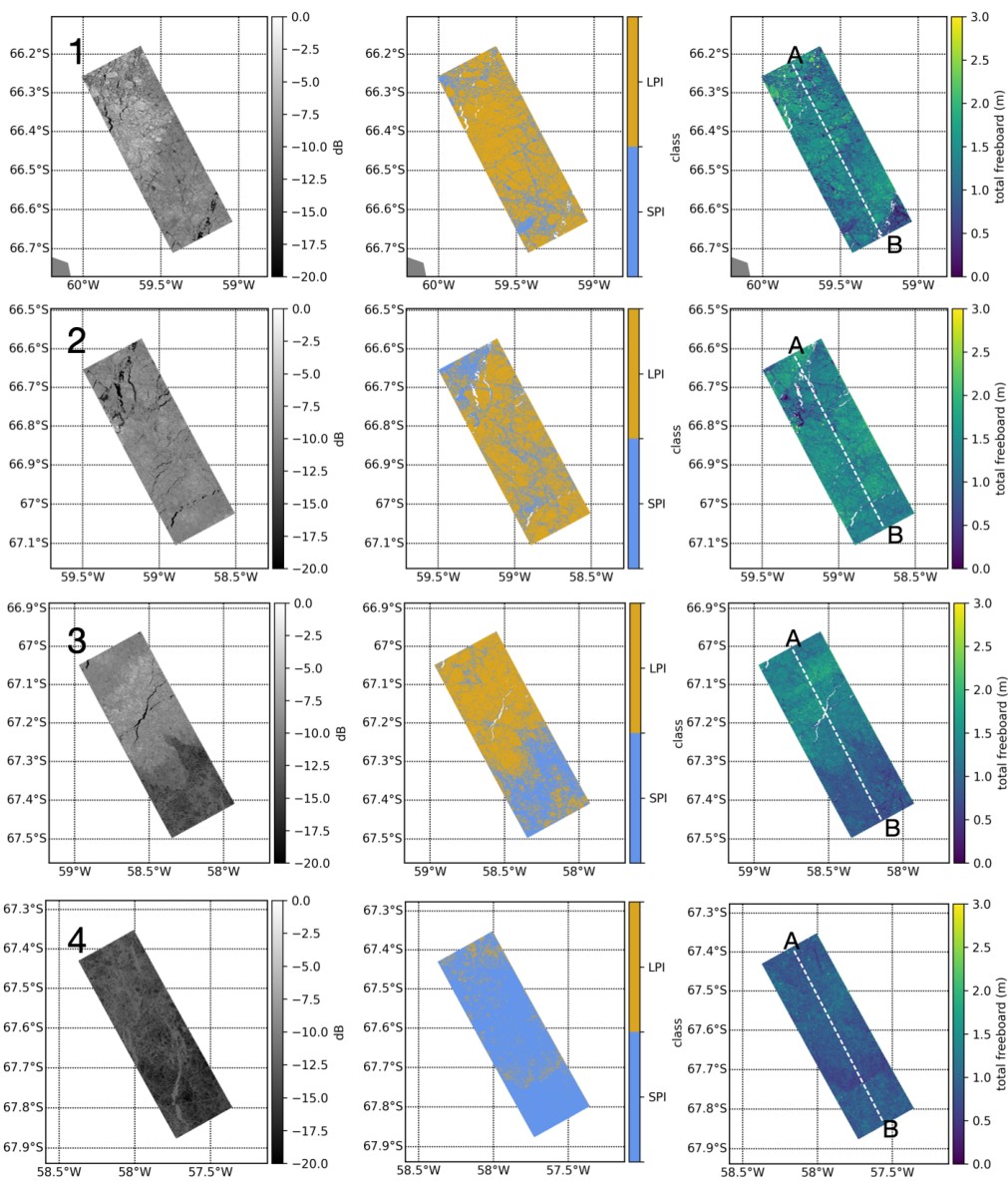

**Figure 6.** First column: SAR backscattering intensity in HH polarization. Second column: sea ice classification. Third column: sea ice DEM ($h_{\mathrm{mod\_SAR}}$) over the four scenarios. Each row corresponds to Scene No.1-4 in Fig. 2, respectively. The spatial resolution is $10 \times 10\,\mathrm{m}$. The void pixels in the second and third columns represent water areas excluded from processing due to $\gamma_{\mathrm{InSAR}} < 0.3$. The white dashed line indicates the flight track overlapped by the DEM DEM ($h_{\mathrm{DMS}}$)

.

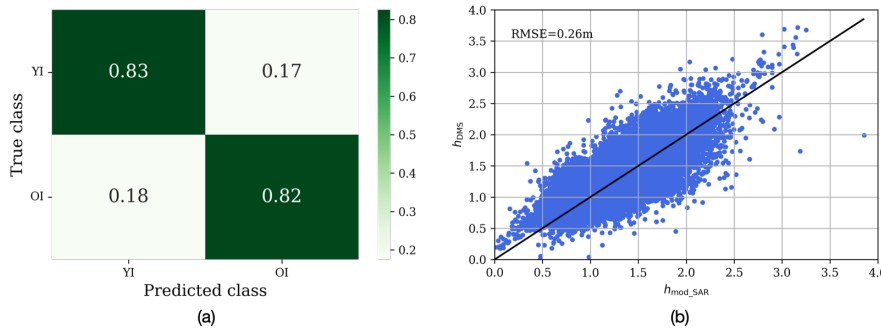

**Figure 7.** (a) Confusion matrix for sea ice classification. (b) Comparison between the reference height and the derived height over LPI and SPI.

less than $0.3\,\mathrm{m}$. Note that the average RMSE value of LPI without compensating the penetration bias is $\sim 1.10\,\mathrm{m}$ (Huang et al., 2021).

The $h_{\mathrm{mod\_SAR}}$ over the entire SAR images are displayed in the right column of Fig. 6. For each scene, the white dash
line delineates a $50\,\mathrm{km} \times 100\,\mathrm{m}$ strip overlapped with DMS DEM. By extracting the values at the center of the strip, the height profiles are presented in Fig. 8, where $h_{\mathrm{mod\_SAR}}$ performs good agreement with the reference data ($h_{\mathrm{DMS}}$) and well capture the topographic variation. Considering that $h_{\mathrm{DMS}}$ already contains an uncertainty of $0.2\,\mathrm{m}$ (Dotson and Arvesen., 2012, updated 2014), these results prove the effectiveness of the proposed two-step approach for sea ice DEM retrieval over both SPI and LPI.

## 4.2 Sea ice topography along the southwards direction

We obtain sea ice DEM over the total 162 images using the two-step approach. A visualization of the derived sea ice DEM can be found in the appendix Fig. A5 and A6. The derived DEM is downsampled to a resolution of $500\,\mathrm{m}$, utilized in the subsequent analyses.

The northernmost location on each segment is selected and marked with star symbol in Fig. 1. Subsequently, we characterize the variation of sea ice category, total freeboard, and roughness moving southward, using distances relative to the northernmost
locations and averaging over every $100\,\mathrm{km}$ interval. These topographic variations along the distance are illustrated in Fig. 9- 11.

The first column shows the LPI percentages estimated from the proposed two-step approach and compared with the Ice Charts. The overall trend of estimated LPI percentages correlates well with dominant ice types and ice concentrations from the Ice Charts across most segments (W1-U, W1-L, W2-U, W4, W5-U, W5-L, and R5). Specifically, W1-U and W1-L are explained as two examples. W1-U from $0-120\,\mathrm{km}$ is covered by $100\%$ MYI, where the LPI percentage reaches its highest
value ($58\%$). As the dominant ice transitions from MYI to FYI from $120-400\,\mathrm{km}$, the LPI percentage decreases accordingly. The dominance of MYI ice from $400-600\,\mathrm{km}$ also corresponds to the increasing LPI percentage. For W1-L, where the distance between $500-600\,\mathrm{km}$ is covered by $100\%$ MYI, and the LPI percentage peaks before decreasing after $650\,\mathrm{km}$ distance as FYI becomes dominant. The lowest LPI percentage is found around $1200\,\mathrm{km}$, consistent with the occurrence of TI at this distance.

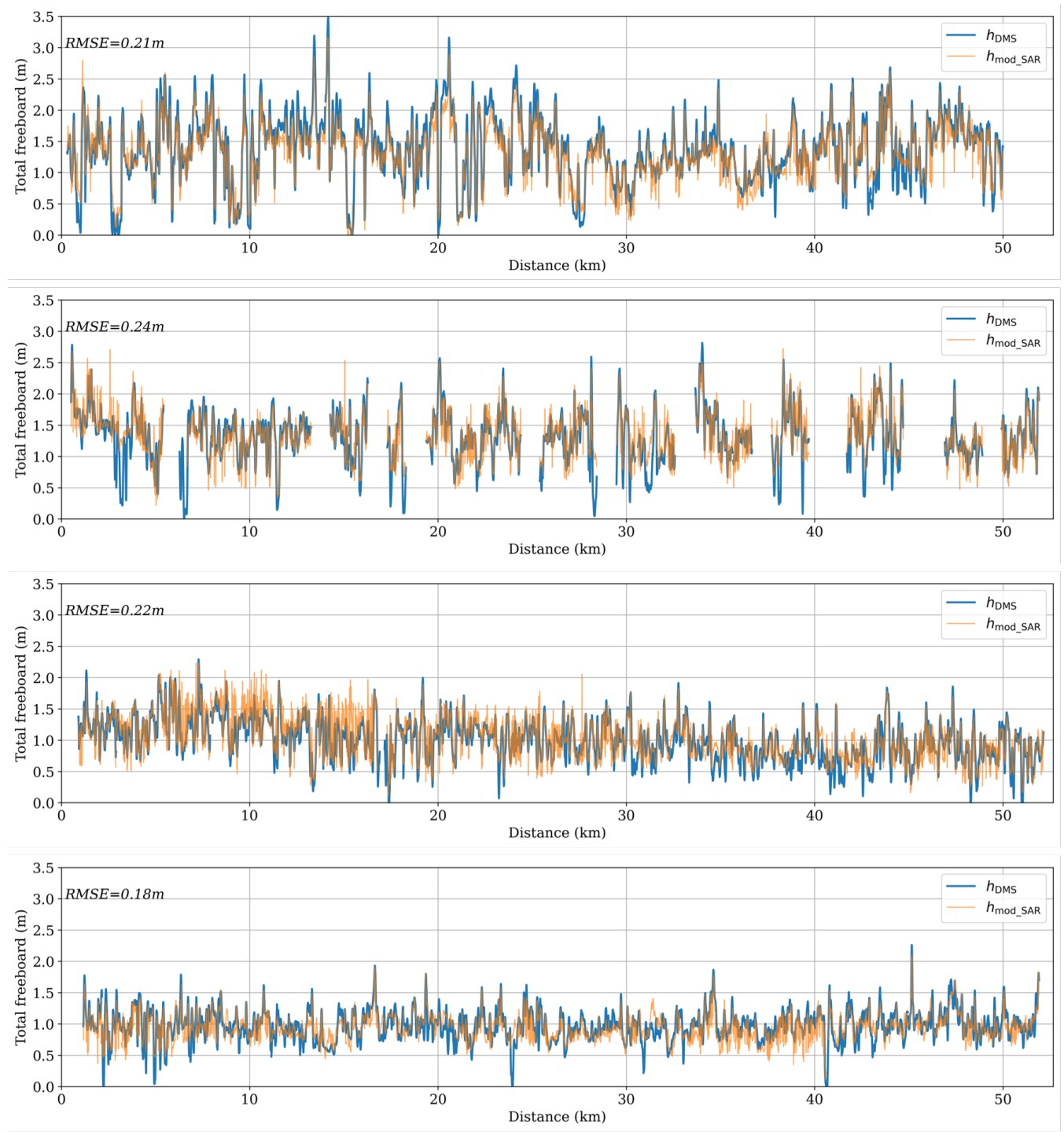

**Figure 8.** Comparison between the total freeboard profiles ($h_{\mathrm{mod\_SAR}}$) from the proposed method and the DMS DEM ($h_{\mathrm{DMS}}$) along the dotted line (from A to B) over Scene No.1-4 in Fig. 6(c). The spatial resolution is $10 \times 10\,\mathrm{m}$. The data gaps are the water areas and segments that were excluded due to the absence of distinctive features, ensuring co-registration quality.

The observed similar trend can be explained by the general assumption that MYI is thicker and less saline, allowing for deeper radar penetration compared to FYI and TI. However, penetration depth is influenced by various factors not only ice age, but also ice salinity, snow condition, flooding effects, and temperature. This explains the discrepancies for other segments (W2-L, W3-U, W3-L, R1-U, R1-L). Furthermore, discrepancies can also be attributed to differences in spatial resolution and temporal gaps between Ice Charts and SAR imagery, considering the dynamic nature of sea ice.

It is essential to clarify that we utilized Ice Charts data as external information to interpret the classification results and spatial variation of topography. However, we did not use Ice Charts to quantitatively validate the proposed method. For validation purposes, we conducted pixel-by-pixel comparisons using co-registered DMS data.

The second and third columns in Fig. 9- 11 display the distance dependency of total freeboard ($h_{\mathrm{mod\_SAR}}$) and roughness ($\sigma_R$), respectively. Sea ice roughness is the standard deviation of the total freeboard within a $50 \times 50\,\mathrm{m}$ area. For each $100\,\mathrm{km}$-distance interval, we calculate and display the average and median values, as well as the first and third quartiles of $h_{\mathrm{mod\_SAR}}$ and $\sigma_R$ using boxplots.

The Ice Charts are also used to validate and explain the topographic variation of sea ice. In general, the region with thicker ice (e.g., MYI) is anticipated to display higher total freeboard or larger roughness compared to the area with thinner ice, such as FYI and TI. This hypothesis is substantiated by the agreement between topographical variations (total freeboard and roughness) and ice types observed in Fig. 9 to 11 across most segments, with the exception of W3-L and W5-L. The sea ice is identified as MYI between $450 - 750\,\mathrm{km}$ in W3-L and $650 - 950\,\mathrm{km}$ in W5-L. However, neither total freeboard nor roughness significantly increases within these specific ranges. Minor discrepancies also persist, for instance, in W1-U and W2-L, where there is no clear reduction in either total freeboard or roughness when FYI is present at around $600\,\mathrm{km}$. These discrepancies may arise due to the local cases where rough FYI exhibits greater roughness than smooth level MYI. FYI may also show higher elevations when covered by very thick snow. In addition, considering that the Ice Charts data are weekly products, the inconsistencies could be attributed to mis-coregistration caused by sea ice drift during the time lag between the Ice Charts and the SAR images.

In the northwestern Weddell Sea, we observe that the sea ice near the Antarctic Peninsula (AP) in the W1-U and W2-U segments exhibits the highest average total freeboard (mean $> 0.7\,\mathrm{m}$) and roughness (mean $= 0.19\,\mathrm{m}$), shown in the first and third rows in Fig. 9. This observation aligns with a previous study using OIB Airborne Topographic Mapper (ATM) data from November 14 and 22, 2017 (Wang et al., 2020), which has reported that the total freeboard near the eastern AP ranges from $1.5 - 2.5\,\mathrm{m}$. Moving outwards from the AP, the total freeboard and roughness along W1-U and W2-U demonstrate a sharp decrease within approximately $0 - 200\,\mathrm{km}$ before gradually increasing as it heads southward. Similar trends are observed in W3-U (first row in Fig. 10) and W5-U (first row in Fig. 11), with a more subtle decrease in total freeboard within the $0 - 200\,\mathrm{km}$ range, compared to W1-U and W2-U, followed by a southward increase. Conversely, in the initial $0 - 200\,\mathrm{km}$ of W4 (third row in Fig. 10), there's no observed decrease in total freeboard or roughness. Instead, a gradual increase in both parameters is evident as one moves southward, consistent with the dominance of MYI beyond $100\,\mathrm{km}$ from the Ice Charts data. In the southeastern region, segments W1-L, W2-L, W3-L, and W5-L exhibit similar patterns in the topographic variation, with total freeboard and roughness generally decreasing towards the south as they approach the Coats Land (see location in Fig. 1). This trend can be explained by the increasing occurrence of FYI or TI beyond around $1000\,\mathrm{km}$ from the Ice Charts data.

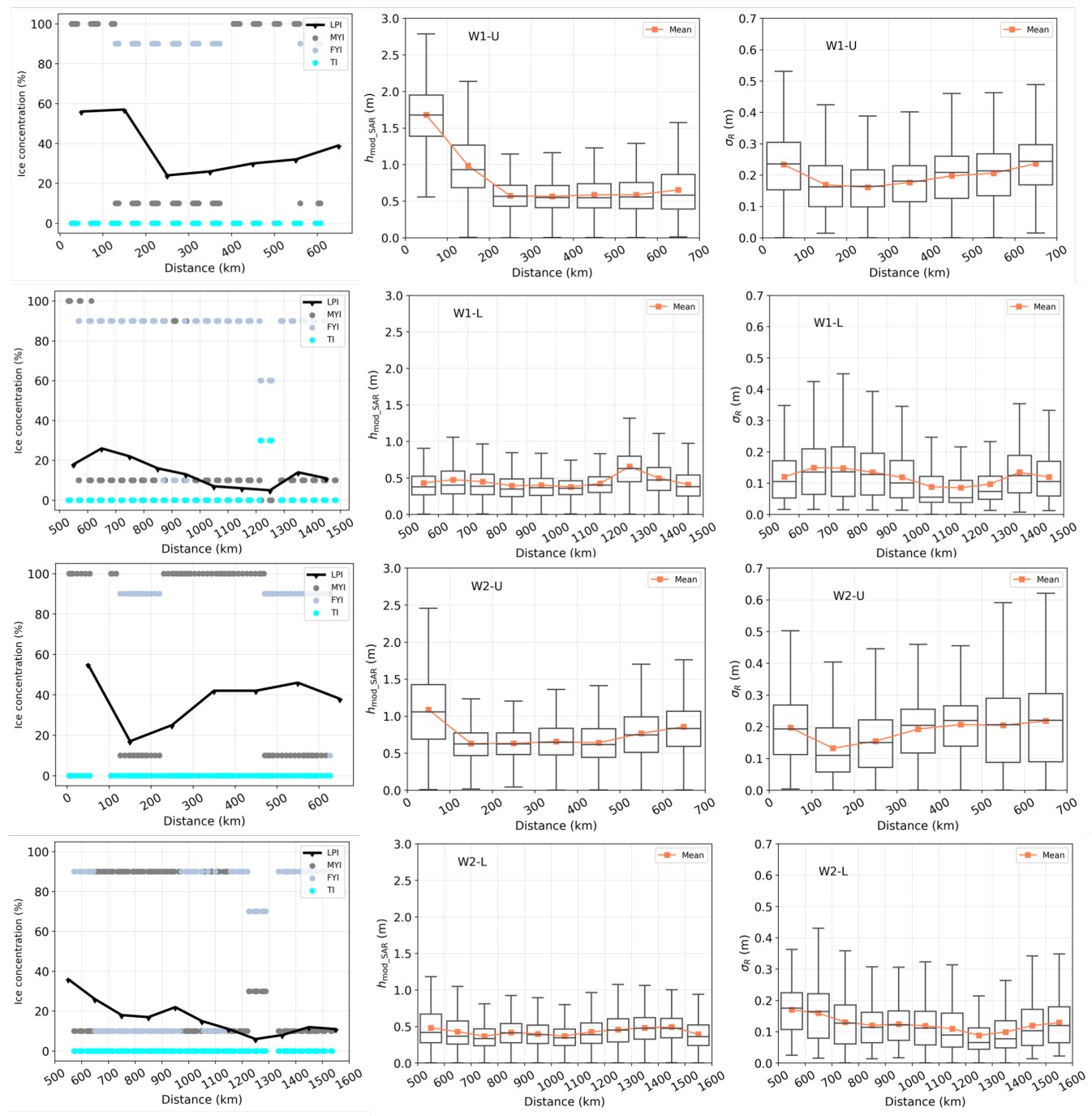

**Figure 9.** Sea ice characteristics along the southwards direction along W1 and W2 segments. The blue line in the first column displays the LPI percentages derived from SAR images, and the blue dot indicates the ice types obtained from the Ice Charts. The second and third columns plot the total freeboard ($h_{\mathrm{mod\_SAR}}$) and roughness ($\sigma_R$), respectively. Distance is measured from the northernmost SAR image reference point towards the south. The orange line denotes the average values of $h_{\mathrm{mod_{SAR}}}$ and $\sigma_R$. The box's upper and lower boundaries represent the first (Q1) and third (Q3) quartiles, while the upper (lower) whisker extends to the last (first) sample outside of Q3 $\pm 1.5\times$(Q3-Q1).

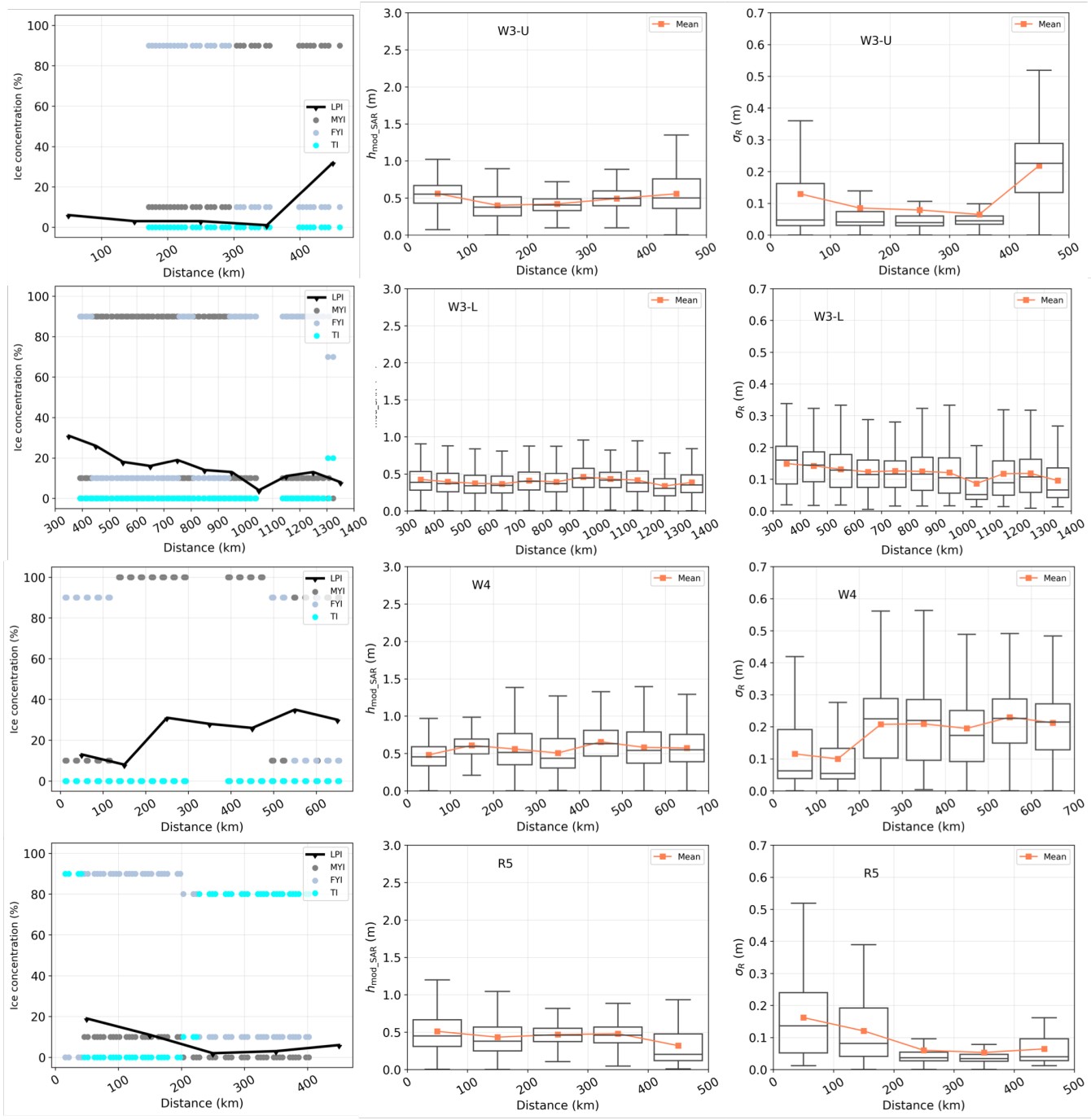

**Figure 10.** Sea ice characteristics along the southwards direction along W3, W4, and R5 segments. The blue line in the first column displays the LPI percentages derived from SAR images, and the blue dot indicates the ice types obtained from the Ice Charts. The second and third columns plot the total freeboard ($h_{\mathrm{mod\_SAR}}$) and roughness ($\sigma_R$), respectively.

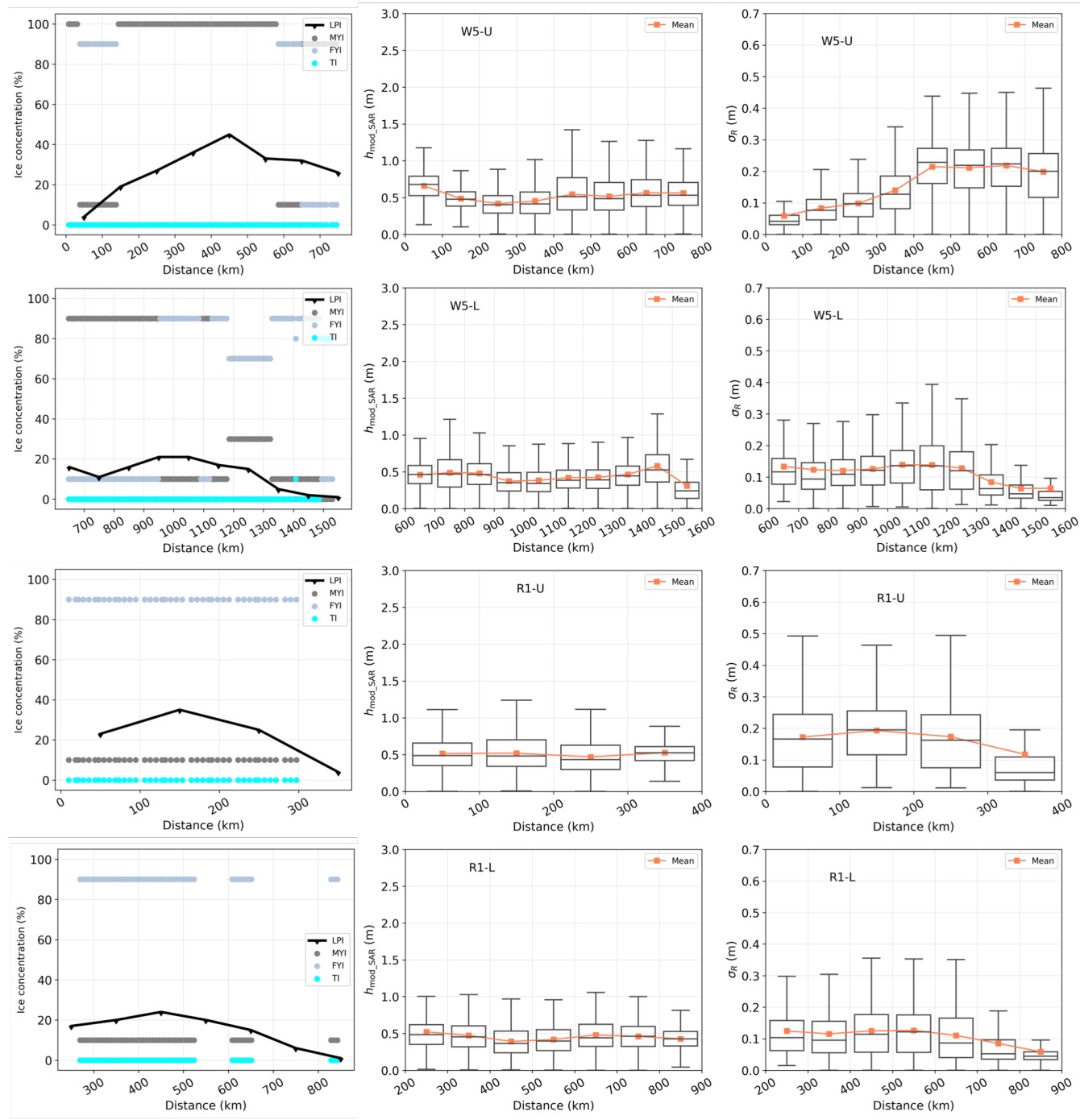

**Figure 11.** Sea ice characteristics along the southwards direction along W5 and R1 segments. The blue line in the first column displays the LPI percentages derived from SAR images, and the blue dot indicates the ice types obtained from the Ice Charts. The second and third columns plot the total freeboard ($h_{\mathrm{mod\_SAR}}$) and roughness ($\sigma_R$), respectively.

The observed variation in sea ice topography can result from the formation and dynamics of sea ice in the East Weddell (E-Wedd) and West Weddell (W-Wedd) regions, which are defined by specific longitude ranges: E-Wedd encompasses 15°E to 40°W, while W-Wedd extends from 40°W to 62°W. Segments of W1-U, W2-U, W3-U, W4, and W5-U are located within the W-Wedd, which has been reported for the presence of MYI in the Antarctic (Lange and Eicken, 1991). Sea ice initially forms in the eastern region and then circulates clockwise within the cyclonic gyre of the southern Weddell Sea. Later, older sea ice drifts outward northwestern (Kacimi and Kwok, 2020). The sea ice undergoes thickening and deformation as it drifts (Vernet et al., 2019; Kacimi and Kwok, 2020), resulting in increased total freeboard and greater roughness in the northwestern Weddell Sea.

In the Western Ross Sea, the sea ice along the R5 segment (last row in Fig. 10) exhibits greater total freeboard and roughness near Terra Nova Bay (see Fig. 1 for the location), with decreasing southeastward. This observation aligns with the transition of dominant ice types from FYI to TI in that direction. This observation is also consistent with recent research (Rack et al., 2021), where airborne measurements in November 2017 revealed deformed sea ice exceeding $10\,\mathrm{m}$ in thickness within the first $100\,\mathrm{km}$ south of Terra Nova Bay, and thinner ice was observed towards the southeastern area near McMurdo Sound (see Fig. 1 for the location). Satellite data also confirmed a region of thinner ice influenced by the Ross Sea Polynya, with thicker ice located westward (Kurtz and Markus, 2012). The observed pattern can be attributed to significant deformation in the Western Ross Sea caused by wind-driven shearing, rafting, and ridging within a convergent sea ice regime (Hollands and Dierking, 2016). This deformation leads to potentially thicker sea ice compared to the eastern part (Rack et al., 2021).

For the R1 segment located in the Eastern Ross Sea (third and fourth rows in Fig. 11), although the sea ice exhibits relatively stable total freeboard, which agrees with the consistent presence of predominantly FYI, the roughness decreases towards the southeastern. This may be attributed to the influence of ocean circulation, considering that the R1 segment is situated farther from the land than the other segments. The variation of the roughness along the R1 segment suggests that ice topography provides add-on information that can be useful to be integrated into the operational ice charting. Furthermore, since the edges of ice floes with open water between the floes can also contribute to the ice roughness, combining ice topography with ice concentration can help characterize the sea ice more comprehensively.

## 4.3 Regional variation of sea ice topography

Figure 12(a) displays the topographic variation across different segments. We present the average and median values, as well as the first and third quartiles of total freeboard and roughness using boxplots. The mean values are listed in Table 3. Additionally, the percentages of the three ice types within each segment, calculated from the Ice Charts for reference, are presented in Fig. 12(b).

Generally, sea ice in the northwestern Weddell Sea (W1-U, W2-U, W3-U, W4, W5-U) exhibits higher average total freeboard ($> 0.5\,\mathrm{m}$) compared to that in the southeastern Weddell Sea and the Ross Sea (W1-L, W2-L, W3-L, W5-L, R1-U, R1-L, R5), see detailed values in Table 3. W1-U and W2-U exhibit the highest average total freeboard of $0.8\,\mathrm{m}$ and $0.72\,\mathrm{m}$, respectively, along with the largest average roughness of $0.19\,\mathrm{m}$. This is comparable with the total freeboard retrieved from ICESat-2 (Kacimi and Kwok, 2020), reporting an average of $0.6 - 0.7\,\mathrm{m}$ total freeboard nearby the Eastern AP between April 1 and

**Table 3.** Average total freeboard and roughness for each segment, as well as the Kolmogorov-Smirnov (KS) values between the observed total freeboard and modeled distributions (i.e., exponential normal and log-normal). The smaller KS value is in **bold**.

| segment | Mean total freeboard (m) | Mean roughness (m) | KS exp-normal | KS log-normal |
|---------|-------------------------|--------------------|---------------|---------------|
| W1-U | 0.80 | 0.19 | **0.083** | 0.106 |
| W1-L | 0.46 | 0.12 | 0.064 | **0.05** |
| W2-U | 0.72 | 0.19 | **0.052** | 0.07 |
| W2-L | 0.42 | 0.12 | 0.079 | **0.054** |
| W3-U | 0.50 | 0.11 | **0.113** | 0.2 |
| W3-L | 0.39 | 0.12 | 0.065 | **0.05** |
| W4 | 0.57 | 0.18 | **0.035** | 0.072 |
| W5-U | 0.52 | 0.16 | **0.074** | 0.076 |
| W5-L | 0.44 | 0.11 | **0.038** | 0.078 |
| R1-U | 0.49 | 0.18 | 0.053 | **0.049** |
| R1-L | 0.45 | 0.11 | **0.049** | 0.106 |
| R5 | 0.46 | 0.10 | 0.052 | **0.039** |
| Overall | | | **0.063** | 0.079 |

November 16, 2019. W4 and W5-U show average total freeboard of $0.57\,\mathrm{m}$ and $0.52\,\mathrm{m}$, and average roughness values of $0.18\,\mathrm{m}$ and $0.16\,\mathrm{m}$, respectively. The above topographic values (i.e., total freeboard and roughness) are consistent with the ice types presented in Fig. 12(b), where W1-U, W2-U, W4, and W5-U exhibit a substantial proportion ($\geq 57\%$) of MYI, known for greater total freeboard and roughness characteristics. W3-U, which consists of $47\%$ MYI, exhibits a total freeboard of $0.50\,\mathrm{m}$ but with relatively lower roughness at $0.11\,\mathrm{m}$, suggesting the possibility of a smooth snow-air interface over older and thicker ice.

For the segments in the southeastern Weddell Sea and the Ross Sea (W1-L, W2-L, W3-L, W5-L, R1-L, and R5), the average total freeboard remains below $0.46\,\mathrm{m}$ and roughness around $0.11\,\mathrm{m}$. The reduced average total freeboard and roughness correspond to ice types with fewer MYI percentages ($\leq 52\%$) and greater amounts of FYI and TI ($> 52\%$). R1-U demonstrates an average total freeboard of $0.49\,\mathrm{m}$ and roughness of $0.18\,\mathrm{m}$, with the presence of predominantly FYI throughout the region. This observation suggests a plausible scenario of a rougher snow-air interface over younger and thinner ice (Tin and Jeffries, 2001; Tian et al., 2020).

## 4.4 Statistical analyses of sea ice topography

Studies on sea ice topography in the Arctic have extensively examined the applicability of statistical distributions such as the log-normal distribution (Landy et al., 2020; Duncan and Farrell, 2022) and the exponentially modified normal (exp-normal) distribution (Yi et al., 2022). However, there remains a gap in understanding the most suitable distribution models for describing

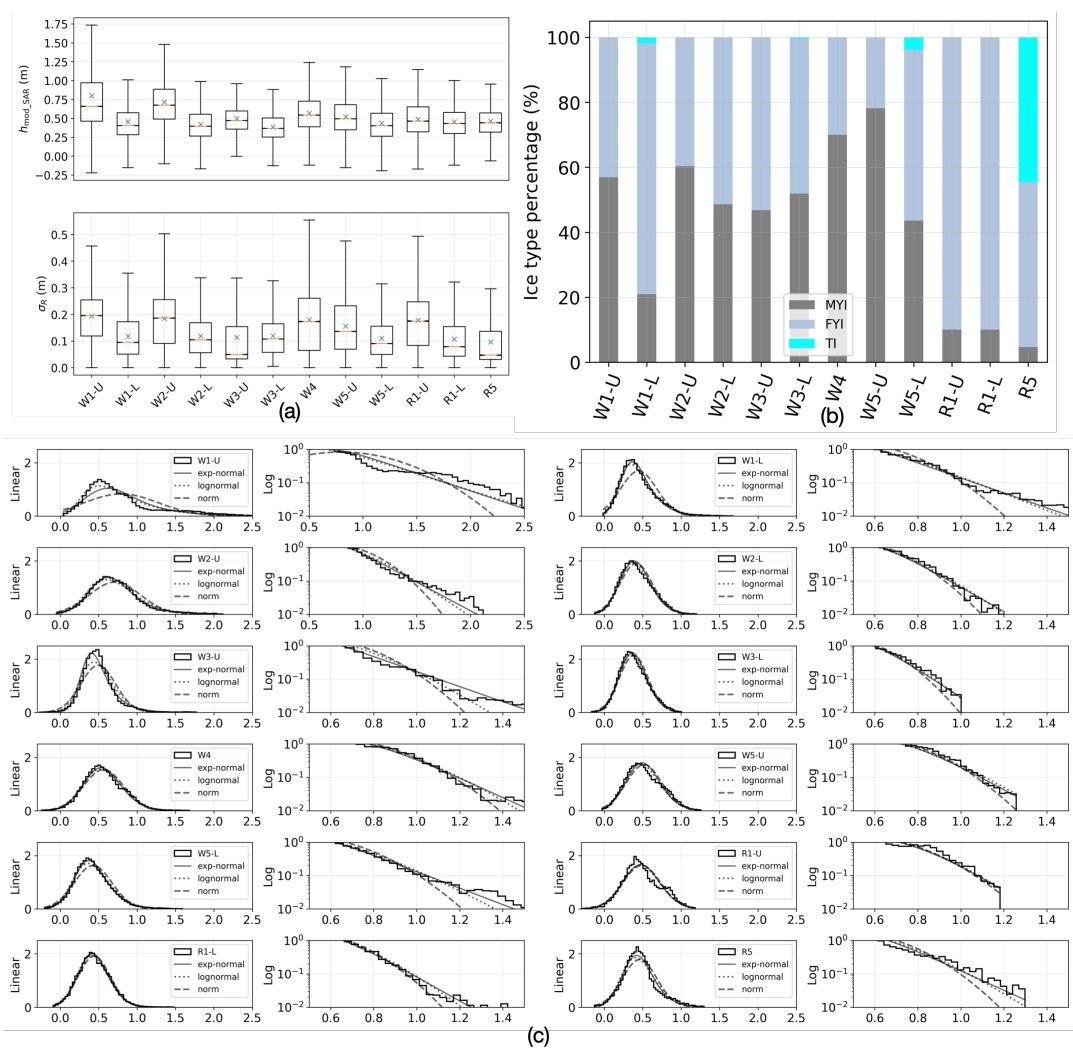

**Figure 12.** (a) Total freeboard ($h_{\mathrm{mod\_SAR}}$) and roughness ($\sigma_R$) derived from SAR images across the 12 segments. (b) The percentage of multi-year ice (MYI), first-year ice (FYI), and thin ice (TI) from the Ice Charts. (c) Probability density function (PDF) of derived total freeboard ($h_{\mathrm{mod\_SAR}}$) and their fits to the exponential-normal, log-normal, and normal distributions.

the total freeboard of Antarctic sea ice. We aim to address this gap by evaluating three distribution models: Gaussian, log-normal, and exp-normal, to determine the most appropriate probability density function (PDF) for describing the sea-ice total freeboard across segments.

The PDF of the Gaussian distribution with mean $\mu_g$ and standard deviation $\sigma_g$ is defined as:

$$G(x) = \frac{1}{\sqrt{2\pi\sigma_g^2}} e^{-\frac{(x-\mu_g)^2}{2\sigma_g^2}} \tag{10}$$

The PDF of the log-normal distribution with mean $e^{(\mu_l + \sigma_l^2/2)}$ and variance $e^{2\mu_l + \sigma_l^2}\left(e^{\sigma_l^2} - 1\right)$ follows (Gaddum, 1945):

$$LG(x) = \frac{1}{x\sigma_l\sqrt{2\pi}} e^{-\frac{(\ln(x)-\mu_l)^2}{2\sigma_l^2}} \tag{11}$$

The PDF of the exp-normal distribution with mean $\mu_e + 1/\lambda$ and variance $\sigma_e^2 + 1/\lambda^2$ is given as (Foley and Dorsey, 1984):

$$EMG(x) = \frac{\lambda}{2} e^{\frac{\lambda}{2}(2\mu_e + \lambda\sigma_e^2 - 2x)} \text{erfc}(\frac{\mu_e + \lambda\sigma_e^2 - x}{\sqrt{2}\sigma_e}) \tag{12}$$

where the erfc($\cdot$) is the complementary error function with $\text{erfc}(x) = \frac{2}{\sqrt{\pi}}\int_x^\infty e^{-t^2}\,dt$.

The observed and modeled distributions of total freeboard over each segment are depicted in the left column of Fig. 12(c). The observed distributions of all segments exhibit asymmetrical with longer tails. A closer examination of the tail regions (right column in Fig. 12(c)) reveals significant deviations from the Gaussian distribution, particularly in segments W1-U and W2-U, which are covered by deformed and thicker sea ice. The observed non-Gaussian nature of total freeboard distribution aligns with the previous studies (Hughes, 1991; Davis and Wadhams, 1995; Castellani et al., 2014; Landy et al., 2019; Huang et al., 2021). To quantitatively evaluate the fit of non-Gaussian distributions (i.e., log-normal and exp-normal) to the observed total freeboard, we employ the Kolmogorov-Smirnov (KS) test (Massey Jr, 1951). This test measures the goodness of fitting by calculating the distance between the observed distribution function and the theoretical cumulative distribution function. The values of the KS test are given in Table 3, where a lower value indicates a better fit.

In the northwestern Weddell Sea, where the segments (W1-U, W2-U, W3-U, W4, and W5-U) have average total freeboard greater than $0.5\,\mathrm{m}$, the exp-normal distribution demonstrates superior fitting performance, as evidenced by smaller KS values. This can be attributed to the exp-normal distribution's incorporation of an exponential component, which enables a better fit to data with heavy or long tails compared to the log-normal distribution. Consequently, the exp-normal distribution is better suited for characterizing the statistics of older and thicker sea ice, which often involves strong deformation and exhibits significant total freeboard.

In the southern Weddell Sea and the Ross Sea, segments average below $0.5\,\mathrm{m}$ in total freeboard, with varying fits between exp-normal and log-normal distributions. The log-normal distribution exhibits a better fit for W1-L, W2-L, W3-L, R1-U, and R5, while the exp-normal distribution is more appropriate for W5-L and R1-L. This observation suggests that the two distributions perform comparably in characterizing the total freeboard of younger and thinner sea ice.

Evaluating the overall performance across all segments, the exp-normal distribution outperforms the log-normal distribution, as indicated by a smaller average KS value of $0.063$.

## 5 Discussion

### 5.1 Factors affecting the model performance

In the proposed two-step method (Fig 4), we obtain snow depth from the AMSR products (Meier et al., 2018), which represents a five-day running average of snow depth over sea ice. Due to the limited spatial ($12.5\,\text{km}$) and temporal resolution in the snow depth data, we assume a constant value of snow depth across one SAR image. Hence, for each SAR acquisition covering a spatial extent of $50 \times 19\,\text{km}$, we compute the mean snow depth and utilize it as input parameter $z_1$ in the TLPV model.

In Fig. 8, it appears the derived total freeboard ($h_{\text{mod\_SAR}}$) underestimates some high and low peaks of the reference data ($h_{\text{DMS}}$). One factor contributing to the underestimation of total freeboard could be the assumption of a constant average snow depth over one SAR scene. Using a single average value of snow depth may lead to an underestimation of snow depth in high-peak areas such as ridges, consequently resulting in an underestimation of the total freeboard. Our prior study (Huang et al., 2021) demonstrated a mean difference of $0.31\,\text{m}$ in the derived total freeboard due to snow depth variations from $0.05$ to $0.75\,\text{m}$ over Scene No.1, highlighting the impact on peak estimation. In the future, it would be interesting to adapt the proposed method over the test sites co-locating with available high-resolution snow depth measurements.

Another factor that could potentially lead to the underestimation of high and low peaks is the residual shift between the SAR and DMS images. Although we carefully co-registered the four SAR scenes with the DMS data, the co-registration can not be perfect. In the process, we divided the entire overlapped transect into small patches (each corresponding to $100 \times 1000\,\text{m}$). We assumed the same drift location over one patch and no rotation; thus, only one shift vector was used for co-registration over each patch. This could result in small residual shifts when the ice floes or features do not drift at the same velocity or involve rotations within the patch. The presence of low- and high-peak ice features with narrow sizes spanning just a few pixels, poses a challenge. Even slight residual shifts, as small as 1-2 pixels, can lead to loss or misalignment of peak structures in SAR images. Consequently, these slightly misaligned SAR images input into the proposed model may result in an underestimation of the total freeboard.

### 5.2 Compare the InSAR-derived total freeboard with existing study

Wang et al. (2020) calculated the mean total freeboard in the Weddell Sea using the IceBridge Laser Altimetry. In this sub-section, we conduct a visual comparison between Wang et al. (2020)'s result (Fig. 13) with the four segments (W2-U, W2-L, W3-U, and W3-L) in our study (Fig. 9 and 10). Note that the window size is ten-of-kilometer scales in Wang et al. (2020)'s study, significantly exceeding the $500 \times 500\,\text{m}$ window size we used.

We denote the region in Fig. 13 with latitude $< 70°\text{S}$ ($> 70°\text{S}$) as the northern (southern) track. The Northern and Southern Track-W segments are partially overlaid with W2-U and W2-L, respectively. In our study, the total freeboard of W2-U and

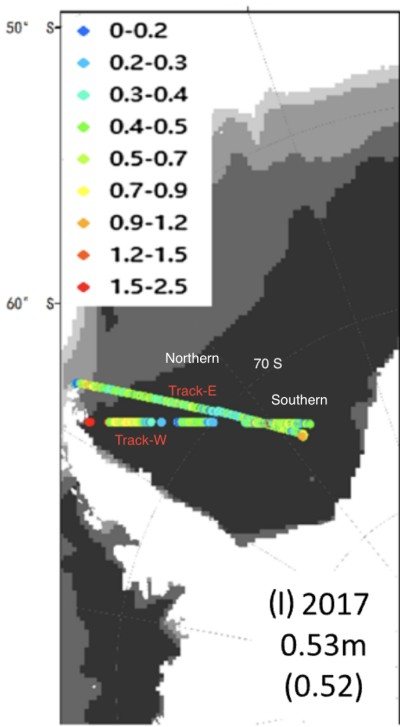

**Figure 13.** The total freeboard calculated from IceBridge Laser Altimetry (copy of Fig.6l by Wang et al. (2020)). We labeled the two tracks Track-W and Track-E. The region with latitude $< 70°$S $(> 70°$S$)$ is referred to as the northern (southern) track.

W2-L in Fig. 9 reaches a mean value of $\sim 1$ m and 75% percentile value of $\sim 1.5$ m within the first $100$ km, which agrees with
the red dot in Fig. 13 Track-W. Then, the total freeboard goes down to a mean value as $\sim 0.7$ m and 75% percentile value of $\sim 0.75$ m from $100 - 200$ m. Although there is a data gap in Fig. 13, we can see that color of the dots changes from red to yellow, which is consistent with the decreasing trend of the total freeboard within $300$ km from our results. For W2-L, the mean total freeboard from our result is around $0.5$ m, agreeing with a mix of green and yellow dots $(0.4 - 0.9$ m$)$ in the Southern Track-W in Fig. 13. Note that the OIB ATM data in (Wang et al., 2020) was acquired on 14 and 22 November 2017, while the
SAR images in our study were acquired on 30 and 25 October 2017 for W2-U and W2-L, respectively. The sea ice drifts and potential melting could induce slight differences between our results and Wang et al. (2020).

     The W3-U and W3-L can be compared with Northern and Southern Track-E, respectively. From Fig. 13, a mix of green and yellow dots in the northern Track-E represent the total freeboard $0.5 - 1.2$ m, which agrees well with our result for W3-U, see the first row in Fig. 10. At around $70°$S, the dots transit to a mix of cyan and blue colors, representing the total freeboard of
$0.2 - 0.7$ m, which is consistent with the W3-L in Fig. 10. The slight difference can be attributed to the temporal difference of SAR images used in our study. Specifically, the image for W3-L was acquired on 26 October 2017, while the Track-E image was acquired on 22 November 2017.

## 6 Conclusions

In this study, we proposed a novel two-step approach integrating machine learning and polarimetric-interferometry techniques to retrieve total freeboard from dual-pol single-pass InSAR images, taking into account the variations in penetration bias over different ice classes. Initially, a random forest classifier was employed to categorize sea ice (i.e, SPI and LPI) based on microwaves' penetration. Subsequently, the standard InSAR processing technique was applied to retrieve the total freeboard over SPI regions, where the penetration depth is negligible. For LPI regions, an inversion algorithm for the TLPV model was developed. This algorithm can effectively compensate for the radar penetration bias into snow and ice, achieving an accurate sea ice DEM (i.e., total freeboard). The uncertainty level is satisfactory for LPI with RMSE of $0.26\,\mathrm{m}$. However, this accuracy is insufficient for thinner ice whose height above sea level is only tens of centimetres or even less. Given that a substantial portion of Antarctic sea ice consists of first-year ice with a thickness of approximately one meter (Scott, 2023), achieving accurate DEM retrieval over thinner ice remains a challenge. In the future, a potential single-pass InSAR configuration using a higher frequency, such as Ku-band, along with a longer cross-track baseline, would result in a smaller height of ambiguity (HoA) of less than $5\,\mathrm{m}$ (López-Dekker et al., 2011). This setup can enhance InSAR sensitivity and improve the accuracy of total freeboard measurements.

The proposed approach was applied to a broad area in Antarctica. Overall, sea ice in the northwestern Weddell Sea exhibits higher average total freeboard ($> 0.5\,\mathrm{m}$) than the southeastern region and the Ross Sea, where the average total freeboard is lower ($< 0.5\,\mathrm{m}$). In the northwestern Weddell Sea, sea ice experiences substantial deformation near the eastern AP, followed by a pronounced decline in both total freeboard and roughness within a range of $0 - 200\,\mathrm{km}$. Subsequently, there is a gradual increase in these parameters as one moves southward. In the southeastern Weddell Sea, the total freeboard and roughness generally decrease towards the south as they approach Coats Land. In the Western Ross Sea, thicker and rougher ice was observed near Terra Nova Bay, while thinner ice was found in the southeastern area near McMurdo Sound. In the Eastern Ross Sea, the stable total freeboard aligns with the prevalent presence of FYI, but roughness decreases towards the southeastern. These findings emphasize that topographic mapping can enhance ice category delineation, providing an in-depth understanding of sea ice characteristics.

Furthermore, the statistical analyses of the total freeboard confirmed its non-Gaussian distribution. The results further suggested that the exp-normal distribution outperforms the log-normal distribution in fitting the total freeboards of regions with an average total freeboard greater than $0.5\,\mathrm{m}$, particularly for older and thicker sea ice, whereas both distributions perform comparably for regions with an average total freeboard lower than $0.5\,\mathrm{m}$.

The spatial distribution of penetration depth (total freeboard minus radar freeboard) can be an interesting topic for future research. In snow-covered sea ice, penetration is significantly influenced by local snow conditions. Hence, conducting a coordinated campaign encompassing TanDEM-X acquisitions, lidar measurements, and in-situ snow assessments holds great promise for analyzing the relation between radar freeboard and total freeboard across different snow conditions. Future studies also involve linking the derived sea ice topographic characteristics associated with oceanographic factors (ocean current and

bathymetry) and climatology parameters (wind and temperature). We aim to further advance our comprehension of sea ice dynamics and evolution in Antarctica.

## Appendix A: Further details of data processing and figures

### A1  InSAR height uncertainty across various height of ambiguity values

The HoA ($h_a$) is the height of ambiguity determined by the specific InSAR configuration such as the radar wavelength, orbit height, incidence angle, and baseline. A larger HoA will elevate the uncertainty in the InSAR-derived height. This uncertainty ($\sigma_h$) can be estimated by (Madsen, 1998)

$$\sigma_h = \frac{h_a}{2\pi} \sigma_{\Delta_\phi}$$

where $\sigma_{\Delta_\phi}$ is the phase noise, which can be expressed as a function of the interferometric coherence ($\gamma_{\mathrm{InSAR}}$) and the inde-
pendent number of looks ($N_L$) (Rosen et al., 2000)

$$\sigma_{\Delta_\phi}^2 = \frac{1}{2N_L} \frac{1 - \gamma_{\mathrm{InSAR}}^2}{\gamma_{\mathrm{InSAR}}^2}$$

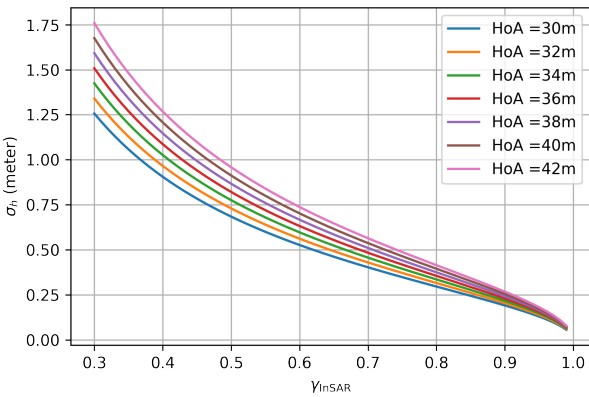

**Figure A1.** Simulation of $\sigma_h$ to the variations in $h_a$ and $\gamma_{\mathrm{InSAR}}$ with $N_L$ set to be 73.

The simulated $\sigma_h$ to the variations in $h_a$ and $\gamma_{\mathrm{InSAR}}$ is illustrated in Fig. A1. At the $\gamma_{\mathrm{InSAR}} = 0.75$, $\sigma_h$ increases from $0.35\,\mathrm{m}$ to $0.48\,\mathrm{m}$ corresponding to $h_a$ ranging from $30\,\mathrm{m}$ to $42\,\mathrm{m}$. Across the studied region, both the mean and median values of $\gamma_{\mathrm{InSAR}}$ are around $0.75$. Consequently, in the case of R5, the larger $h_a$ induces a relatively larger average uncertainty in the
derived InSAR height ($h_{\mathrm{InSAR}}$) compared to the smaller $h_a$ InSAR configuration in our dataset.

## A2 SAR thermal noise removal

The SAR-measured backscattering intensity ($\sigma_{\mathrm{measure}}$) containing additive thermal noise can be denoted as

$$\sigma_{\mathrm{measure}} = < (S_{\mathrm{denoised}} + N) \times (S_{\mathrm{denoised}} + N)^* > \tag{A1}$$

where $S_{\mathrm{denoised}}$ is the noise-subtracted backscattering amplitude, and $N$ is the additive thermal noise. Considering $S_{\mathrm{denoised}}$ and $N$ to be uncorrelated, the noise-subtracted backscattering intensity can be obtained from the following simple equation (Nghiem et al., 1995)

$$\sigma_{\mathrm{denoised}} = \sigma_{\mathrm{measure}} - NESZ \tag{A2}$$

where $NESZ$ is the noise floor (i.e., the noise equivalent sigma zero (NESZ)), and all terms are in the linear scale.

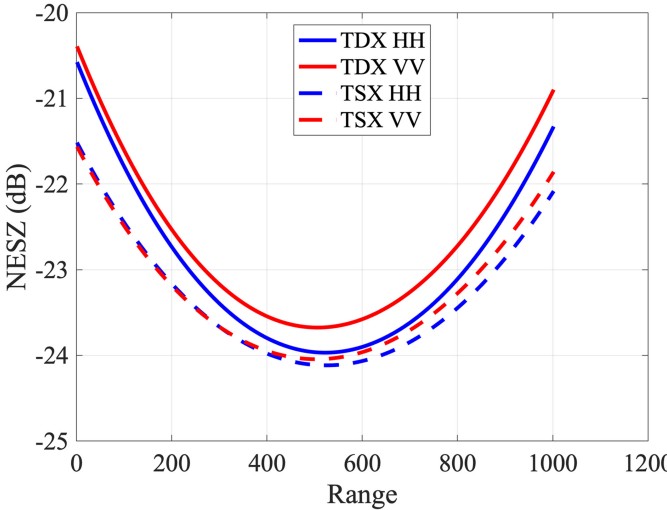

**Figure A2.** NESZ patterns for one TanDEM-X acquisition (Scene No.1 in Fig. 2) as an example.

The TanDEM-X product contains a set of polynomial coefficients that describe the NESZ pattern for each polarization along the range direction (Eineder et al., 2008) for both the TanDEM-X (TDX) and TerraSAR-X (TSX) images. An example of the calculated $NESZ$ is shown in Fig.A2 in dB scale. By converting to the linear scale, the $\sigma_{\mathrm{denoised}}$ can be calculated by subtracting $NESZ$ from the $\sigma_{\mathrm{measure}}$. We calculate the NESZ pattern for each SAR acquisition and employ Eq. A2 to generate denoised backscattering intensities for the different polarizations (i.e., HH, VV, Pauli-1 (HH+VV), and Pauli-2 (HH-VV)) from the TSX image. Note that for Pauli-1 and Pauli-2, we use the average $NESZ$ between HH and VV channels.

## A3 Ranking of SAR features

The Gini importance computed from the Random Forest (RF) classifier is given in Fig A3.

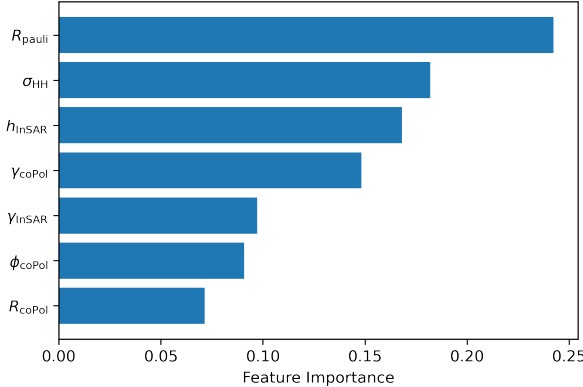

**Figure A3.** Gini importance computed from the Random Forest (RF) classifier.

The Gini importance computed from the Random Forest (RF) regressor for estimating $m$ and $h_v$ is given in Fig A4.

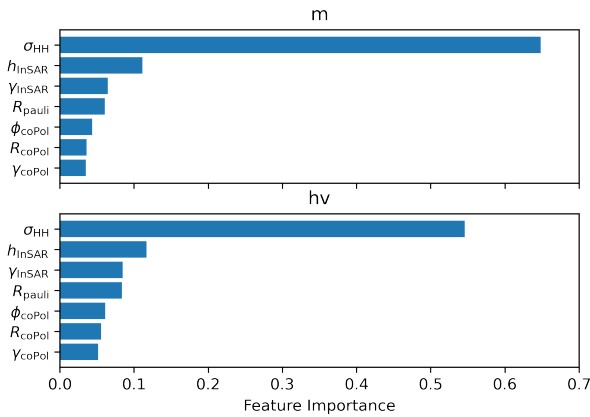

**Figure A4.** Gini importance computed from the Random Forest (RF) regressor for estimating $m$ and $h_v$.

## A4 Overview of the InSAR-derived total freeboard

The derived total freeboard over the Weddell and Ross Seas are given in Fig. A5 and A6.

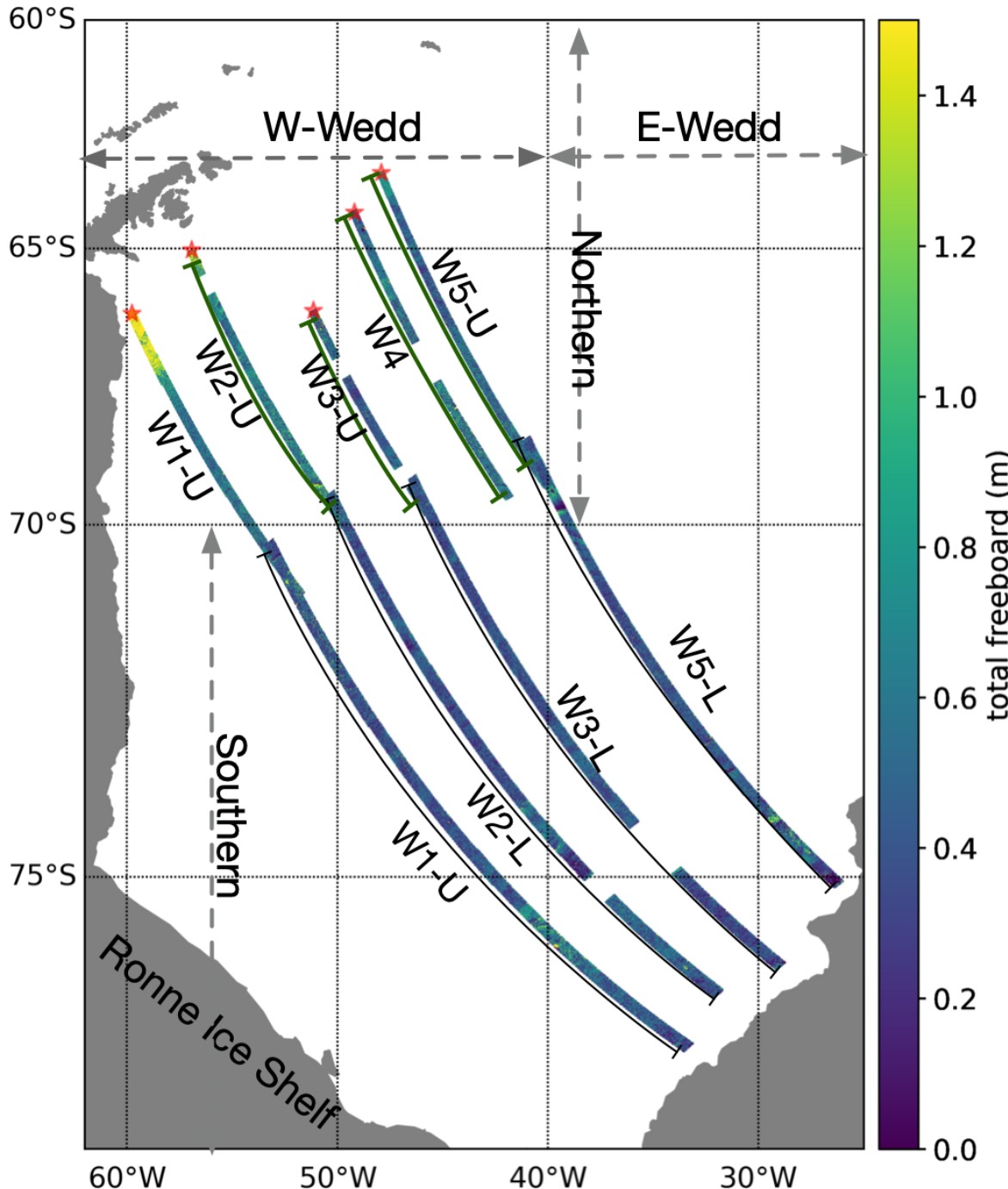

**Figure A5.** Sea ice DEM ($h_{\mathrm{mod\_SAR}}$) over the Weddell Sea retrieved from SAR images. The northernmost locations on each segment are marked with star symbols and serve as reference points for calculating the relative distance. $h_{\mathrm{mod\_SAR}}$ was downsampled to $500\,\mathrm{m}$ pixel size.

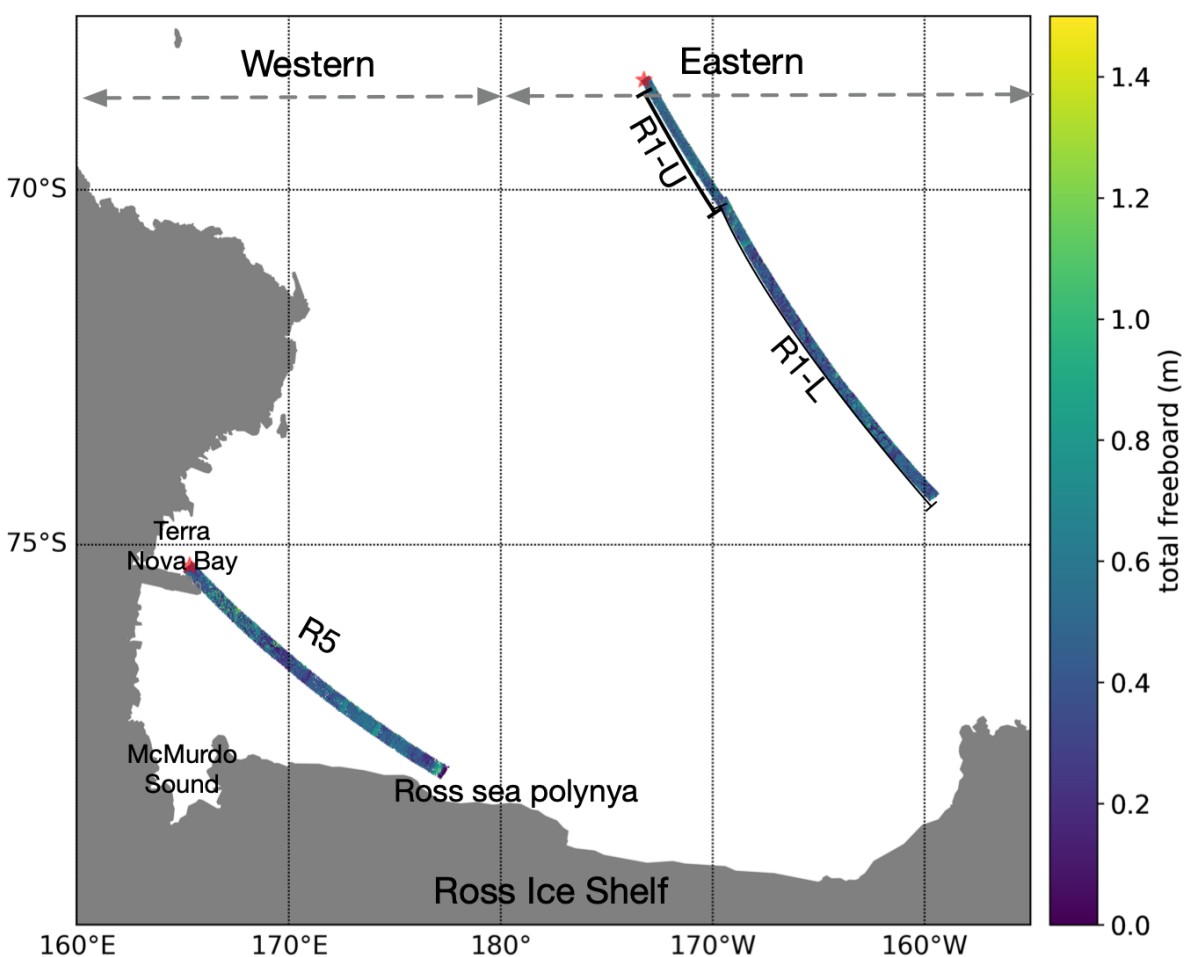

**Figure A6.** Sea ice DEM ($h_{\mathrm{mod\_SAR}}$) over the Ross Sea retrieved from SAR images. The northernmost locations on each segment are marked with star symbols and serve as reference points for calculating the relative distance. $h_{\mathrm{mod\_SAR}}$ was downsampled to $500\,\mathrm{m}$ pixel size.

*Data availability.* TanDEM-X imagery can be acquired from the German Aerospace Center (DLR) by submitting a scientific proposal, accessible at https://eoweb.dlr.de. Additionally, DMS data can be obtained from the National Snow and Ice Data Center at https://nsidc.org/data/icebridge, while Ice Charts data are available at https://nsidc.org/data/G10033/versions/1, also from the National Snow and Ice Data Center.

*Author contributions.* LH carried out SAR image processing, designed the methodology, analyzed the results, and drafted the manuscript. IH provided valuable input on method development and result analysis, contributing to the enhancement of the manuscript.

*Competing interests.* The authors have no conflicts of interest to disclose.

*Acknowledgements.* The authors express gratitude to the Swiss National Science Foundation Postdoc Mobility Fellowship (P500PN_217817) for partially providing financial support for this research. We would like to thank all individuals involved in the OTASC campaign, jointly conducted by DLR and NASA. Special thanks to Dr. Thomas Busche for his invaluable assistance in accessing the TanDEM-X data. We also express our appreciation to Dr. Vishnu Nandan and the two anonymous reviewers for their comments, which significantly improved the paper. Finally, Lanqing Huang would like to express her gratitude to her boyfriend, Dr. Xun Jack Li, for his accompany, support, and abundant encouragement during her PhD and postdoc years in Zurich.

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
