# Peer review of "A Study of Sea Ice Topography in the Weddell and Ross Seas Using Dual Polarimetric TanDEM-X Imagery"

_EGUsphere, 2023_

## Author Comment (AC1)

**Response to the comments of Reviewer 1**

First of all, we would like to thank the anonymous reviewer for the careful review and valuable suggestions. We carefully revised the manuscript following the suggestions. Hereby we give a point-by-point reply to address the comments. In this document, the words in *italics are the reviewers' comments*, the words in blue are the modifications we have made in the revision, and others are our responses.

**Q1:** *The work aims to study the sea ice freeboard in areas around the Antarctic peninsula. Improved sea ice surface topography is a useful product and can be implemented in other studies, such as ice drift product development and climate studies. The manuscript is reasonably well written and mostly easy to follow. Some of the terminology is at times somewhat confusing. There are a great many figures in the manuscript, could some perhaps be moved to supplementary information.*

**A1:** We thank the reviewer for the positive comment about our research. We have carefully revised the manuscript based on the following comments.

**Q2:** *In the abstract the terms sea ice DEM, i.e. snow freeboard is introduced. How does this relate to the sea ice topography? Why is the sea ice DEM not = sea ice and snow freeboard? How is the air-ocean-ice system related to the sea ice topography? The statement as it stands right now is a bit challenging to interpret.*

**A2:** We apologize for any confusion caused by the term "snow freeboard." In this paper, when referring to sea ice DEMs, we actually mean the total freeboard, which includes both the ice freeboard and the thickness of the snow layer. Therefore, whenever we mention "sea ice elevation," it actually means the total freeboard. In the revised version, we have explicitly defined sea ice DEMs as total freeboard (snow+ice). We have made corrections by replacing "snow freeboard" with "total freeboard" and updated the term "sea ice elevation" to "total freeboard" throughout the manuscript, including both the text and figures, to ensure clarity and consistency.

In the revision, we have restated the sentence in the abstract: The total freeboard (snow+ice) is crucial for reflecting sea ice dynamics and interpreting the geophysical environments of polar oceans.

In the Introduction section, we have included detailed explanations about how the air-ocean-ice system is related to sea ice topography:

The sea ice topography plays a crucial role in reflecting sea ice dynamics and interactions within the air-ocean-ice system. It showcases the spatial distribution of distinct surface features such as snow dunes (Trujillo et al., 2016; Iacozza and Barber, 1999) and deformed ice (Haas et al., 1999; Petty et al., 2016), which are impacted by the forces from winds and currents. Moreover, the atmospheric drag coefficient over sea ice, which is topography-dependent, is an important parameter for understanding interactions at the ice-atmosphere boundary (Garbrecht et al., 2002; Castellani et al., 2014).

Castellani, G., Lüpkes, C., Hendricks, S., and Gerdes, R.: Variability of Arctic sea-ice topography and its impact on the atmospheric surface drag, J. Geophys. Res.: Oceans, 119, 6743–6762, https://doi.org/10.1002/2013JC009712, 2014.

Garbrecht, T., Lüpkes, C., Hartmann, J., and Wolff, M.: Atmospheric drag coefficients over sea ice–validation of a parameterisation concept, Tellus A: Dynamic Meteorology and Oceanography, 54, 205–219, https://doi.org/10.3402/tellusa.v54i2.12129, 2002.

Haas, C., Quanhua, L., and Thomas, M.: Retrieval of Antarctic sea-ice pressure ridge frequencies from ERS SAR imagery by means of in situ laser profiling and usage of a neural network, International Journal of Remote Sensing, 20, 3111–3123, 1999.

Iacozza, J. and Barber, D. G.: An examination of the distribution of snow on sea-ice, Atmosphere-Ocean, 37, 21–51, https://doi.org/10.1080/07055900.199 1999.

Petty, A. A., Tsamados, M. C., Kurtz, N. T., Farrell, S. L., Newman, T., Harbeck, J. P., Feltham, D. L., and Richter-Menge, J. A.: Characterizing Arctic sea ice topography using high-resolution IceBridge data, The Cryosphere, 10, 1161–1179, https://doi.org/10.5194/tc-10- 1161-2016, 2016.

Trujillo, E., Leonard, K., Maksym, T., and Lehning, M.: Changes in snow distribution and surface topography following a snowstorm on Antarctic sea ice, Journal of Geophysical Research: Earth Surface, 121, 2172–2191, https://doi.org/10.1002/2016JF003893, 2016.

**Q3:** *R21. If we assume that the DEM is snow freeboard, should it then be assumed that no penetration if the snow is possible?*

**A3:** The term "sea ice DEMs" refers to the total freeboard, which is ice freeboard plus snow depth. We have modified it to 'total freeboard' throughout the manuscript.

**Q4:** *R22-23. Please elaborate how this product is essential for assessing the impact of climate change on sea ice.*

**A4:** We have modified in the introduction: Moreover, the DEM (i.e., total freeboard) can be converted to thickness with the knowledge of snow depth and the assumed values of snow, ice, and seawater densities (Kwok and Kacimi, 2018). Estimating sea ice thickness over time offers valuable insights into the overall stability of sea ice in the changing climate.

**Q5:** *R58-59. How can the DEM help separate the different ice types?*

**A5:** In R58-59, we do not intend to imply that the DEM can help separate different ice types. Rather, we mean that sea ice types can serve as prior knowledge for generating DEM from InSAR. The InSAR-derived height represents the elevation of the InSAR phase center, which is not necessarily at the snow-air surface due to radar penetration. The depth of penetration varies depending on the ice type, with sea ice having lower salinity and being covered by dry snow allowing for deeper penetration compared to ice with higher salinity and wet snow cover. To ensure accurate sea ice DEM retrieval, we proposed first classifying the sea ice into small-penetration and larger-penetration conditions, followed by retrieving the sea ice DEM using standard InSAR processing and the proposed TLPV model for each ice type, respectively.

In the revision, we have rewritten the sentence for clarity:

The initial step involves the development of a random forest classifier using specific SAR features to categorize sea ice into two groups: small-penetration condition ice (SPI) and large-penetration condition ice (LPI), based on the penetration depth of microwaves. Subsequently, a sea ice DEM is created for each ice type. In the case of SPI, standard InSAR processing is applied to determine the elevation. For LPI, a novel inversion algorithm is proposed to estimate the parameters of the developed TLPV model (Huang et at., 2021). This allows for correcting penetration bias in the InSAR signal over LPI, resulting in an accurate retrieval of total freeboard. We validate the proposed method against the photogrammetric DEM from the IceBridge aircraft. A root-mean-square error (RMSE) of $0.26\,\mathrm{m}$ between the derived DEM and reference data indicates an improved accuracy in total freeboard retrieval.

**Q6:** *R67. What is meant with Ice Chart here? An operational ice charts such as those provided by the ice services. What is meant is explained on R108. This one of the terminology words introduced it the introduction that gets explained later in the manuscript. This terminology should either be removed from the introduction or needs to be explained here.*

**A6:** We have removed the terminology and revised the sentences as:

...in sea ice classes obtained from an operational product from the U.S. National Ice Center.

**Q7:** *R91. Perhaps state how the denoising is done then why it's useful/essential to do so here.*

**A7:** The SAR-measured backscattering intensity ($\sigma_{\mathrm{measure}}$) containing additive thermal noise can be denoted as

$$\sigma_{\mathrm{measure}} = <(S_{\mathrm{denoised}} + N) \times (S_{\mathrm{denoised}} + N)^*> \tag{1}$$

where $S_{\mathrm{denoised}}$ is the noise-subtracted backscattering amplitude, and $N$ is the additive thermal noise. Considering $S_{\mathrm{denoised}}$ and $N$ to be uncorrelated, the noise-subtracted backscattering intensity can be obtained from the following simple equation (Nghiem et al., 1995)

$$\sigma_{\mathrm{denoised}} = \sigma_{\mathrm{measure}} - NESZ \tag{2}$$

where $NESZ$ is the noise floor (i.e., the noise equivalent sigma zero (NESZ)), and all terms are in the linear domain.

The TanDEM-X product contains a set of polynomial coefficients that describe the NESZ pattern for each polarization along the range direction (Eineder et al., 2008) for both the TanDEM-X (TDX) and TerraSAR-X (TSX) images. An example of the calculated $NESZ$ is shown in Fig.1 in dB scale. By converting to the linear scale, the $\sigma_{\mathrm{denoised}}$ can be calculated by subtracting $NESZ$ from the $\sigma_{\mathrm{measure}}$. We calculated the NESZ pattern for each SAR acquisition and employed Eq. 2 to generate denoised backscattering intensities for the different polarizations (i.e., HH, VV, Pauli-1 (HH+VV), and Pauli-2 (HH-VV)) from the TSX image. Note that for Pauli-1 and Pauli-2, we use the average $NESZ$ between HH and VV channels.

[Figure]

Fig. 1: NESZ patterns for one TanDEM-X acquisition (Scene No.1, see Fig.2 in the paper) as an example.

In the revision, we have added the above description in the supplementary information.

Thermal noise can contaminate the SAR backscattering intensity. Removing the thermal noise allows for a better representation of sea ice features from SAR image, which is crucial for ice classification. The denoised SAR backscattering intensity was input as the features for the random forest classifier in Section 3.1.

In the revision, we have added the usefulness of the denoising processing:

The backscattering intensity $\sigma_{\mathrm{measure}}$ of the images includes additive thermal noise, which can be described by the noise equivalent sigma zero (NESZ) and assumed to be uncorrelated with the signal (Nghiem et al., 1995). Removing the thermal noise allows for a better representation of sea ice features, which is crucial for ice classification. We denoised backscattering intensities for the different polarizations (i.e., HH, VV, Pauli-1 (HH+VV), and Pauli-2 (HH-VV)) by subtracting the noise equivalent sigma zero (NESZ) from the $\sigma_{\mathrm{measure}}$, see details in the supplementary information.

Nghiem, S., Kwok, R., Yueh, S., and Drinkwater, M.: Polarimetric signatures of sea ice: 2. Experimental observations, J. Geophys. Res.:490 Oceans, 100, 13 681–13 698, https://doi.org/10.1080/08843759508947700, 1995.

Eineder, M., Fritz, T., Mittermayer, J., Roth, A., Boerner, E., & Breit, H. (2008). TerraSAR-X ground segment, basic product specification docu-ment (Tech. Rep.). Cluster Applied Remote Sensing (CAF).

**Q8:** *Table 1. One of the datasets (R5) has a higher HoA. Does this affect the results presented here?*

**A8:** The HoA ($h_a$) is the height of ambiguity determined by the specific InSAR configuration such as the radar wavelength, orbit height, incidence angle, and baseline. $h_a$ is used in converting InSAR phase ($\phi_\gamma$) into height ($h_{\mathrm{InSAR}}$) through $h_{\mathrm{InSAR}} = \frac{\phi_\gamma}{2\pi} h_a$.

However, a larger HoA (ranging from 40 to 42 meters) will elevate the uncertainty in InSAR-derived height. This uncertainty ($\sigma_h$) can be estimated by (Madsen and Zebker, 1998):

$$\sigma_h = \frac{h_a}{2\pi} \sigma_{\Delta_\phi}$$

where $\sigma_{\Delta_\phi}$ is the phase noise, which can be expressed as a function of the interferometric coherence ($\gamma_{\mathrm{InSAR}}$) and the independent number of looks ($N_L$) (Rosen et al., 2000):

$$\sigma^2_{\Delta_\phi} = \frac{1}{2N_L} \frac{1 - \gamma^2_{\mathrm{InSAR}}}{\gamma^2_{\mathrm{InSAR}}}$$

The simulated $\sigma_h$ to the variations in $h_a$ and $\gamma_{\mathrm{InSAR}}$ is illustrated in Fig. 2. At the $\gamma_{\mathrm{InSAR}} = 0.75$, $\sigma_h$ increases from $0.35\,\mathrm{m}$ to $0.48\,\mathrm{m}$ corresponding to $h_a$ ranging from $30\,\mathrm{m}$ to $42\,\mathrm{m}$. Across the studied region, both the mean and median values of $\gamma_{\mathrm{InSAR}}$ are around $0.75$. Consequently, in the case of R5, the larger $h_a$ induces a relatively larger average uncertainty (around $0.13\,\mathrm{m}$) in the derived InSAR height ($h_{\mathrm{InSAR}}$) compared to the smallest $h_a$ InSAR configuration in our dataset.

[Figure]

Fig. 2: Simulation of $\sigma_h$ to variations in $h_a$ and $\gamma_{\text{InSAR}}$. In our case $N_L = 73$.

In the revision, we have added the above analyses in the supplementary information.

Madsen, S. N. and Zebker, H. A.: Imaging Radar Interferometry, in: Principles and Applications of Imaging Radar, Manual of Remote Sensing, 3rd Edn., John Wiley & Sons, New York, 2, 359380, 1998.

Rosen, P. A., Hensley, S., Joughin, I. R., Li, F. K., Madsen, S. N., Rodriguez, E., and Goldstein, R. M.: Synthetic aperture radar interferometry, P. IEEE, 88, 333–382, https://doi.org/10.1109/5.838084, 2000.

**Q9:** *R111. The spatial resolution of the Ice Charts is 10 x 10 km. How wide are the SAR images used? Will more than a few pixels be comparable between the Ice Charts and the SAR images?*

**A9:** In Figures 13-15, we present the sea ice topography variation (total freeboard and roughness from SAR) at $100\,\text{km}$ intervals. Specifically, we selected the SAR pixels at each $100\,\text{km}$ distance and computed statistics for their total freeboard and roughness. The resolution of each SAR pixel is $500 \times 500\,\text{m}$. Given that the Ice Charts have a spatial resolution of $\sim 10 \times 10\,\text{km}$, we directly plotted the values from them. Consequently, in the first columns, you can observe approximately 10 points from the Ice Charts at every $100\,\text{km}$ interval.

There are more than a few SAR pixels compared to a single Ice Chart pixel. We do not down-sample the SAR results to match the Ice Charts resolution, since our objective is to generate a high-resolution (sub-kilometer) sea ice DEM and explore its role in understanding the spatial variation of sea ice topography (as detailed in Section 4.2). It is also essential to clarify that we utilized Ice Charts data merely as an external information source in interpreting the topographic variation. However, we did not employ the Ice Charts for validating the proposed method. For validation purposes, we conducted pixel-by-pixel comparisons using co-registered DMS data, which were down-sampled to the same resolution level as the SAR result.

**Q10:** *Figure 3, 4 and 7. The schematic in Figure 3 in itself is good but it's challenging to understand if perhaps Figure 4 is step 1, and if so why this isn't stated in Figure 4. Please indicate how these 3 flow charts are interconnected. It appears as if Step 1 is in part explained in Figure 4 but it's unclear as more information than the TanDEM-X SAR images are used as input data? And the classification map at the end of Figure 4 appears to perhaps be the first box in Step 1. Figure 7 appears to be an explanation of the top right box in Step 2 in Figure 3. Please clarify these flow charts.*

**A10:** Figure 3 provides a detailed explanation of the training process for the sea ice classifier in Step 1. Figure 4 elaborates on the PolInSAR height retrieval module in Step 2, including inverting the proposed TLPV model to generate $h_{\text{mod}}$. During the training and validation phases of both the sea ice classifier and the PolInSAR height retrieval module, DMS DEMs were input as reference data. With the trained classifier and the module, the two-step method was applied to TanDEM-X SAR images together with AMSR level-3 snow depth measurements as input. We have merged the three figures into one for clarification:

[Figure]

Fig. 3: (a) The proposed two-step approach for sea ice DEM retrieval. (b) The details (training and validation) of the sea ice classifier and (c) the PolInSAR height retrieval module.

The above flowchart has been added to the revision.

**Q11:** *R180-184. Are some parameters more important for one of specific ice types? Or is the importance level presented in Fig 5 universal?*

**A11:** The Gini importance is a metric used in a random forest classifier to measure the relative importance of each feature in making classification decisions. It is calculated based on the decrease in Gini impurity that each feature contributes to the overall model.

The computed Gini importance is not inherently specific to a particular class or ice type. Instead, it represents the relative importance of features in making overall classification decisions within the context of the entire dataset. Therefore, the importance level determined by Gini importance is not specific to individual ice types but reflects the significance of features for the classifier's overall predictive performance across all classes (i.e., SPI and LPI).

**Q12:** *Figure 9. Some of the leads appear to have a light blue color, not the same as for the YI. Why is that? Which ice type do they represent? They appear to in 1, 2 and 3 have the highest E. What is the unit E? Does a low SNR perhaps get mistaken as a thick sea ice? Perhaps could a noise analysis remove erroneous values?*

**A12:** We apologize for the confusion. The light blue is the color of the base map used for plotting. During InSAR processing, we excluded pixels with an InSAR coherence less than 0.3, setting their values to NaN. When plotted, NaN values are rendered as void areas, showing the color of the base map. Since pixels with low InSAR coherence often correspond to water areas, it appears that all leads and water areas are colored in light blue. The SAR backscattering intensities in the water area mostly range below -19dB. Note that the system noise level of TanDEM-X is around -19 to -26dB. These pixels exhibiting low SNR induce significant uncertainty in InSAR processing. As a result, we have excluded these regions (i.e., open water leads) from

further analyses. In the revision, we have changed the color of the base map to transparent (white) and added a statement in the caption: The void pixels in the second and third columns represent water areas excluded from processing due to $\gamma_{\text{InSAR}} < 0.3$.

In the third column of Fig. 9, the label is a vertically printed "m" which stands for meter, not "E." This column represents the derived sea ice DEM (total freeboard, in the unit of meter) using the proposed two-step method. Scenes 1, 2, and 3 near the Antarctic Peninsula exhibit higher total freeboard. The analysis of the higher freeboard and the spatial variation along the transect are provided in Section 4.2. In the revision, we have changed the colorbar's label to meter for clarity.

**Q13:** *Figure 11. In the top, upper middle and bottom figures, it appears as if the SAR estimates are underestimating the high and low peaks. Is this a resolution issue? Or is there some other explanation behind this?*

**A13:** In the postprocessing, we geocoded the DMS DEM into the SAR coordinate and down-sampled the DMS DEM into the same resolution as the SAR pixel size ($10 \times 10\,\text{m}$ in the ground range and azimuth). Therefore, resolution is not likely the cause of underestimation.

One factor contributing to the underestimation of total freeboard could be the assumption of a constant average snow depth over one SAR scene (spatial coverage of $50 \times 19\,\text{km}$). The constant average snow depth is an input parameter $z_1$ in the TLPV model. In our methodology, we assume this snow depth remains uniform across one SAR image due to the limited spatial resolution of available snow measurements (AMSR Level-3 data with a resolution of $12.5\,\text{km}$). However, this uniformity may lead to an underestimation of snow depth in high-peak areas such as ridges, consequently resulting in an underestimation of the total freeboard. Our prior study (Huang et al., 2021) demonstrated a mean difference of $0.31\,\text{m}$ in the derived total freeboard due to snow depth variations from $0.05$ to $0.75\,\text{m}$ over Scene No.1, highlighting the impact on peak estimation. In the future, it would be interesting to adapt the proposed total freeboard retrieval method over the test sites co-locating with available high-resolution snow depth measurements.

Another factor that could potentially lead to the underestimation of high and low-peak areas is the residual shift between the SAR and DMS images. Although we carefully co-registered the four SAR scenes with the DMS data, the co-registration can not be perfect. In the process, we divided the entire overlapped transect into small segments (each corresponding to $100 \times 1000\,\text{m}$). We assumed the same drift location over one segment and no rotation; thus, only one shift vector was used for co-registration over each segment. This could result in small residual shifts when the ice floes or features do not drift at the same velocity or involve rotations. For high and low-pixel ice features with narrow sizes covering one or a few pixels, even 1-2 pixels of residual shifts could lead to some loss or mismatched information of the ice structure from the SAR images. The mismatched SAR images input into the proposed model could result in underestimation when compared with the total freeboard from the DMS DEM.

We have added the above text in a new Section Discussion in the revision.

Huang, L., Fischer, G., and Hajnsek, I.: Antarctic snow-covered sea ice topography derivation from TanDEM-X using polarimetric SAR455 interferometry, The Cryosphere, 15, 5323–5344, https://doi.org/10.5194/tc-15-5323-2021, 2021.

**Q14:** *Figure 12. This figure could perhaps be moved to supplementary information as it doesn't add much to the understanding of the results. It's very challenging to see the elevations, if kept perhaps make the SAR images a lot larger?*

**A14:** In the revision, we have enlarged the size of Fig. 12. However, due to the ratio of the width and length of the transects, the enlarged plot occupies a full page, so we decided to move it to supplementary information.

**Q15:** *Figure 13, 14, 15. Consider coloring the y-axis and the color used in the plot the same color for easier interpretation of the information contained within the figures. Add a legend to the two rightmost columns, to explain what the blue and the orange represents.*

**A15:** In the revision, we have improved the first column in Fig. 13-15 by incorporating visualizations of ice concentration for each ice type (MYI, FYI, and TI) from the Ice Charts, instead of only showing the "average ice type" as in the previous manuscript version. We have utilized a consistent colormap (same as Fig.16) to represent each ice type. We have included a legend for clarity in the second and third columns as suggested.

Note that these updated plots are primarily for improved visual comparison between the Ice Charts and the SAR results, with no alterations to the main observations or conclusions within the Section Results.

The updated Fig13-15 are given below:

[Figure]

Fig. 4: Sea ice characteristics along the southwards direction along W1 and W2 segments. The blue line in the first column displays the OI percentages derived from SAR images, and the blue dot indicates the ice types obtained from the Ice Charts. The second and third columns plot the elevation ($h_{\mathrm{mod\_SAR}}$) and roughness ($\sigma_R$), respectively. Distance is measured from the northernmost SAR image reference point towards the south. The orange line denotes the average values of $h_{\mathrm{mod_SAR}}$ and $\sigma_R$. The box's upper and lower boundaries represent the first (Q1) and third (Q3) quartiles, while the upper (lower) whisker extends to the last (first) sample outside of Q3 $\pm 1.5 \times$(Q3-Q1).

[Figure]

Fig. 5: Sea ice characteristics along the southwards direction along W3, W4, and R5 segments. The blue line in the first column displays the OI percentages derived from SAR images, and the blue dot indicates the ice types obtained from the Ice Charts. The second and third columns plot the elevation ($h_{\mathrm{mod\_SAR}}$) and roughness ($\sigma_R$), respectively.

[Figure]

Fig. 6: Sea ice characteristics along the southwards direction along W5 and R1 segments. The blue line in the first column displays the OI percentages derived from SAR images, and the blue dot indicates the ice types obtained from the Ice Charts. The second and third columns plot the elevation ($h_{\mathrm{mod\_SAR}}$) and roughness ($\sigma_R$), respectively.

**Q16:** *R2 "… a digital …" or "digital elevation models"*
**A16:** Done

**Q17:** *R2-3 should it be drifting sea ice instead of drift sea ice?*
**A17:** Done

**Q18:** *R60-61. "sea ice elevation" has already been defined earlier in the manuscript.*

**A18:** The repeated statement has been removed.

**Q19:** *R76. With sequence is it meant orbit?*

**A19:** Yes, it means a series of acquisitions within some seconds along the same orbit. We revised the sentence: The footprints consist of 12 segments, each corresponding to a sequence of SAR images within the same orbit, all acquired at nearly the same time, with only seconds varying between them.

**Q20:** *R143. Wakabayashi et al 2004 used L-band SAR, how will this compare to the X-band SAR used here? Can we derive sea ice thickness using X-band SAR?*

**A20:** The work (Wakabayashi et al 2004) suggests that the co-polarization ratio from L-band SAR image has been demonstrated related to ice-thickness. However, as far as we know, no published results demonstrate a relation between the co-polarization ratio from X-band SAR and ice-thickness. Nevertheless, the co-polarization coherence from TerraSAR-X has been demonstrated to be correlated to ice thickness over multi-year sea ice (Kim et al., 2011). This reference was cited in Section 2.6.3.

Kim, J.-W., Kim, D.-j., and Hwang, B. J.: Characterization of Arctic sea ice thickness using high-resolution spaceborne polarimetric SAR data, IEEE Trans. Geosci. Remote Sens., 50, 13–22, https://doi.org/10.1109/TGRS.2011.2160070, 2011.

In the revision, we clarified that the reference is based on L-band SAR data:

$R_{\mathrm{coPol}}$ extracted from L-band SAR images, which is related to the dielectric constant, has been considered as an indicator of ice thickness (Wakabayashi et al., 2004). Further investigation is required to determine if $R_{\mathrm{coPol}}$ from the X-band can also serve as a proxy for ice thickness.

**Q21:** *R198. The reference can be shortened to (Meier, Markus and Comiso, 2018)*
**A21:** Done

**Q22:** *R248. "… in the Ice Charts"*
**A22:** Done

**Q23:** *R283. Sea ice doesn't evolve from MYI to TI. TI can evolve to MYI through surviving at least 2 seasonal cycles.*
**A23:** We refer to the spatial transition of sea ice types from MYI to TI in the southeastward direction.
We have revised the sentence as:
This observation aligns with the transition of ice types from MYI to TI in that direction.

**Q24:** *R363-367. It this information needed here?*
**A24:** We prefer to keep these sentences as a summary of our observations regarding sea ice DEMs. This enables readers who may skip the detailed reading of Sections 4.2 and 4.3 to still grasp some key take-home messages.

Again, we sincerely thank the editor and reviewers for helping us improving the manuscript.

---

## Author Comment (AC2)

**Response to the comments of Reviewer 2**

First of all, we would like to thank the anonymous reviewer for the careful review and valuable suggestions. We carefully revised the manuscript following the suggestions. Hereby we give a point-by-point reply to address the comments. In this document, the words in *italics are the reviewers' comments*, the words in blue are the modifications we have made in the revision, and others are our responses.

**Q1:** *In this interesting paper, a new method for the retrieval of total ice freeboard (ice freeboard plus snow thickness) from single-pass interferometric SAR is developed and applied to the Weddell and Ross Seas. The SAR-derived sea ice topography is validated by independently measured sea ice freeboard profiles and analyzed in comparison to several studies, which support the results. The paper should definitely be published, but I recommend modifications which concern the use of certain terms and the need for additional information. The latter is in particular important for the description of the method.*

**A1:** We thank the reviewer for the positive comment about our research. We have carefully revised the manuscript based on the following comments.

**Q2:** *Line 3: "accurate sea ice DEMs (i.e., snow freeboard)" The term "snow freeboard" (see also line 21 in the introduction) is misleading. Better use "total freeboard" which is ice freeboard plus snow layer thickness*

**A2:** We apologize for any confusion caused by the term "snow freeboard." In this paper, when referring to sea ice DEMs, we actually mean the total freeboard, which includes both the ice freeboard and the thickness of the snow layer. Therefore, whenever we mention "sea ice elevation," it actually means the total freeboard. In the revised version, we have explicitly defined sea ice DEMs as total freeboard (snow+ice). Moreover, we have replaced the old term "sea ice elevation" with "total freeboard" throughout the manuscript, including both the text and figures, to ensure clarity and consistency.

**Q3:** *Lines 21-22: It is the mass of the ice above the water surface plus snow load (not snow freeboard) from which ice thickness can be estimated.*

**A3:** We apologize for the confusing term. The sentence has been revised as: Moreover, the DEM (i.e., total freeboard) can be converted to thickness with the knowledge of snow depth and the assumed values of snow, ice, and seawater densities (Kwok and Kacimi, 2018). Estimating sea ice thickness over time offers valuable insights into the overall stability of sea ice in the changing climate.

**Q4:** *Line 35: As far as I remember does the Dierking paper discuss problems and requirements for retrieving the sea ice surface topography of drifting ice but demonstrates it only for landfast ice.*

**A4:** Yes, Dierking's paper theoretically discussed the impacts of sea-ice drifting velocity on the retrieval of topographic heights and calculated the interferometric sensitivity. However, an example of InSAR retrieval was conducted over landfast sea ice near the coastline of Barrow. We have revised the sentence as:

Notably, the advent of single-pass interferometric SAR (InSAR) sensors, exemplified by TanDEM-X, presents an unprecedented opportunity to generate sea ice DEMs over landfast sea ice (Dierking et al., 2017; Yitayew et al., 2018). For drifting ice, the accuracy of InSAR-derived DEMs can be affected by additional phase shifts induced by ice motion. Dierking et al. (2017) calculated and theoretically discussed the sensitivity of InSAR-derived DEMs concerning ice-drifting velocity, InSAR frequency, and baseline configuration.

**Q5:** *Line 43 and lines 54-55: "Antarctic old ice" – what precisely is "old ice"? The separation between "young ice" and "old ice" based on the criterion of penetration depth (the difference between DMS and SAR elevation) is not suitable, since salinity (as the major factor influencing the μ-wave penetration) is not only linked to ice age but also to other factors (e.g. saline snow crusts at the ice surface, effects of ice flooding). This is also visible in your data, Figs. 13-15. I propose that you instead use the categories "low-penetration condition" and "large-penetration condition".*

**A5:** The "Antarctic old ice" refers to the ice that has a penetration depth (the difference between DMS and SAR elevation) larger than $0.3\,\mathrm{m}$. We agree with the reviewer that the penetration does not simply depend on the age but on the geophysical

conditions of snow and ice. In the revision, we have modified all the "older ice (OI)" and "younger ice (YI)" into the large-penetration condition ice (LPI) and small-penetration condition ice (SPI).

**Q6:** *Lines 59-61: Sentences: "A root-mean-square error (RMSE) of 0.26m between the derived DEM and reference data signifies a precise elevation mapping for both YI and OI. Throughout the paper, "sea ice elevation" is the entire vertical height (including snow depth) above the local sea surface." Actually, 0.26 m (for averages over areas of several meters side length) can locally be a rather high (but mostly acceptable) uncertainty, considering that a large fraction of Antarctic sea ice is first-year with a thickness of around one meter (https://www.climate.gov/news-features/understanding-climate/understanding-climate-antarctic-sea-ice-extent) and correspondingly much less elevation above the water surface. "Precise" means that repeated measurements are close to one another – here the term "accurate" may be more appropriate.*

**A6:** In the Section Introduction, we have revised the sentence as:

A root-mean-square error (RMSE) of $0.26\,\mathrm{m}$ between the derived DEM and reference data indicates an improved accuracy in elevation mapping.

In the Section Conclusion, we have added some texts to discuss the RMSE for both large-penetration condition ice and small-penetration condition ice a bit more:

The uncertainty level is satisfactory for LPI with RMSE of $0.26\,\mathrm{m}$. However, this accuracy is insufficient for thinner ice whose height above sea level is only tens of centimetres or even less. Given that a substantial portion of Antarctic sea ice consists of first-year ice with a thickness of approximately one meter (Scott 2023), achieving precise DEM retrieval over thinner ice remains a challenge. In the future, a potential single-pass InSAR configuration using a higher frequency, such as Ku-band, along with a longer cross-track baseline, would result in a smaller height of ambiguity (HoA) of less than $5\,\mathrm{m}$ (López-Dekker et al., 2011). This setup can enhance InSAR sensitivity and improve the accuracy of total freeboard measurements.

Scott, M.: Understanding climate: Antarctic sea ice extent., NOAA Climate Government, https://www.climate.gov/news-features/ understanding-climate/understanding-climate-antarctic-sea-ice-extent, accessed March 22, 2024., 2023.

López-Dekker, Paco, et al. "TanDEM-X first DEM acquisition: A crossing orbit experiment." IEEE Geoscience and Remote Sensing Letters 8.5 (2011): 943-947.

**Q7:** *Line 87: Here it is ground-range? Is the pixel size of 10 × 10 m used for both the classification process and for elevation retrieval? Should be stated.*

**A7:** Yes, it is ground-range and used for the following sea ice classification and DEM retrieval. We have added a statement:

The multilooking processing was conducted using a $4 \times 12$ window, resulting in a $\sim 10 \times 10\,\mathrm{m}$ pixel spacing in azimuth and ground range. This resolution of $\sim 10 \times 10\,\mathrm{m}$ was subsequently utilized for the sea ice classification and DEM retrieval detailed in Section 3.

**Q8:** *Line 97: The vertical accuracy of the DMS data (line 232) should also be mentioned here. Which reference surface was used for the height values? The local water surface or a reference ellipsoid? In the User Guide by Dotson and Arvesen I found "The IceBridge DMS L3 Photogrammetric DEMs are GeoTIFF imagery, in meters and above the WGS-84 ellipsoid." (page 5). The WGS-84 ellipsoid is usually not at the same level as the local water surface.*

**A8:** We have added the vertical accuracy:

...the OIB aircraft captured optical images (Dominguez, 2010, updated 2018) and generated DEM using photogrammetric techniques at a spatial resolution of approximately $40\,\mathrm{cm} \times 40\,\mathrm{cm}$ with a vertical accuracy of $0.2\,\mathrm{m}$ (Dotson and Arvesen., 2012, updated 2014).

Yes, the height values directly extracted from DMS DEM are above the WGS-84 ellipsoid. In the postprocessing of DMS DEM, we calibrated the values to the local sea level by selecting the water-surface reference from DMS images. For each SAR image, we labeled around ten points as water-surface references according to the DMS images. The average height of the open-water points was subtracted from the origin DMS DEMs to obtain the values relative to the average sea level.

The above processing was described in our previous work (Huang et al., 2021). In the revision, we have added some texts in Section 2.3 for clarification:

Note that DMS DEM gives height values relative to the WGS-84 ellipsoid. To obtain the total freeboard, we calibrated the DMS DEM to the local sea level through a manual process involving the selection of the water surface from DMS images

(Huang et al., 2021). The calibrated DMS DEM, henceforth referred to as DMS DEM for brevity, is utilized as the validation data throughout the paper.

Huang, L., Fischer, G., and Hajnsek, I.: Antarctic snow-covered sea ice topography derivation from TanDEM-X using polarimetric SAR455 interferometry, The Cryosphere, 15, 5323–5344, https://doi.org/10.5194/tc-15-5323-2021, 202

**Q9:** *Lines 115-117: I checked the types of ice charts at the US National Ice Center (https://usicecenter.gov/Products/AntarcHome). I recommend that you provide a link for your reference ("U.S. National Ice Center., 2020") and show the ice chart which you actually used. E.g., I did not find any hint that the ice charts are averages over 7 days, but are produced once a week (https://nsidc.org/sites/default/files/documents/user-guide/g10013-v001-userguide.pdf).*

**A9:** We agree with the reviewer. We have updated the statement:

The U.S. National Ice Center's Antarctic sea ice charts (referred to as Ice Charts hereafter) offer weekly products detailing total sea ice concentration, partial concentration, and stage of development (U.S. National Ice Center., 2022).

We updated the reference with a link:

U.S. National Ice Center.: U.S. National Ice Center Arctic and Antarctic Sea Ice Charts in SIGRID-3 Format, Version 1, https://doi.org/10.7265/4b7s-rn93, 2022.

**Q10:** *Section 2.4: The definition of an "average ice type" does not make sense. With regard to the notation, it is more an "ice condition index". A meaningful comparison between ice type and topography is achieved when the concentration of the respective ice type in your window is sufficiently large, e.g. > 80% or even larger. One has to consider that the topography for one ice type can be highly variable (determined by zones of deformation and their areal fractions relative to the smooth level ice areas). Hence one better concentrates on windows for which one ice type is clearly dominant. In your case that should not be a problem.*

**A10:** We revised the manuscript according to this comment together with the latter one Q21.

We did not establish a strict threshold to select the dominant ice type; instead, the ice concentration percentages for each ice type (MYI, FYI, and TI) from the Ice Charts are displayed in the plot, allowing for a more comprehensive visual comparison with the SAR-derived results.

Note that we have checked carefully that the updated plots do not change any given interpretations or conclusions within the Section Results.

The updated plots are shown below and have been added to the revision:

[Figure]

Fig. 1: Sea ice characteristics along the southwards direction along W1 and W2 segments. The blue line in the first column displays the OI percentages derived from SAR images, and the blue dot indicates the ice types obtained from the Ice Charts. The second and third columns plot the elevation ($h_{\mathrm{mod\_SAR}}$) and roughness ($\sigma_R$), respectively. Distance is measured from the northernmost SAR image reference point towards the south. The orange line denotes the average values of $h_{\mathrm{mod_{SAR}}}$ and $\sigma_R$. The box's upper and lower boundaries represent the first (Q1) and third (Q3) quartiles, while the upper (lower) whisker extends to the last (first) sample outside of Q3 $\pm 1.5\times$(Q3-Q1).

[Figure]

Fig. 2: Sea ice characteristics along the southwards direction along W3, W4, and R5 segments. The blue line in the first column displays the OI percentages derived from SAR images, and the blue dot indicates the ice types obtained from the Ice Charts. The second and third columns plot the elevation ($h_{\mathrm{mod\_SAR}}$) and roughness ($\sigma_R$), respectively.

[Figure]

Fig. 3: Sea ice characteristics along the southwards direction along W5 and R1 segments. The blue line in the first column displays the OI percentages derived from SAR images, and the blue dot indicates the ice types obtained from the Ice Charts. The second and third columns plot the elevation ($h_{\mathrm{mod\_SAR}}$) and roughness ($\sigma_R$), respectively.

**Q11:** *Line 148: what are "Pauli-1" and "Pauli-2"-polarizations? Do you refer to the Pauli-representation? Then Pauli-1 is the first and Pauli-2 the second component of the Pauli decomposition which relates surface and volume scattering?*

**A11:** The Pauli decomposition leads to the generation of the "Pauli feature vector" as:

$$k = \frac{1}{\sqrt{2}}[s_{\mathrm{HH}} + s_{\mathrm{VV}}, s_{\mathrm{HH}} - s_{\mathrm{VV}}, 2s_{\mathrm{HV}}]$$

where $s_{\mathrm{HH}}$, $s_{\mathrm{VV}}$ and $s_{\mathrm{HV}}$ are the complex backscattering signal in HH, VV, and HV polarizations, respectively. This representation allows separating odd, even and the 45° tilted bounce components.

Pauli-1 = $s_{\text{HH}} + s_{\text{VV}}$ represents odd bounce, usually the surface scattering. Pauli-2 = $s_{\text{HH}} - s_{\text{VV}}$ represents even bounce. In our SAR data, we do not have HV polarizations therefore only Pauli-1 and Pauli-2 were calculated and used.

In the revision, we have updated the Eq.6 as

Similarly, we can obtain the Pauli-polarization ratio ($R_{\text{pauli}}$) by

$$R_{\text{Pauli}} = \frac{\sigma_{\text{P1}}}{\sigma_{\text{P2}}} = \frac{|s_{\text{HH}} + s_{\text{VV}}|^2}{|s_{\text{HH}} - s_{\text{VV}}|^2} \tag{1}$$

where $\sigma_{\text{P1}}$ and $\sigma_{\text{P2}}$ are denoised SAR backscattering intensity in Pauli-1 and Pauli-2 polarizations in linear scale, respectively. $s_{\text{HH}}$ and $s_{\text{VV}}$ are single-look complex images in dual-pol channels, respectively.

**Q12:** *Lines 166 – 172: see comment (4) above. There is also a misprint on line 172, it should read "hpene >=0.3 m are OI". Was there a special criterion for selecting a threshold of 0.3 m for separating YI and OI?*

**A12:** We have corrected the statement:

$h_{\text{pene}} \geq 0.3\,\text{m}$ are OI

Hallikainen and Winebrenner (1992), in Figure 3-8, suggested that the radar penetration depth of sea ice over multiyear ice ranges from 0.3 to 1 meter at X-band (9.65 GHz - 10 GHz), depending on temperature. Dierking et al. (2017) suggested that ice blocks within ridges often undergo desalination, leading to a greater effective penetration depth within the ridged ice compared to adjacent level ice. This difference in penetration depth can reduce the apparent ridge height relative to the level ice surface as observed in the interferogram. Given that the study area is covered by snow and deformed ice such as ridges, we have selected a penetration depth of 0.3 meters as a threshold for distinguishing between low-penetration and high-penetration ice conditions.

We have added the above texts in Section 3.1 in the revision:

Note that the sea ice penetration depth over multiyear ice is suggested to be $0.3 - 1\,\text{m}$ at the X-band, which varies with temperature (Hallikainen and Winebrenner, 1992). Desalination within ice ridges increases the effective penetration depth compared to level ice (Dierking et al., 2017). Considering these findings and given the study area's snow cover and the presence of deformed ice such as ridges, we chose a penetration depth of $0.3\,\text{m}$ as the threshold for distinguishing the two ice types (i.e., SPI and LPI).

Hallikainen, M. and Winebrenner, D. P.: The physical basis for sea ice remote sensing, Microwave remote sensing of sea ice, 68, 29–46, https://doi.org/10.1029/GM068p0029, 1992

Dierking, W., Lang, O., and Busche, T.: Sea ice local surface topography from single-pass satellite InSAR measurements: a feasibility study, The Cryosphere, 11, 1967–1985, https://doi.org/10.5194/tc-11-1967-2017, 2017.

**Q13:** *Lines 185 – 186: The question is how your "YI" class is related to the ice types listed in Table 2. The WMO-category defines young ice as ice between 10 cm and 30 cm in thickness, as correctly listed in your table. It can be assumed that a penetration depth up to 0.3 m (your "YI") covers the categories "Thin Ice" and "First-Year Ice". The "old ice" with a penetration depth >= 0.3 m cover the thicker FY ice and MY ice. Conclusion is that you should not use the notation YI for penetration depths < 0.3m, see comment 4 above. Check also notations used in section "Conclusions".*

**A13:** We agree with the reviewer. As suggested in A5, we have replaced the "YI" and "OI" terms with the small-penetration condition ice (SPI) and large-penetration condition ice (LPI) throughout the manuscript.

**Q14:** *Figure 6 and Section 2.5: How do you relate hInSAR to the local water surface? Or in other words: which reference surface do you actually use when calculating hInSAR? I suppose that in the initial InSAR processing it is not the local water surface but also the WGS84-elliposid?*

**A14:** The initial height $h_{\text{InSAR\_ini}}$ generated from InSAR processing was referenced to the WGS84 ellipsoid. In theory, we could identify the pixels of the water area and calibrate the InSAR height relative to the water surface by $h_{\text{InSAR}} = h_{\text{InSAR\_ini}} - h_{\text{InSAR\_water}}$. However, due to the low InSAR coherence (less than 0.3) in water areas, these water pixels were masked out as the $h_{\text{InSAR\_water}}$ are inaccurate and can not be used for analyses. Hence, instead of identifying water pixels, we selected smooth and new ice regions, assuming they are thin enough and their elevation (i.e., radar freeboard) is negligible and approximately equal to the water surface. These areas typically exhibit very low SAR backscattering intensity values in both HH and VV polarizations, ranging from -19dB to -18dB, slightly above the noise level of TanDEM-X (-19dB). By generating

a histogram of $h_{\text{InSAR\_ini}}$ values for the thin-ice pixels, we determined the 3rd percentile of the height of these thin-ice pixels to be the local water surface elevation ($\approx h_{\text{InSAR\_water}}$). The threshold value, i.e., the 3rd percentile, is chosen by conducting the aforementioned method over the four SAR scenarios overlaid with DMS DEM (refer to Fig. 2 in the manuscript). When choosing the 3rd percentile as the water surface level, we verify that the calibrated $h_{\text{InSAR}}$ matches the $h_{\text{DMS}}$ at the same level, confirming the validity of the threshold value.

Note that we performed the aforementioned processing for each SAR scene, covering an area of $50 \times 19\,\text{km}$. This approach resulted in one value representing the local water surface for each SAR scene. This method may introduce inaccuracies due to the centimeters level of the radar freeboard of the selected thin and new ice, as well as the fluctuating water surface within each SAR scenario. In the revision, we have included the above text in a new Section Discussion.

**Q15:** *Section 3.2. Here, many things are unclear to me. In summary, I recommend to rewrite this section for the sake of clarity. Single issues: (a) what is the exact definition of m, does it refer to thickness of layer 1 and layer 2? (b) Is hmod the surface elevation above the water surface (or reference ellipsoid)? (c) Which AMSR Level 3 data did you use for retrieving the snow depth over your test sites? How large is the uncertainty of those snow depth values? (d) line 202: "$\phi_{DMS}$ can be transformed into $\phi_{DMS}$ by Eq. (3)" : equation 3 describes the relation between hInSAR and $\phi_\gamma$. I wonder whether this equation can be simply applied using hDMS to derive $\phi_{DMS}$ because for the DMS data there is no height of ambiguity. Is $\phi_{DMS}$ assumed to be the phase at the snow-air interface, hence $\phi_{DMS} = \phi_0$? (e) From equation 8, you obtain m and hv, according to the given definition and Fig. 6 hv is ice thickness which could be mentioned. (f) Why is the second step needed, namely to combine the SAR features with m and hv to obtain modified values m' and hv' ? With the classified images (your maps of ice types), one can directly link the m and hv values from the first step in Fig. 7 with the corresponding ice classification map. The second step is not needed.*

**A15:** (a) In previous study, we proposed a layer-to-layer scattering ratio ($m$) (Huang et al. 2021), inspired from the concept of the layer-to-volume scattering ratio utilized in ice sheeting (Fischer et al., 2018).

The layer-to-layer scattering ratio ($m$) refers to the backscattering power ratio between the top and bottom layer:

$$m = \frac{\sigma_{bottom}(\vec{\omega})}{\sigma_{top}(\vec{\omega})}$$

where $\sigma_{top}\vec{\omega})$ and $\sigma_{bottom}(\vec{\omega})$ denotes the backscattering power from the top and bottom interface, respectively, for a given polarization channel $\vec{\omega}$.

$m$ potentially reveals the relative importance of scattering from these interfaces, depending on factors like interface roughness, dielectric constant, and radar polarization. A larger value of $m$ signifies that surface scattering from the bottom layer predominates, while a smaller $m$ indicates that surface scattering from the top layer is more significant.

(b) $h_{\text{mod}}$ refers to the elevation of the snow-air interface above the water surface.

(c) We used the AMSR-E/AMSR2 Unified L3 Daily 12.5 km Brightness Temperatures, Sea Ice Concentration, Motion & Snow Depth Polar Grids, Version 1 dataset (https://doi.org/10.5067/RA1MIJOYPK3P) for our study. The provided snow depth over sea ice represents a five-day running average. Due to the limited spatial and temporal resolution in the snow depth data, we assumed a constant value of snow depth across one SAR image. For each SAR acquisition, covering a spatial extent of $50 \times 19\,\text{km}$, we computed the mean snow depth and utilized it as input parameter $z_1$ in the TLPV model. In our previous study (Huang et al., 2021), we evaluated the TLPV model's performance concerning variations in snow depth. Our findings indicated a mean discrepancy of $0.31\,\text{m}$ in the derived total freeboard due to snow depth fluctuations ranging from 0.05 to $0.75\,\text{m}$, which covers the major ($> 95\%$) snow depth range in the Weddell Sea during Webster et al. (2018). In future investigations, it would be promising to adapt our proposed total freeboard retrieval method to test sites that coincide with available high-resolution snow depth measurements.

(d) For InSAR, there is a general equation accounting for the relation between the interferometric phase ($\phi$) and height ($h$)

$$h = h_a \frac{\phi}{2\pi}$$

where $h_a$ is the height of ambiguity determined by the specific InSAR configuration such as the radar wavelength, orbit height, incidence angle, and baseline. In instances where we have an external DEM, such as from photogrammetry or lidar, we can also simulate the interferometric phase ($\phi_{\text{DMS}}$) from the height ($h_{\text{DMS}}$) using the same equation tailored to a specific InSAR

configuration giving specific $h_a$.

Yes, $\phi_{\mathrm{DMS}} = \phi_0$.

(e) $hv = z_1 - z_2$ represents the depth between the top ($z_1$) and bottom ($z_2$) interfaces, which contribute to surface scattering effects. However, it's important to note that the bottom layer is not always at the ice-water interface. In some cases, a lower basal sea ice layer can exist, as discussed in Nghiem et al. (2022), which induces strong surface scattering from the ice-basal layer interface. This basal saline layer contains brine inclusions with higher salinity, transitioning toward the ice-seawater interface, as observed in the Western Weddell Sea (Tison et al., 2008). The salinity in this basal layer can be attributed to past flooding events on younger, thinner ice that eventually becomes older and thicker. Furthermore, brines in the upper ice volume above the sea level drain down, further salinating the lower basal layer (Nghiem et al., 2022). Therefore, in scenarios where a high-salinity basal layer is present, $hv$ may not be equivalent to the ice thickness.

Nghiem, S. V., Huang, L., and Hajnsek, I.: Theory of radar polarimetric interferometry and its application to the retrieval of sea ice elevation in the Western Weddell Sea, Antarctic, Earth Space Sci., 9, e2021EA002 191, https://doi.org/10.1029/2021EA002191, 2022.

Tison, J.-L., Worby, A., Delille, B., Brabant, F., Papadimitriou, S., Thomas, D., & Haas, C. Temporal evolution of decaying summer first-year sea ice in the Western Weddell Sea, Antarctica. Deep-Sea Research Part II: Topical Studies in Oceanography, 55(8–9), 975–987. https://doi.org/10.1016/j.dsr2.2007.12.021, 2008

(f) First, $m$ and $h_v$ were calculated by inverting Eq.(8) using $\phi_{\mathrm{DMS}}$ ($= \phi_0$) as input. In the subsequent step, assuming $m$ and $h_v$ from the first step as true values, we established an empirical relation (RF regression) between SAR features and $m$ and $h_v$, enabling estimation of these parameters solely using SAR data, denoted as $\hat{m}$ and $\hat{h}_v$. For the SAR scene without overlapped DMS measurements, the $\hat{m}$ and $\hat{h}_v$ are input into Eq.(8) to generate $h_{\mathrm{mod}}$, which is the total freeboard for the large-penetration condition ice.

In the revision, we have added the above texts and rewritten the section for clarity. We have also updated the method flowchart for clarity, see below:

[Figure]

Fig. 4: (a) The proposed two-step approach for sea ice DEM retrieval. (b) The details (training and validation) of the sea ice classifier and (c) the PolInSAR height retrieval module.

**Q16:** *Fig 9: the right-most color bar is "E", is this the DEM given in meters?*

**A16:** The DEM is given in meters. We apology for the confusion. It is "m" (stands for meters) but printed vertically. We have updated the colorbar label to "meter".

**Q17:** *Figure 11: What is the explanation for the data gaps in the second profile from the top?*

**A17:** The data gaps are the water areas and segments that were excluded due to inaccurate data co-registration. During InSAR processing, pixels with InSAR coherence below $0.3$ were masked out. Additionally, during data co-registration between DMS and SAR, segments lacking distinctive sea ice features in optical and SAR images, and hence, unable to ensure co-registration quality, were eliminated.

We have added explanations in Section 2.3:

Note that the segments lacking distinctive sea ice features in optical and SAR images were eliminated to ensure co-registration quality.

We also added an explanation in the caption of Fig 11:

Figure 11. Comparison between the elevation profiles ($h_{\text{mod\_SAR}}$) from the proposed method and the DMS DEM ($h_{\text{DMS}}$) along the dotted line (from A to B) over Scene No.1-4 in Fig.9(c). The data gaps are the water areas and segments that were excluded due to inaccurate data co-registration.

**Q18:** *Results shown in Figs. 9, 10, 11: hmodSAR is elevation = total freeboard relative to the water surface (or reference ellipsoid), retrieved from pixels of 10 × 10 m in size?*

**A18:** Yes, $h_{\mathrm{modSAR}}$ are total freeboard relative to the water surface retrieved from $10 \times 10\,\mathrm{m}$ pixel size. In the revision, we have added the texts at the second paragraph in Section 4.1:

Note that $h_{\mathrm{modSAR}}$ represents total freeboard relative to the water surface retrieved from the pixel at $10 \times 10\,\mathrm{m}$ spacing size.

**Q19:** *Again Figs. 9, 10, 11: For "old ice", height values = total freeboard of even larger than 3 m are measured. Intuitively, this seems to be not very realistic, although DMS and SAR values match. Is there any information available about the area regarding ice and snow conditions which seems to be special when comparing to the results of the other profiles shown in Figs. 13-15 which reveal smaller heights? If you used the WGS84 ellipsoid as reference: Is the difference between reference ellipsoid and mean sea level larger close to the Antarctic Peninsula? Perhaps also icebergs biased the measurements?*

**A19:** The four scenarios depicted in Figures 9, 10, and 11 in the manuscript are located near the Eastern Antarctic Peninsula (see Fig 2 in the manuscript for the geolocation). Notably, we observed considerable deformation and prominent ridges in the sea ice from the DMS photo. While, upon thorough examination of DMS photos across the four scenes, no icebergs were identified. Therefore, we think it is the deformed ice and ridges contribute to the high total freeboard values.

Here, we show a DMS photograph captured a sub-region within Scene No.1, see Fig. 5(a) below. In Fig. 5(b), we used the DMS DEM derived from this photograph, annotated specific sea ice features, and extracted corresponding elevation values.

[Figure]

Fig. 5: The red rectangle indicates the selected sub-region overlapped by DMS measurements. (b) The optical photo alongside elevations derived from the DMS DEM within the selected sub-region. The elevation values are referenced to the WGS84 Ellipsoid.

The elevation values in Fig. 1(b) are referenced to the WGS84 Ellipsoid, representing the original values obtained from the DMS DEM products. The water surface ranges between 12.4-12.5m, while the thin ice adjacent to the water ranges from 12.5-13.0m, indicating a relative elevation of 10-50 centimeters compared to the water surface. On the ice floes, the smooth surfaces measure around 13.0-14.0m, representing an elevation of 0.5-1.5m above the water surface. The rough surfaces, including deformed ice and ridges, exhibit elevations ranging from 14.5-17m, with corresponding relative elevations to the water surface exceeding 3m.

**Q20:** *Figure 12: The color bars show values of hmodSAR down-sampled to a resolution of 500 m?*

**A20:** Yes, the results in Sections 4.2, 4.3, and 4.4 (including Fig.12-16) are all based on the $h_{\mathrm{modSAR}}$ down-sampled to $500\,\mathrm{m}$. We stated this in the beginning of Section 4.2. In the revision, we have added the statement in the caption of Fig.12:

$h_{\mathrm{mod\_SAR}}$ was downsampled to $500\,\mathrm{m}$ pixel size.

**Q21:** *Figures 13,14,15, left column: The percentage of "OI" refers to the large-penetration-depth category. It would be much easier to discuss the relationship between the OI and real ice types when concentrations of MY, FY, and TI are shown*

*instead of an index related to the "average ice type". Lines 241-243: In Fig. 13, the percentage of OI is only 58% for distances between 0 and 160 m, but the ice type in the ice chart seems to be almost 100% MYI ice since the "average ice type" value is close to 3. In the range between approximately 300 and 600 km, MYI is also close to 100% concentration, but the percentage of OI is down to 20-25%. (Note that according to section 2.4, the indices for TI, FYI, and MYI should be 0,1,2, respectively, and not 1,2,3) This demonstrates that the penetration depth is not directly linked to the ice types shown in the ice chart.*

**A21:** We have renamed the "OI" as large-penetration condition ice (LPI) in the revision. We have updated the results by showing the ice types with concentrations, see **A10**.

The ice concentration values for MYI from Ice Charts and the LPI percentage from SAR are not necessarily identical, as they are classified by different criteria. Nevertheless, our study observed and discussed a similar trend between the dominant ice types and ice concentrations from the Ice Charts and the LPI percentage from SAR. We have replaced lines 241-248 in the previous with the texts below:

The overall trend of estimated OI percentages correlates well with dominant ice types and ice contractions from the Ice Charts across most segments (W1-U, W1-L, W2-U, W4, W5-U, W5-L, and R5). Specifically, W1-U and W1-L are explained as two examples. W1-U from 0-120km is covered by 100% MYI, where the LPI percentage reaches its highest value (58%). As the dominant ice transitions from MYI to FYI from 120-400km, the LPI percentage decreases accordingly. For W1-L, where the distance between 500-600km is covered by 100% MYI, and the LPI percentage peaks before decreasing after 650km distance as FYI becomes dominant. The lowest LPI percentage is found around 1200km, consistent with the occurrence of TI at this distance.

The observed similar trend can be explained by the general assumption that MYI is thicker and less saline, allowing for deeper radar penetration compared to FYI and TI. However, penetration depth is influenced by various factors not only ice age, but also ice salinity, snow condition, flooding effects, and temperature. This explains the discrepancies for other segments (W2-L, W3-U, W3-L, R1-U, R1-L). Furthermore, discrepancies can also be attributed to differences in spatial resolution and temporal gaps between Ice Charts and SAR imagery, considering the dynamic nature of sea ice.

It is essential to clarify that we utilized Ice Charts data as an external information source to interpret the classification results and spatial variation of topography. However, we did not use Ice Charts to quantitatively validate the proposed method. For validation purposes, we conducted pixel-by-pixel comparisons using co-registered DMS data.

**Q22:** *Line 253-261: Referring to the statement: "In general, the region with thicker ice (e.g., MYI) is anticipated to display higher elevation or larger roughness compared to the area with thinner ice, such as FYI and TI": Locally, rough FYI may reveal a larger roughness than smooth level MYI, and it may reveal a higher elevation when covered by a very thick snow cover. This may also explain discrepancies.*

**A22:** Agree. In the revision, we have added the explanation for the discrepancies as suggested:

These discrepancies may arise due to the local cases where rough FYI exhibits greater roughness than smooth level MYI. FYI may also show higher elevations when covered by very thick snow.

**Q23:** *Line 262: "sea ice ... exhibits...highest elevation". You should again clearly state that the values of elevation are total freeboard, i.e. also include the snow layer.*

**A23:** We agree. In the revision, we have replaced "elevation" with "total freeboard" throughout the manuscript.

**Q24:** *Line 265: Figs. 6g-l in the paper by Wang et al (2020) are indeed well suited for comparing with your results. The values they show (Fig. 6l for 2017), however, have only a narrow peak at 1.5 to 2.5 m, otherwise values are lower. Figs 6g-l should also be mentioned with regard to your Figs 13-15, considering window sizes for averaging the elevation.*

**A24:** Upon thorough examination of Wang et al.'s work (2020), we were unable to find the resolution (window size) they used for plotting Fig. 6l. However, based on their mention of a $20\,km$ width track, we estimate that each dot corresponds to a window size larger than $20\,km$, significantly exceeding the $500 \times 500\,m$ window size we utilized. Unfortunately, Wang et al. did not provide the processed data used in Fig. 6l in the supplements, preventing a quantitative comparison with our results. Therefore, we conducted a visual comparison based on geolocation, comparing Fig. 6l in Wang et al. (2020) with the four segments (W2-U, W2-L, W3-U, and W3-L) in our study, as shown in Fig. 13 and 14 in the manuscript.

We have included Fig. 6l below (as shown in Fig. 6), and we define the two tracks in Fig.6l as Track-W and Track-E. The region with latitude $< 70°S$ ($> 70°S$) is referred to as the northern (southern) track.

[Figure]

Fig. 6: The total freeboard from the paper by Wang et al., 2020 (Fig.6l). We labeled the two tracks Track-W and Track-E. The region with latitude $< 70°$S $(> 70°$S) is referred to as the northern (southern) track.

The Northern and Southern Track-W segments are partially overlaid with W2-U and W2-L, respectively. In our study, the total freeboard of W2-U and W2-L are shown in the third and fourth rows (medium column) in Fig. 13 (in the manuscript). Our result shows that the total freeboard reaches a mean value of $\sim 1$ m and 75% percentile value of $\sim 1.5$ m within the first $100$ km, which agrees with the red dot in Fig.6l Track-W. Then, the total freeboard goes down to a mean value as $\sim 0.7$ m and 75% percentile value of $\sim 0.75$ m from $100 - 200$ m. Although there is a data gap in the Fig.6l (Wang's paper), we can see that the color of dots changes from red to yellow, which is consistent with the decreasing trend of the total freeboard within $300$ km from our results. For W2-L, the total freeboard from results are around at the mean values of $0.5$ m, agree with a mix of green and yellow dots ($0.5 - 0.9$ m) in the Southern Track-W in Fig.6l (Wang's paper).

Note that the OIB ATM data used for Fig.6l was acquired on 14th and 22rd November 2017, while the SAR images in our study were acquired on 30th and 25th October 2017 for W2-U and W2-L. The sea ice drift or potential melting could induce slight differences in our results and Wang's results.

The W3-U and W3-L can be compared with Northern and Southern Track-E, respectively. From Fig. 6l, a mix of yellow and green dots in the northern Track-E represent the total freeboard $0.5 - 1.2$ m total freeboard, which agrees well with our result in W3-U, see the first row in Fig.14 (in the manuscript). At around $70°$S degree, the dots transit to cyan and blue, representing the total freeboard of $0.2 - 0.7$ m, which is consistent with the W3-L in Fig.14 (in the manuscript). The slight difference can be attributed to the temporal difference of SAR images used in our study. Specifically, the image for W3-L was acquired on October 26, 2017, while the Track-E image was acquired on November 22, 2017

We have added the above analyses in the revision Sections 4.2 and 4.3.

**Q25:** *Line 283: ice type "MTI"? I think it should be MYI.*

**A25:** Corrected.

**Q26:** *Line 294-295: Sentence: "The variation of the roughness along the R1 segment also highlights the importance of combining topographic mapping with ice category mapping to comprehensively characterize sea ice features." Since there is no direct relationship between ice type and topography data, it is not per se "important" to combine both, but can be useful in certain cases, e.g. for operational ice charting. Since the edges of ice floes with open water between the floes contribute to the ice roughness, ice concentration may also be a useful parameter to be combined with ice topography in some cases.*

**A26:** We agree with the reviewer. We have modified the statement in the revision:

The variation of the roughness along the R1 segment suggests that ice topography provides add-on information that can be useful to be integrated into the operational ice charting. Furthermore, Since the edges of ice floes with open water between the floes can also contribute to the ice roughness, combining ice topography with ice concentration can help characterize the sea ice cover more comprehensively.

**Q27:** *A point of my interest: For radar applications, it would also be useful to show how large deviations between DMS and SAR values (= your penetration depth) can be, and where this occurs (spatial distribution along your tracks).*

**A27:** This is a super interesting point. Considering the studied area is mainly covered by snow, we believe the penetration largely depends on the local snow conditions and snow spatial distributions. Unfortunately, our current dataset does not have in-situ measurements of snow properties, so we cannot investigate the deviation with the snow parameters. In the future, it would be interesting to propose/conduct fieldwork that includes TanDEM-X acquisitions, lidar measurement, and in-situ snow measurements and analyze the relation between the radar freeboard and total freeboard at varying snow depths.

We have added texts in the Section Conclusion as:

The spatial distribution of penetration depth (total freeboard minus radar freeboard) can be an interesting topic for future research. In snow-covered sea ice, penetration is significantly influenced by local snow conditions. Hence, conducting a coordinated campaign encompassing TanDEM-X acquisitions, lidar measurements, and in-situ snow assessments holds great promise for analyzing the relation between radar freeboard and total freeboard across different snow depths.

Again, we sincerely thank the editor and reviewers for helping us improving the manuscript.

---

## Author Response (AR1)

**Response to the comments of Reviewer 1**

First of all, we would like to thank the anonymous reviewer for the careful review and valuable suggestions. We carefully revised the manuscript following the suggestions. Hereby we give a point-by-point reply to address the comments. In this document, the words in *italics are the reviewers' comments*, the words in blue are the modifications we have made in the revision, and others are our responses.

**Q1:** *The work aims to study the sea ice freeboard in areas around the Antarctic peninsula. Improved sea ice surface topography is a useful product and can be implemented in other studies, such as ice drift product development and climate studies. The manuscript is reasonably well written and mostly easy to follow. Some of the terminology is at times somewhat confusing. There are a great many figures in the manuscript, could some perhaps be moved to supplementary information.*

**A1:** We thank the reviewer for the positive comment about our research. We have carefully revised the manuscript based on the following comments. We have relocated Figures 5, 8, and 12 from the previous version of the manuscript to the appendix.

**Q2:** *In the abstract the terms sea ice DEM, i.e. snow freeboard is introduced. How does this relate to the sea ice topography? Why is the sea ice DEM not = sea ice and snow freeboard? How is the air-ocean-ice system related to the sea ice topography? The statement as it stands right now is a bit challenging to interpret.*

**A2:** We apologize for any confusion caused by the term "snow freeboard." In this paper, when referring to sea ice DEMs or "sea ice elevation," we mean the total freeboard, which includes both the ice freeboard and the thickness of the snow layer. In the revised version, we have explicitly defined sea ice DEMs as total freeboard (snow+ice). We have made corrections by replacing "snow freeboard" with "total freeboard" and updated the term "sea ice elevation" to "total freeboard" throughout the manuscript, including both the text and figures, to ensure clarity and consistency.

In the revision, we have restated the sentence in the abstract Line 2: The total freeboard (snow+ice) is crucial for reflecting sea ice dynamics and interpreting the geophysical environments of polar oceans.

In the Introduction section Line 20-24, we have included detailed explanations about how the air-ocean-ice system is related to sea ice topography:

The sea ice topography plays a crucial role in reflecting sea ice dynamics and interactions within the air-ocean-ice system. It showcases the spatial distribution of distinct surface features such as snow dunes (Trujillo et al., 2016; Iacozza and Barber, 1999) and deformed ice (Haas et al., 1999; Petty et al., 2016), which are impacted by the forces from winds and currents. Moreover, the atmospheric drag coefficient over sea ice, which is topography-dependent, is an important parameter for understanding interactions at the ice-atmosphere boundary (Garbrecht et al., 2002; Castellani et al., 2014).

Castellani, G., Lüpkes, C., Hendricks, S., and Gerdes, R.: Variability of Arctic sea-ice topography and its impact on the atmospheric surface drag, J. Geophys. Res.: Oceans, 119, 6743–6762, https://doi.org/10.1002/2013JC009712, 2014.

Garbrecht, T., Lüpkes, C., Hartmann, J., and Wolff, M.: Atmospheric drag coefficients over sea ice–validation of a parameterisation concept, Tellus A: Dynamic Meteorology and Oceanography, 54, 205–219, https://doi.org/10.3402/tellusa.v54i2.12129, 2002.

Haas, C., Quanhua, L., and Thomas, M.: Retrieval of Antarctic sea-ice pressure ridge frequencies from ERS SAR imagery by means of in situ laser profiling and usage of a neural network, International Journal of Remote Sensing, 20, 3111–3123, 1999.

Iacozza, J. and Barber, D. G.: An examination of the distribution of snow on sea-ice, Atmosphere-Ocean, 37, 21–51, https://doi.org/10.1080/07055900.199 1999.

Petty, A. A., Tsamados, M. C., Kurtz, N. T., Farrell, S. L., Newman, T., Harbeck, J. P., Feltham, D. L., and Richter-Menge, J. A.: Characterizing Arctic sea ice topography using high-resolution IceBridge data, The Cryosphere, 10, 1161–1179, https://doi.org/10.5194/tc-10-1161-2016, 2016.

Trujillo, E., Leonard, K., Maksym, T., and Lehning, M.: Changes in snow distribution and surface topography following a snowstorm on Antarctic sea ice, Journal of Geophysical Research: Earth Surface, 121, 2172–2191, https://doi.org/10.1002/2016JF003893, 2016.

**Q3:** *R21. If we assume that the DEM is snow freeboard, should it then be assumed that no penetration if the snow is possible?*

**A3:** The term "sea ice DEMs" refers to the total freeboard, which is ice freeboard plus snow depth. We have modified it to 'total freeboard' throughout the manuscript.

**Q4:** *R22-23. Please elaborate how this product is essential for assessing the impact of climate change on sea ice.*

**A4:** We have revised in the introduction Line 26-28: The DEM (i.e., total freeboard) can be converted to thickness with the knowledge of snow depth and the assumed values of snow, ice, and seawater densities (Kwok and Kacimi, 2018). Estimating sea ice thickness over time offers valuable insights into the overall stability of sea ice in the changing climate.

**Q5:** *R58-59. How can the DEM help separate the different ice types?*

**A5:** We do not intend to imply that the DEM can help separate different ice types. Rather, we mean that sea ice types can serve as prior knowledge for generating DEM from InSAR. The InSAR-derived height represents the elevation of the InSAR phase center, which is not necessarily at the snow-air surface due to radar penetration. The depth of penetration varies depending on the ice type, with sea ice having lower salinity and being covered by dry snow allowing for deeper penetration compared to ice with higher salinity and wet snow cover. To ensure accurate sea ice DEM retrieval, we proposed first classifying the sea ice into small-penetration and larger-penetration conditions, followed by retrieving the sea ice DEM using standard InSAR processing and the proposed TLPV model for each ice type, respectively.

In the revision Line 61-68, we have rewritten the sentence for clarity:

The initial step involves the development of a random forest classifier using specific SAR features to categorize sea ice into two groups: small-penetration condition ice (SPI) and large-penetration condition ice (LPI), based on the penetration depth of microwaves into the snow and ice. Subsequently, a sea ice DEM is created for each ice type. In the case of SPI, standard InSAR processing is applied to determine the total freeboard. For LPI, a novel inversion algorithm is proposed to estimate the parameters of the developed TLPV model (Huang et al., 2021). This model allows for correcting penetration bias in the InSAR signal over LPI, resulting in an accurate retrieval of the total freeboard. We validate the proposed method against the photogrammetric DEM from the IceBridge aircraft. A root-mean-square error (RMSE) of $0.26 \, \text{m}$ between the derived DEM and reference data indicates an improved accuracy in total freeboard retrieval.

**Q6:** *R67. What is meant with Ice Chart here? An operational ice charts such as those provided by the ice services. What is meant is explained on R108. This one of the terminology words introduced it the introduction that gets explained later in the manuscript. This terminology should either be removed from the introduction or needs to be explained here.*

**A6:** We have removed the terminology and revised the sentence (Line 74) as:

...sea ice classes obtained from an operational product from the U.S. National Ice Center.

**Q7:** *R91. Perhaps state how the denoising is done then why it's useful/essential to do so here.*

**A7:** The SAR-measured backscattering intensity ($\sigma_{\text{measure}}$) containing additive thermal noise can be denoted as

$$\sigma_{\text{measure}} = < (S_{\text{denoised}} + N) \times (S_{\text{denoised}} + N)^* > \tag{1}$$

where $S_{\text{denoised}}$ is the noise-subtracted backscattering amplitude, and $N$ is the additive thermal noise. Considering $S_{\text{denoised}}$ and $N$ to be uncorrelated, the noise-subtracted backscattering intensity can be obtained from the following simple equation (Nghiem et al., 1995)

$$\sigma_{\text{denoised}} = \sigma_{\text{measure}} - NESZ \tag{2}$$

where $NESZ$ is the noise floor (i.e., the noise equivalent sigma zero (NESZ)), and all terms are in the linear scale.

The TanDEM-X product contains a set of polynomial coefficients that describe the NESZ pattern for each polarization along the range direction (Eineder et al., 2008) for both the TanDEM-X (TDX) and TerraSAR-X (TSX) images. An example of the calculated $NESZ$ is shown in Fig.1 in dB scale. By converting to the linear scale, the $\sigma_{\text{denoised}}$ can be calculated by subtracting $NESZ$ from the $\sigma_{\text{measure}}$. We calculate the NESZ pattern for each SAR acquisition and employ Eq. 2 to generate denoised backscattering intensities for the different polarizations (i.e., HH, VV, Pauli-1 (HH+VV), and Pauli-2 (HH-VV)) from the TSX image. Note that for Pauli-1 and Pauli-2, we use the average $NESZ$ between HH and VV channels.

[Figure]

Fig. 1: NESZ patterns for one TanDEM-X acquisition (Scene No.1, see Fig.2 in the paper) as an example.

In the revision, we have added the above description in the appendix A2.

Thermal noise can contaminate the SAR backscattering intensity. Removing the thermal noise allows for a better representation of sea ice features from SAR image, which is crucial for ice classification. The denoised SAR backscattering intensity was input as the features for the random forest classifier in Section 3.1.

In the revision, we have added the usefulness of the denoising processing, Line 99-104:

The backscattering intensity $\sigma_{\mathrm{measure}}$ of the images includes additive thermal noise, which can be described by the noise equivalent sigma zero (NESZ) and assumed to be uncorrelated with the signal (Nghiem et al., 1995). Removing the thermal noise allows for a better representation of sea ice features, which is crucial for ice classification. We denoised backscattering intensities for the different polarizations (i.e., HH, VV, Pauli-1 (HH+VV), and Pauli-2 (HH-VV)) by subtracting the noise equivalent sigma zero (NESZ) from the $\sigma_{\mathrm{measure}}$. More details about the thermal noise removal can be found in the appendix A2.

Nghiem, S., Kwok, R., Yueh, S., and Drinkwater, M.: Polarimetric signatures of sea ice: 2. Experimental observations, J. Geophys. Res.:490 Oceans, 100, 13 681–13 698, https://doi.org/10.1080/08843759508947700, 1995.

Eineder, M., Fritz, T., Mittermayer, J., Roth, A., Boerner, E., & Breit, H. (2008). TerraSAR-X ground segment, basic product specification docu-ment (Tech. Rep.). Cluster Applied Remote Sensing (CAF).

**Q8:** *Table 1. One of the datasets (R5) has a higher HoA. Does this affect the results presented here?*
**A8:** The HoA ($h_a$) is the height of ambiguity determined by the specific InSAR configuration such as the radar wavelength, orbit height, incidence angle, and baseline. $h_a$ is used in converting InSAR phase ($\phi_\gamma$) into height ($h_{\mathrm{InSAR}}$) through $h_{\mathrm{InSAR}} = \frac{\phi_\gamma}{2\pi}h_a$.

However, a larger HoA will elevate the uncertainty in the InSAR-derived height. This uncertainty ($\sigma_h$) can be estimated by (Madsen and Zebker, 1998)

$$\sigma_h = \frac{h_a}{2\pi}\sigma_{\Delta_\phi}$$

where $\sigma_{\Delta_\phi}$ is the phase noise, which can be expressed as a function of the interferometric coherence ($\gamma_{\mathrm{InSAR}}$) and the independent number of looks ($N_L$) (Rosen et al., 2000)

$$\sigma_{\Delta_\phi}^2 = \frac{1}{2N_L}\frac{1 - \gamma_{\mathrm{InSAR}}^2}{\gamma_{\mathrm{InSAR}}^2}$$

The simulated $\sigma_h$ to the variations in $h_a$ and $\gamma_{\mathrm{InSAR}}$ is illustrated in Fig. 2. At the $\gamma_{\mathrm{InSAR}} = 0.75$, $\sigma_h$ increases from $0.35\,\mathrm{m}$ to $0.48\,\mathrm{m}$ corresponding to $h_a$ ranging from $30\,\mathrm{m}$ to $42\,\mathrm{m}$. Across the studied region, both the mean and median values of

$\gamma_{\mathrm{InSAR}}$ are around $0.75$. Consequently, in the case of R5, the larger $h_a$ induces a relatively larger average uncertainty in the derived InSAR height ($h_{\mathrm{InSAR}}$) compared to the smaller $h_a$ InSAR configuration in our dataset.

[Figure]

Fig. 2: Simulation of $\sigma_h$ to the variations in $h_a$ and $\gamma_{\mathrm{InSAR}}$. In our case $N_L = 73$.

In the revision, we have added sentences in Section 2.2 Line 94-96 and have included the above analyses in the Appendix A1.

Note that for R5, the larger HoA leads to a relatively higher average uncertainty in the derived InSAR height ($h_{\mathrm{InSAR}}$) compared to other InSAR configurations with smaller HoA. More details can be found in the appendix A1.

Madsen, S. N. and Zebker, H. A.: Imaging Radar Interferometry, in: Principles and Applications of Imaging Radar, Manual of Remote Sensing, 3rd Edn., John Wiley & Sons, New York, 2, 359380, 1998.

Rosen, P. A., Hensley, S., Joughin, I. R., Li, F. K., Madsen, S. N., Rodriguez, E., and Goldstein, R. M.: Synthetic aperture radar interferometry, P. IEEE, 88, 333–382, https://doi.org/10.1109/5.838084, 2000.

**Q9:** *R111. The spatial resolution of the Ice Charts is 10 x 10 km. How wide are the SAR images used? Will more than a few pixels be comparable between the Ice Charts and the SAR images?*

**A9:** In Fig. 8-10, we presented the sea ice topography variation (total freeboard and roughness from SAR) at $100\,\mathrm{km}$ intervals. Specifically, we selected the SAR pixels at each $100\,\mathrm{km}$ distance and computed statistics for their total freeboard and roughness. The resolution of each SAR pixel is $500 \times 500\,\mathrm{m}$. As we directly plotted the values from the Ice Charts which have a spatial resolution of $\sim 10 \times 10\,\mathrm{km}$, there are more than a few SAR pixels compared to a single Ice Chart pixel. We do not down-sample the SAR results to match the Ice Charts resolution, since our objective is to generate a high-resolution (sub-kilometer) sea ice DEM and explore its role in understanding the spatial variation of sea ice topography (as detailed in Section 4.2). It is also essential to clarify that we utilized Ice Charts data merely as external information in interpreting the topographic variation. However, we did not employ the Ice Charts for validating the proposed method. For validation purposes, we conducted pixel-by-pixel comparisons using co-registered DMS data, which were down-sampled to the same resolution level as the SAR result.

**Q10:** *Figure 3, 4 and 7. The schematic in Figure 3 in itself is good but it's challenging to understand if perhaps Figure 4 is step 1, and if so why this isn't stated in Figure 4. Please indicate how these 3 flow charts are interconnected. It appears as if Step 1 is in part explained in Figure 4 but it's unclear as more information than the TanDEM-X SAR images are used as input data? And the classification map at the end of Figure 4 appears to perhaps be the first box in Step 1. Figure 7 appears to be an explanation of the top right box in Step 2 in Figure 3. Please clarify these flow charts.*

**A10:** Figure 3 gives an overview of the proposed method which includes Steps 1 and 2. Figure 4 provides a detailed explanation of the training process for the sea ice classifier in Step 1. Figure 7 elaborates on the PolInSAR height retrieval module in Step 2, including inverting the proposed TLPV model to generate $h_{\mathrm{mod}}$. During the training and validation phases

of both the sea ice classifier and the PolInSAR height retrieval module, DMS DEMs were input as reference data. With the trained classifier and the module, the two-step method was applied to TanDEM-X SAR images together with AMSR level-3 snow depth measurements as input. We have merged the three figures into one for clarification, see Fig. 3:

[Figure]

Fig. 3: (a) The proposed two-step approach for sea ice DEM retrieval. (b) The details (training and validation) of the sea ice classifier and (c) the PolInSAR height retrieval module.

The above flowchart has been added to the revision.

**Q11:** *R180-184. Are some parameters more important for one of specific ice types? Or is the importance level presented in Fig 5 universal?*

**A11:** The Gini importance is a metric used in a random forest classifier to measure the relative importance of each feature in making classification decisions. It is calculated based on the decrease in Gini impurity that each feature contributes to the overall model.

We have added the more texts in the revision Line 212-215:

Note that the computed Gini importance is not inherently specific to a particular class or ice type. Instead, it represents the relative importance of features in making overall classification decisions within the context of the entire dataset. Therefore, the importance level determined by Gini importance is not specific to individual ice types but reflects the significance of features for the classifier's overall predictive performance across all classes.

**Q12:** *Figure 9. Some of the leads appear to have a light blue color, not the same as for the YI. Why is that? Which ice type do they represent? They appear to in 1, 2 and 3 have the highest E. What is the unit E? Does a low SNR perhaps get mistaken as a thick sea ice? Perhaps could a noise analysis remove erroneous values?*

**A12:** We apologize for the confusion. The light blue is the color of the base map used for plotting. During InSAR processing, we excluded pixels with an InSAR coherence less than 0.3, setting their values to NaN. When plotted, NaN values are rendered as void areas, showing the color of the base map. Since pixels with low InSAR coherence often correspond to water areas, it appears that all leads and water areas are colored in light blue. The SAR backscattering intensities in the water area mostly range below -19dB. Note that the system noise level of TanDEM-X is around -19 to -26dB. These pixels exhibiting low SNR induce significant uncertainty in InSAR processing. As a result, we have excluded these regions (i.e., water/ leads) from further analyses.

In the revision, we have updated the Fig. 5. We have changed the color of the base map to transparent (white) and added a statement in the caption: The void pixels in the second and third columns represent water areas excluded from processing due to $\gamma_{\mathrm{InSAR}} < 0.3$.

In the third column of the figure, the label was a vertically printed "m" which stands for meter, not "E." This column represents the derived sea ice DEM (i.e., total freeboard, in the unit of meter) using the proposed two-step method. Scenes 1, 2, and 3 near the Antarctic Peninsula exhibit higher total freeboard. The analysis of the higher freeboard and the spatial variation along the transect are provided in Section 4.2.

For Fig.5 in the revision, we have changed the colorbar's label to "total freeboard (m)" for clarity.

**Q13:** *Figure 11. In the top, upper middle and bottom figures, it appears as if the SAR estimates are underestimating the high and low peaks. Is this a resolution issue? Or is there some other explanation behind this?*

**A13:** In the postprocessing, we geocoded the DMS DEM into the SAR coordinate and down-sampled the DMS DEM into the same resolution as the SAR pixel size ($10 \times 10\,\mathrm{m}$ in the ground range and azimuth). Therefore, resolution is not likely the cause of underestimation.

One factor contributing to the underestimation of total freeboard could be the assumption of a constant average snow depth over one SAR scene (spatial coverage of $50 \times 19\,\mathrm{km}$). In our methodology, we assume this snow depth remains uniform across one SAR image due to the limited spatial resolution of available snow measurements (AMSR Level-3 data with a resolution of $12.5\,\mathrm{km}$). Therefore a constant value (i.e., average snow depth across one SAR scene) is input as parameter $z_1$ to the TLPV model. However, this uniformity may lead to an underestimation of snow depth in high-peak areas such as ridges, consequently resulting in an underestimation of the total freeboard. Our prior study (Huang et al., 2021) demonstrated a mean difference of $0.31\,\mathrm{m}$ in the derived total freeboard due to snow depth variations from $0.05$ to $0.75\,\mathrm{m}$ over Scene No.1, highlighting the impact on peak estimation. In the future, it would be interesting to adapt the proposed total freeboard retrieval method over the test sites co-locating with available high-resolution snow depth measurements.

Another factor that could potentially lead to the underestimation of high and low peaks is the residual shift between the SAR and DMS images. Although we carefully co-registered the four SAR scenes with the DMS data, the co-registration can not be perfect. In the process, we divided the entire overlapped transect into small patches (each corresponding to $100 \times 1000\,\mathrm{m}$). We assumed the same drift location over one patch and no rotation; thus, only one shift vector was used for co-registration over each patch. This could result in small residual shifts when the ice floes or features do not drift at the same velocity or involve rotations. The presence of low- and high-peak ice features with narrow sizes spanning just a few pixels, poses a challenge. Even slight residual shifts, as small as 1-2 pixels, can lead to loss or misalignment of peak structures in SAR images. Consequently, these slightly misaligned SAR images input into the proposed model may result in an underestimation of the total freeboard.

We have added the above text in a new Section 5.1 in the revision.

Huang, L., Fischer, G., and Hajnsek, I.: Antarctic snow-covered sea ice topography derivation from TanDEM-X using polarimetric SAR455 interferometry, The Cryosphere, 15, 5323–5344, https://doi.org/10.5194/tc-15-5323-2021, 2021.

**Q14:** *Figure 12. This figure could perhaps be moved to supplementary information as it doesn't add much to the understanding of the results. It's very challenging to see the elevations, if kept perhaps make the SAR images a lot larger?*

**A14:** In the revision, we have enlarged the size of the figure and have moved it to the appendix: Fig. A5 and A6.

**Q15:** *Figure 13, 14, 15. Consider coloring the y-axis and the color used in the plot the same color for easier interpretation of the information contained within the figures. Add a legend to the two rightmost columns, to explain what the blue and the orange represents.*

**A15:** In the revision, we have improved these figures by incorporating visualizations of ice concentration for each ice type (MYI, FYI, and TI) from the Ice Charts, instead of only showing the "average ice type" as in the previous manuscript version. We have utilized a consistent colormap (same as Fig.11(b) in the manuscript) to represent each ice type. We have included a legend for clarity in the second and third columns as suggested.

Note that these updated plots are primarily for improved visual comparison between the Ice Charts and the SAR results, with no alterations to the main observations or conclusions within the Section Results.

The updated figures can be found in Fig 8-10 in the revision.

**Q16:** *R2 "... a digital ..." or "digital elevation models"*
**A16:** Done

**Q17:** *R2-3 should it be drifting sea ice instead of drift sea ice?*
**A17:** Done

**Q18:** *R60-61. "sea ice elevation" has already been defined earlier in the manuscript.*
**A18:** The repeated statement has been removed.

**Q19:** *R76. With sequence is it meant orbit?*
**A19:** Yes, it means a series of acquisitions within some seconds along the same orbit. We revised the sentence Line 83-84: The footprints consist of 12 segments, each corresponding to a sequence of SAR images within the same orbit, all acquired at nearly the same time, with only seconds varying between them.

**Q20:** *R143. Wakabayashi et al 2004 used L-band SAR, how will this compare to the X-band SAR used here? Can we derive sea ice thickness using X-band SAR?*
**A20:** The work (Wakabayashi et al 2004) suggests that the co-polarization ratio from L-band SAR image can be related to ice-thickness. However, as far as we know, no published results demonstrate a relation between the co-polarization ratio from X-band SAR and ice-thickness. Nevertheless, the co-polarization coherence from TerraSAR-X has been demonstrated to be correlated to ice thickness over multi-year sea ice (Kim et al., 2011). This reference was cited in Section 2.6.3.

Kim, J.-W., Kim, D.-j., and Hwang, B. J.: Characterization of Arctic sea ice thickness using high-resolution spaceborne polarimetric SAR data, IEEE Trans. Geosci. Remote Sens., 50, 13–22, https://doi.org/10.1109/TGRS.2011.2160070, 2011.

In the revision Line 168-171, we have clarified that the reference is based on L-band SAR data:

$R_{\text{coPol}}$ extracted from L-band SAR images is associated with the dielectric constant and has therefore been used as an indicator of ice thickness (Wakabayashi et al., 2004). Further investigation is required to determine if $R_{\text{coPol}}$ from the X-band can also serve as a proxy for ice thickness.

**Q21:** *R198. The reference can be shortened to (Meier, Markus and Comiso, 2018)*
**A21:** Done

**Q22:** *R248. "... in the Ice Charts"*
**A22:** Done

**Q23:** *R283. Sea ice doesn't evolve from MYI to TI. TI can evolve to MYI through surviving at least 2 seasonal cycles.*
**A23:** We refer to the spatial transition of sea ice types from FYI to TI in the southeastward direction.
We have revised the sentence in Line 342:
This observation aligns with the transition of dominant ice types from FYI to TI in that direction.

**Q24:** *R363-367. It this information needed here?*
**A24:** We prefer to keep these sentences as a summary of our observations regarding sea ice DEMs. This enables readers who may skip the detailed reading of Sections 4.2 and 4.3 to still grasp some key take-home messages.

Again, we sincerely thank the editor and reviewers for helping us improve the manuscript.

**Response to the comments of Reviewer 2**

First of all, we would like to thank the anonymous reviewer for the careful review and valuable suggestions. We carefully revised the manuscript following the suggestions. Hereby we give a point-by-point reply to address the comments. In this document, the words in *italics are the reviewers' comments*, the words in blue are the modifications we have made in the revision, and others are our responses.

**Q1:** *In this interesting paper, a new method for the retrieval of total ice freeboard (ice freeboard plus snow thickness) from single-pass interferometric SAR is developed and applied to the Weddell and Ross Seas. The SAR-derived sea ice topography is validated by independently measured sea ice freeboard profiles and analyzed in comparison to several studies, which support the results. The paper should definitely be published, but I recommend modifications which concern the use of certain terms and the need for additional information. The latter is in particular important for the description of the method.*

**A1:** We thank the reviewer for the positive comment about our research. We have carefully revised the manuscript based on the following comments.

**Q2:** *Line 3: "accurate sea ice DEMs (i.e., snow freeboard)" The term "snow freeboard" (see also line 21 in the introduction) is misleading. Better use "total freeboard" which is ice freeboard plus snow layer thickness*

**A2:** We apologize for any confusion caused by the term "snow freeboard." In this paper, when referring to sea ice DEMs or "sea ice elevation,", we mean the total freeboard, which includes both the ice freeboard and the thickness of the snow layer. In the revised version, we have explicitly defined sea ice DEMs as total freeboard (snow+ice). Moreover, we have replaced the "sea ice elevation" with "total freeboard" throughout the manuscript, including both the text and figures, to ensure clarity and consistency.

**Q3:** *Lines 21-22: It is the mass of the ice above the water surface plus snow load (not snow freeboard) from which ice thickness can be estimated.*

**A3:** We apologize for the confusing term. In the revision Line 26-28, the sentence has been revised as: The DEM (i.e., total freeboard) can be converted to thickness with the knowledge of snow depth and the assumed values of snow, ice, and seawater densities (Kwok and Kacimi, 2018). Estimating sea ice thickness over time offers valuable insights into the overall stability of sea ice in the changing climate.

**Q4:** *Line 35: As far as I remember does the Dierking paper discuss problems and requirements for retrieving the sea ice surface topography of drifting ice but demonstrates it only for landfast ice.*

**A4:** Yes, Dierking's paper theoretically discussed the impacts of sea-ice drifting velocity on the retrieval of topographic heights and calculated the interferometric sensitivity. An example of InSAR retrieval was conducted over landfast sea ice near the coastline of Barrow. We have revised the sentences (Line 39-43) as:

Notably, the single-pass interferometric SAR (InSAR) sensor, exemplified by TanDEM-X, presents an unprecedented opportunity to generate sea ice DEMs over landfast sea ice (Dierking et al., 2017; Yitayew et al., 2018). For drifting ice, the accuracy of InSAR-derived DEMs can be affected by additional phase shifts induced by ice motion. Dierking et al. (2017) calculated and theoretically discussed the sensitivity of InSAR-derived DEMs concerning ice-drifting velocity, InSAR frequency, and baseline configuration.

**Q5:** *Line 43 and lines 54-55: "Antarctic old ice" – what precisely is "old ice"? The separation between "young ice" and "old ice" based on the criterion of penetration depth (the difference between DMS and SAR elevation) is not suitable, since salinity (as the major factor influencing the μ-wave penetration) is not only linked to ice age but also to other factors (e.g. saline snow crusts at the ice surface, effects of ice flooding). This is also visible in your data, Figs. 13-15. I propose that you instead use the categories "low-penetration condition" and "large-penetration condition".*

**A5:** The "Antarctic old ice" refers to the ice that has a penetration depth (the difference between DMS and SAR elevation) larger than $0.3\,\mathrm{m}$. We agree with the reviewer that the penetration does not simply depend on the age but on the geophysical

conditions of snow and ice. In the revision, we have modified all the "older ice (OI)" and "younger ice (YI)" into the large-penetration condition ice (LPI) and small-penetration condition ice (SPI).

**Q6:** *Lines 59-61: Sentences: "A root-mean-square error (RMSE) of 0.26m between the derived DEM and reference data signifies a precise elevation mapping for both YI and OI. Throughout the paper, "sea ice elevation" is the entire vertical height (including snow depth) above the local sea surface." Actually, 0.26 m (for averages over areas of several meters side length) can locally be a rather high (but mostly acceptable) uncertainty, considering that a large fraction of Antarctic sea ice is first-year with a thickness of around one meter (https://www.climate.gov/news-features/understanding-climate/understanding-climate-antarctic-sea-ice-extent) and correspondingly much less elevation above the water surface. "Precise" means that repeated measurements are close to one another – here the term "accurate" may be more appropriate.*

**A6:** In the Section Introduction Line 67-68, we have revised the sentence as:

A root-mean-square error (RMSE) of $0.26\,\mathrm{m}$ between the derived DEM and reference data indicates an improved accuracy in total freeboard retrieval.

In the Section Conclusion Line 464-470, we have added some texts to discuss the RMSE for both large-penetration condition ice and small-penetration condition ice a bit more:

The uncertainty level is satisfactory for LPI with RMSE of $0.26\,\mathrm{m}$. The uncertainty level is satisfactory for LPI with RMSE of 0.26 m. However, this accuracy is insufficient for thinner ice whose height above sea level is only tens of centimetres or even less. Given that a substantial portion of Antarctic sea ice consists of first-year ice with a thickness of approximately one meter (Scott, 2023), achieving accurate DEM retrieval over thinner ice remains a challenge. In the future, a potential single-pass InSAR configuration using a higher frequency, such as Ku-band, along with a longer cross-track baseline, would result in a smaller height of ambiguity (HoA) of less than $5\,\mathrm{m}$ (López-Dekker et al., 2011). This setup can enhance InSAR sensitivity and improve the accuracy of total freeboard measurements.

Scott, M.: Understanding climate: Antarctic sea ice extent., NOAA Climate Government, https://www.climate.gov/news-features/ understanding-climate/understanding-climate-antarctic-sea-ice-extent, accessed March 22, 2024., 2023.

López-Dekker, Paco, et al. "TanDEM-X first DEM acquisition: A crossing orbit experiment." IEEE Geoscience and Remote Sensing Letters 8.5 (2011): 943-947.

**Q7:** *Line 87: Here it is ground-range? Is the pixel size of 10 × 10 m used for both the classification process and for elevation retrieval? Should be stated.*

**A7:** Yes, it is ground-range. $10 \times 10\,\mathrm{m}$ is used for the following sea ice classification and DEM retrieval. In the revision Line 97-99, we have added a statement:

The multilooking processing was conducted using a $4 \times 12$ window, resulting in a $\sim 10 \times 10\,\mathrm{m}$ pixel spacing in azimuth and ground range. This resolution ($\sim 10 \times 10\,\mathrm{m}$) was subsequently utilized for the sea ice classification and DEM retrieval detailed in Section 3.

**Q8:** *Line 97: The vertical accuracy of the DMS data (line 232) should also be mentioned here. Which reference surface was used for the height values? The local water surface or a reference ellipsoid? In the User Guide by Dotson and Arvesen I found "The IceBridge DMS L3 Photogrammetric DEMs are GeoTIFF imagery, in meters and above the WGS-84 ellipsoid." (page 5). The WGS-84 ellipsoid is usually not at the same level as the local water surface.*

**A8:** We have added the vertical accuracy in Line 109-110:

...the OIB aircraft captured optical images (Dominguez, 2010, updated 2018) and generated DEM using photogrammetric techniques at a spatial resolution of approximately $40\,\mathrm{cm} \times 40\,\mathrm{cm}$ with a vertical accuracy of $0.2\,\mathrm{m}$ (Dotson and Arvesen., 2012, updated 2014).

Yes, the height values directly extracted from DMS DEM products are above the WGS-84 ellipsoid. In the postprocessing, we calibrated the values to the local sea level by selecting the water-surface reference from DMS images. For each SAR image, we labeled around ten points as water-surface references according to the DMS images. The average height of the open-water points was subtracted from the origin DMS DEMs to obtain the values relative to the average sea level.

The above processing was described in our previous work (Huang et al., 2021). In the revision, we have added texts in Section 2.3 Line 114-116 for clarification:

Note that DMS DEM gives height values relative to the WGS-84 ellipsoid. To obtain the total freeboard, we calibrated the DMS DEM to the local sea level through a manual selection of the water surface from DMS images (Huang et al., 2021). The calibrated DMS DEM, henceforth is referred to as DMS DEM for brevity.

Huang, L., Fischer, G., and Hajnsek, I.: Antarctic snow-covered sea ice topography derivation from TanDEM-X using polarimetric SAR455 interferometry, The Cryosphere, 15, 5323–5344, https://doi.org/10.5194/tc-15-5323-2021, 202

**Q9:** *Lines 115-117: I checked the types of ice charts at the US National Ice Center (https://usicecenter.gov/Products/AntarcHome). I recommend that you provide a link for your reference ("U.S. National Ice Center., 2020") and show the ice chart which you actually used. E.g., I did not find any hint that the ice charts are averages over 7 days, but are produced once a week (https://nsidc.org/sites/default/files/documents/user-guide/g10013-v001-userguide.pdf).*

**A9:** We agree with the reviewer. We have updated the statement in Line 124-125:

The U.S. National Ice Center's Antarctic sea ice charts (referred to as Ice Charts hereafter) offer weekly products detailing total sea ice concentration, partial concentration, and stage of development (U.S. National Ice Center., 2022).

We have updated the reference with a link:

U.S. National Ice Center.: U.S. National Ice Center Arctic and Antarctic Sea Ice Charts in SIGRID-3 Format, Version 1, https://doi.org/10.7265/4b7s-rn93, 2022.

**Q10:** *Section 2.4: The definition of an "average ice type" does not make sense. With regard to the notation, it is more an "ice condition index". A meaningful comparison between ice type and topography is achieved when the concentration of the respective ice type in your window is sufficiently large, e.g. > 80% or even larger. One has to consider that the topography for one ice type can be highly variable (determined by zones of deformation and their areal fractions relative to the smooth level ice areas). Hence one better concentrates on windows for which one ice type is clearly dominant. In your case that should not be a problem.*

**A10:** We have revised the manuscript according to this comment together with the latter one Q21.

We agree with the review and revised the manuscript to utilize the dominant ice type instead of the average ice type. Specifically, we have generated plots displaying the ice concentration percentages for each ice type (MYI, FYI, and TI) from the Ice Charts, making it easier to identify the dominant ice type. This allows a more comprehensive comparison with the SAR-derived results.

Note that we have checked carefully that the updated plots do not change any given interpretations or conclusions within the Section Results.

The updated plots are shown in Fig.8-10 in the revision.

**Q11:** *Line 148: what are "Pauli-1" and "Pauli-2"-polarizations? Do you refer to the Pauli-representation? Then Pauli-1 is the first and Pauli-2 the second component of the Pauli decomposition which relates surface and volume scattering?*

**A11:** The Pauli decomposition leads to the generation of the "Pauli feature vector" as:

$k = \frac{1}{\sqrt{2}}[s_{HH} + s_{VV}, s_{HH} - s_{VV}, 2s_{HV}]$

where $s_{HH}$, $s_{VV}$ and $s_{HV}$ are the complex backscattering signal in HH, VV, and HV polarizations, respectively. This representation allows separating odd, even and the 45° tilted bounce components.

Pauli-1 = $s_{HH} + s_{VV}$ represents odd bounce, usually the surface scattering. Pauli-2 = $s_{HH} - s_{VV}$ represents even bounce. In our SAR data, we do not have HV polarizations therefore only Pauli-1 and Pauli-2 were calculated and used.

In the revision, we have updated the Eq.6 as:

Similarly, we can obtain the Pauli-polarization ratio ($R_{pauli}$) by

$$R_{Pauli} = \frac{\sigma_{P1}}{\sigma_{P2}} = \frac{|s_{HH} + s_{VV}|^2}{|s_{HH} - s_{VV}|^2} \tag{1}$$

where $\sigma_{P1}$ and $\sigma_{P2}$ are denoised SAR backscattering intensity in Pauli-1 and Pauli-2 polarizations in linear scale, respectively. $s_{HH}$ and $s_{VV}$ are single-look complex images in dual-pol channels, respectively.

**Q12:** *Lines 166 – 172: see comment (4) above. There is also a misprint on line 172, it should read "hpene >=0.3 m are OI". Was there a special criterion for selecting a threshold of 0.3 m for separating YI and OI?*

**A12:** We have corrected the statement in Line 205:

$h_{\mathrm{pene}} \geq 0.3\,\mathrm{m}$ are LPI

Hallikainen and Winebrenner (1992), in Figure 3-8, suggested that the radar penetration depth of sea ice over multiyear ice ranges from 0.3 to 1 meter at X-band (9.65 GHz - 10 GHz), depending on temperature. Dierking et al. (2017) suggested that ice blocks within ridges often undergo desalination, leading to a greater effective penetration depth within the ridged ice compared to adjacent level ice. This difference in penetration depth can reduce the apparent ridge height relative to the level ice surface as observed in the interferogram. Given that the study area is covered by snow and deformed ice such as ridges, we have selected a penetration depth of 0.3 meters as a threshold for distinguishing between low-penetration and high-penetration ice conditions.

We have added the above texts in Section 3.1 Line 201-205 in the revision:

The sea ice penetration depth over multiyear ice is suggested to be $0.3 - 1\,\mathrm{m}$ at the X-band, which varies with temperature (Hallikainen and Winebrenner, 1992). Desalination within ice ridges increases the effective penetration depth compared to level ice (Dierking et al., 2017). Considering these findings and given the study area's snow cover and the presence of deformed ice such as ridges, we chose a penetration depth of $0.3\,\mathrm{m}$ as the threshold for distinguishing the two ice types. Hence, pixels with $h_{\mathrm{pene}} < 0.3\,\mathrm{m}$ are labeled as SPI, whereas those with $h_{\mathrm{pene}} \geq 0.3\,\mathrm{m}$ are LPI.

Hallikainen, M. and Winebrenner, D. P.: The physical basis for sea ice remote sensing, Microwave remote sensing of sea ice, 68, 29–46, https://doi.org/10.1029/GM068p0029, 1992

Dierking, W., Lang, O., and Busche, T.: Sea ice local surface topography from single-pass satellite InSAR measurements: a feasibility study, The Cryosphere, 11, 1967–1985, https://doi.org/10.5194/tc-11-1967-2017, 2017.

**Q13:** *Lines 185 − 186: The question is how your "YI" class is related to the ice types listed in Table 2. The WMO-category defines young ice as ice between 10 cm and 30 cm in thickness, as correctly listed in your table. It can be assumed that a penetration depth up to 0.3 m (your "YI") covers the categories "Thin Ice" and "First-Year Ice". The "old ice" with a penetration depth >= 0.3 m cover the thicker FY ice and MY ice. Conclusion is that you should not use the notation YI for penetration depths < 0.3m, see comment 4 above. Check also notations used in section "Conclusions".*

**A13:** We agree with the reviewer. As suggested in A5, we have replaced the "YI" and "OI" terms with the small-penetration condition ice (SPI) and large-penetration condition ice (LPI) throughout the manuscript.

**Q14:** *Figure 6 and Section 2.5: How do you relate hInSAR to the local water surface? Or in other words: which reference surface do you actually use when calculating hInSAR? I suppose that in the initial InSAR processing it is not the local water surface but also the WGS84-elliposid?*

**A14:** The initial height $h_{\mathrm{InSAR\_ini}}$ generated from InSAR processing was referenced to the WGS84 ellipsoid. In theory, we could identify the pixels of the water area and calibrate the InSAR height relative to the water surface by $h_{\mathrm{InSAR}} = h_{\mathrm{InSAR\_ini}} - h_{\mathrm{InSAR\_water}}$. However, due to the low InSAR coherence (less than 0.3) in water areas, these water pixels were masked out as the $h_{\mathrm{InSAR\_water}}$ are inaccurate and can not be used for analyses. Hence, instead of identifying water pixels, we selected smooth and new ice regions, assuming they are thin enough and their elevation (i.e., radar freeboard) is negligible and approximately equal to the water surface. These regions typically exhibit very low SAR backscattering intensities but good InSAR coherence (larger than 0.3). Therefore, we selected pixels with backscattering intensities within the range of $-19\,\mathrm{dB}$ to $-18\,\mathrm{dB}$, slightly above TanDEM-X's noise level ($-19\,\mathrm{dB}$), and generated a histogram of $h_{\mathrm{InSAR\_ini}}$ values for these pixels. We determined the 3rd percentile of the height of these thin-ice pixels to be the water surface elevation ($\approx h_{\mathrm{InSAR\_water}}$). The threshold value, i.e., the 3rd percentile, is chosen by conducting the aforementioned method over the four SAR scenarios overlaid with DMS DEM. By choosing the 3rd percentile as the water surface level, we ensured alignment between the water surface levels derived from InSAR and those from the DMS data, thus validating the threshold value.

Note that we estimated a single value representing the water surface for each SAR scene, which covers an area of $50 \times 19\,\mathrm{km}$. This method may introduce inaccuracies due to the centimeters level of the radar freeboard of the selected thin and new ice, as well as the fluctuating water surface within each SAR scenario.

In the revision, we have included the above text in Section 2.5 Line 149-160.

**Q15:** *Section 3.2. Here, many things are unclear to me. In summary, I recommend to rewrite this section for the sake of clarity. Single issues: (a) what is the exact definition of m, does it refer to thickness of layer 1 and layer 2? (b) Is hmod the*

*surface elevation above the water surface (or reference ellipsoid)? (c) Which AMSR Level 3 data did you use for retrieving the snow depth over your test sites? How large is the uncertainty of those snow depth values? (d) line 202: "$\phi_{DMS}$ can be transformed into $\phi_{DMS}$ by Eq. (3)" : equation 3 describes the relation between hInSAR and $\phi_{\gamma}$. I wonder whether this equation can be simply applied using hDMS to derive $\phi_{DMS}$ because for the DMS data there is no height of ambiguity. Is $\phi_{DMS}$ assumed to be the phase at the snow-air interface, hence $\phi_{DMS} = \phi_0$? (e) From equation 8, you obtain m and hv, according to the given definition and Fig. 6 hv is ice thickness which could be mentioned. (f) Why is the second step needed, namely to combine the SAR features with m and hv to obtain modified values m' and hv' ? With the classified images (your maps of ice types), one can directly link the m and hv values from the first step in Fig. 7 with the corresponding ice classification map. The second step is not needed.*

**A15:** We have extensively revised Section 3.2 to enhance clarity, addressing points (a) to (f).

(a) In a previous study, we proposed a layer-to-layer scattering ratio ($m$) (Huang et al. 2021), inspired by the concept of the layer-to-volume scattering ratio utilized in ice sheeting (Fischer et al., 2018).

The layer-to-layer scattering ratio ($m$) refers to the backscattering power ratio between the top and bottom layers:

$$m = \frac{\sigma_{bottom}(\vec{\omega})}{\sigma_{top}(\vec{\omega})}$$

where $\sigma_{top}\vec{\omega})$ and $\sigma_{bottom}(\vec{\omega})$ denotes the backscattering power from the top and bottom interface, respectively, for a given polarization channel $\vec{\omega}$.

$m$ potentially reveals the relative importance of scattering from these interfaces, depending on factors like interface roughness, dielectric constant, and radar polarization. A larger value of $m$ signifies that surface scattering from the bottom layer predominates, while a smaller $m$ indicates that surface scattering from the top layer is more significant.

In the revision, we have included the above texts in Line 236-242.

(b) $h_{\mathrm{mod}}$ refers to the elevation of the snow-air interface above the water surface. In the revision Line 259, we have added that $h_{\mathrm{mod}}$ is the total freeboard: ...transformed into total freeboard $h_{\mathrm{mod}}$ using Eq. (3).

(c) We used the AMSR-E/AMSR2 Unified L3 Daily 12.5 km Brightness Temperatures, Sea Ice Concentration, Motion & Snow Depth Polar Grids, Version 1 dataset (https://doi.org/10.5067/RA1MIJOYPK3P) for our study. The provided snow depth over sea ice represents a five-day running average. Due to the limited spatial and temporal resolution in the snow depth data, we assume a constant value of snow depth across one SAR image. For each SAR acquisition, covering a spatial extent of $50 \times 19\,\mathrm{km}$, we compute the mean snow depth and utilize it as input parameter $z_1$ in the TLPV model. In our previous study (Huang et al., 2021), we evaluated the TLPV model's performance concerning variations in snow depth. Our findings indicated a mean discrepancy of $0.31\,\mathrm{m}$ in the derived total freeboard due to snow depth fluctuations ranging from 0.05 to $0.75\,\mathrm{m}$, which covers the major ($> 95\%$) snow depth range in the Weddell Sea (Webster et al. 2018). In future investigations, it would be promising to adapt our proposed total freeboard retrieval method to test sites that coincide with available high-resolution snow depth measurements.

In the revision, we have included the above texts in Line 415-426.

(d) For InSAR, there is a general equation accounting for the relation between the interferometric phase ($\phi$) and height ($h$)

$$h = h_a \frac{\phi}{2\pi}$$

where $h_a$ is the height of ambiguity determined by the specific InSAR configuration such as the radar wavelength, orbit height, incidence angle, and baseline. In instances where we have an external DEM, such as from photogrammetry or lidar, we can also simulate the interferometric phase ($\phi_{\mathrm{DMS}}$) from the height ($h_{\mathrm{DMS}}$) using the same equation tailored to a specific InSAR configuration giving specific $h_a$.

Yes, $\phi_{\mathrm{DMS}} = \phi_0$.

In the revision, we have updated texts in Line 248-249.

(e) $hv = z_1 - z_2$ represents the depth between the top ($z_1$) and bottom ($z_2$) interfaces, which contribute to surface scattering effects. However, it's important to note that the bottom layer is not always at the ice-water interface. In some cases, a lower basal sea ice layer can exist, as discussed in Nghiem et al. (2022), which induces strong surface scattering from the ice-basal layer interface. This basal saline layer contains brine inclusions with higher salinity, transitioning toward the ice-seawater interface, as observed in the Western Weddell Sea (Tison et al., 2008). The salinity in this basal layer can be attributed to past

flooding events on younger, thinner ice that eventually becomes older and thicker. Furthermore, brines in the upper ice volume above the sea level drain down, further salinating the lower basal layer (Nghiem et al., 2022). Therefore, in scenarios where a high-salinity basal layer is present, $hv$ may not be equivalent to the ice thickness.

Nghiem, S. V., Huang, L., and Hajnsek, I.: Theory of radar polarimetric interferometry and its application to the retrieval of sea ice elevation in the Western Weddell Sea, Antarctic, Earth Space Sci., 9, e2021EA002 191, https://doi.org/10.1029/2021EA002191, 2022.

Tison, J.-L., Worby, A., Delille, B., Brabant, F., Papadimitriou, S., Thomas, D., & Haas, C. Temporal evolution of decaying summer first-year sea ice in the Western Weddell Sea, Antarctica. Deep-Sea Research Part II: Topical Studies in Oceanography, 55(8–9), 975–987. https://doi.org/10.1016/j.dsr2.2007.12.021, 2008

In the revision, we have included the above texts in Line 231-234.

(f) First, for the SAR scene overlaid by the DMS measurements, $m$ and $h_v$ were calculated by inverting Eq.(8) using $\phi_{\mathrm{DMS}}$ ($= \phi_0$) as input. In the subsequent step, assuming $m$ and $h_v$ from the first step as true values, we established an empirical relation (RF regressor) between SAR features and $m$ and $h_v$. For the SAR scene without overlapped DMS measurements, the estimated $\hat{m}$ and $\hat{h}_v$ are calculated from the well-trained RF regressor and then input into Eq.(8) to generate $h_{\mathrm{mod}}$, which is the total freeboard for the large-penetration condition ice.

In the revision, we have included the above texts in Line 245-260.

**Q16:** *Fig 9: the right-most color bar is "E", is this the DEM given in meters?*

**A16:** The DEM is given in meters. We apologize for the confusion. It is "m" (stands for meters) but printed vertically. In the revision Fig.5, we have updated the colorbar's label to "total freeboard (m)".

**Q17:** *Figure 11: What is the explanation for the data gaps in the second profile from the top?*

**A17:** The data gaps are the water areas and segments that were excluded due to inaccurate data co-registration. During InSAR processing, pixels with InSAR coherence below $0.3$ were masked out. Additionally, during data co-registration between DMS and SAR, segments lacking distinctive sea ice features in optical and SAR images, and hence, unable to ensure co-registration quality, were eliminated.

We have added an explanation in the caption of the figure:

Figure 7. Comparison between the elevation profiles ($h_{\mathrm{mod\_SAR}}$) from the proposed method and the DMS DEM ($h_{\mathrm{DMS}}$) along the dotted line (from A to B) over Scene No.1-4 in Fig.5(c). The data gaps are the water areas and segments that were excluded due to inaccurate data co-registration.

**Q18:** *Results shown in Figs. 9, 10, 11: hmodSAR is elevation = total freeboard relative to the water surface (or reference ellipsoid), retrieved from pixels of 10 × 10 m in size?*

**A18:** Yes, $h_{\mathrm{mod\_SAR}}$ are total freeboard relative to the water surface retrieved from $10 \times 10\,\mathrm{m}$ pixel size. In the revision Line 273, we have added the texts:

Note that $h_{\mathrm{mod\_SAR}}$ represents total freeboard relative to the water surface retrieved from the pixel at $10 \times 10\,\mathrm{m}$ spacing size.

**Q19:** *Again Figs. 9, 10, 11: For "old ice", height values = total freeboard of even larger than 3 m are measured. Intuitively, this seems to be not very realistic, although DMS and SAR values match. Is there any information available about the area regarding ice and snow conditions which seems to be special when comparing to the results of the other profiles shown in Figs. 13-15 which reveal smaller heights? If you used the WGS84 ellipsoid as reference: Is the difference between reference ellipsoid and mean sea level larger close to the Antarctic Peninsula? Perhaps also icebergs biased the measurements?*

**A19:** The four scenarios depicted in these figures in the manuscript are located near the Eastern Antarctic Peninsula. Notably, we observed considerable deformation and prominent ridges in the sea ice from the DMS photo. While, upon thorough examination of DMS photos across the four scenes, no icebergs were identified. Therefore, we think it is the deformed ice and ridges contribute to the high total freeboard values.

Here, we show a DMS photograph captured a sub-region within Scene No.1, see Fig.1(a) below. In Fig.1(b), we used the DMS DEM derived from this photograph, annotated specific sea ice features, and extracted corresponding elevation values.

[Figure]

(a)          (b)

Fig. 1: The red rectangle indicates the selected sub-region overlapped by DMS measurements. (b) The optical photo alongside elevations derived from the DMS DEM within the selected sub-region. The elevation values are referenced to the WGS84 Ellipsoid.

The elevation values in Fig. 1(b) are referenced to the WGS84 Ellipsoid, representing the original values obtained from the DMS DEM products. The water surface ranges between 12.4-12.5m, while the thin ice adjacent to the water ranges from 12.5-13.0m, indicating a relative elevation of 10-50 centimeters compared to the water surface. On the ice floes, the smooth surfaces measure around 13.0-14.0m, representing an elevation of 0.5-1.5m above the water surface. The rough surfaces, including deformed ice and ridges, exhibit elevations ranging from 14.5-17m, with corresponding relative elevations to the water surface exceeding 3m.

During personal correspondence with Dr. Son Nghiem from NASA JPL, he provided us with a photo captured during the campaign in the study area, shown in Fig. 2. We observed prevalent deformed features that surpass the height of a person.

[Figure]

Fig. 2: A photo taken during the OTASC campaign. Courtesy of Dr. Son Nghiem.

**Q20:** *Figure 12: The color bars show values of hmodSAR down-sampled to a resolution of 500 m?*

**A20:** Yes, the results in Sections 4.2, 4.3, and 4.4 (including Fig.12-16) are all based on the $h_{\mathrm{mod\_SAR}}$ down-sampled to $500\,\mathrm{m}$. We stated this at the beginning of Section 4.2. In the revision, the plot has been moved to the appendix and we have added the statement in the caption of Fig.A5 and A6:

$h_{\mathrm{mod\_SAR}}$ was downsampled to $500\,\mathrm{m}$ pixel size.

**Q21:** *Figures 13,14,15, left column: The percentage of "OI" refers to the large-penetration-depth category. It would be much easier to discuss the relationship between the OI and real ice types when concentrations of MY, FY, and TI are shown instead of an index related to the "average ice type". Lines 241-243: In Fig. 13, the percentage of OI is only 58% for distances between 0 and 160 m, but the ice type in the ice chart seems to be almost 100% MYI ice since the "average ice type" value is close to 3. In the range between approximately 300 and 600 km, MYI is also close to 100% concentration, but the percentage of OI is down to 20-25%. (Note that according to section 2.4, the indices for TI, FYI, and MYI should be 0,1,2, respectively, and not 1,2,3) This demonstrates that the penetration depth is not directly linked to the ice types shown in the ice chart.*

**A21:** We have renamed the "OI" as large-penetration condition ice (LPI) in the revision. Instead of using average ice type, we have updated the results by showing the three ice types (MYI, FYI, and TI) with concentrations, see **A10**.

The ice concentration values for MYI from Ice Charts and the LPI percentage from SAR are not necessarily identical, as they are classified by different criteria. Nevertheless, our study observed and discussed a similar trend between the dominant ice types and ice concentrations from the Ice Charts and the LPI percentage from SAR. In the revision, we have updated Line 291-305 for clarification:

The overall trend of estimated LPI percentages correlates well with dominant ice types and ice concentrations from the Ice Charts across most segments (W1-U, W1-L, W2-U, W4, W5-U, W5-L, and R5). Specifically, W1-U and W1-L are explained as two examples. W1-U from 0-120 km is covered by 100% MYI, where the LPI percentage reaches its highest value (58%). As the dominant ice transitions from MYI to FYI from 120-400 km, the LPI percentage decreases accordingly. The dominance of MYI ice from 400-600 km also corresponds to the increasing LPI percentage. For W1-L, where the distance between 500-600 km is covered by 100% MYI, the LPI percentage peaks before decreasing after 650 km distance as FYI becomes dominant. The lowest LPI percentage is found around 1200 km, consistent with the occurrence of TI at this distance.

The observed similar trend can be explained by the general assumption that MYI is thicker and less saline, allowing for deeper radar penetration compared to FYI and TI. However, penetration depth is influenced by various factors not only ice age, but also ice salinity, snow condition, flooding effects, and temperature. This explains the discrepancies for other segments (W2-L, W3-U, W3-L, R1-U, R1-L). Furthermore, discrepancies can also be attributed to differences in spatial resolution and temporal gaps between Ice Charts and SAR imagery, considering the dynamic nature of sea ice.

It is essential to clarify that we utilized Ice Charts data as external information to interpret the classification results and spatial variation of topography. However, we did not use Ice Charts to quantitatively validate the proposed method. For validation purposes, we conducted pixel-by-pixel comparisons using co-registered DMS data.

**Q22:** *Line 253-261: Referring to the statement: "In general, the region with thicker ice (e.g., MYI) is anticipated to display higher elevation or larger roughness compared to the area with thinner ice, such as FYI and TI": Locally, rough FYI may reveal a larger roughness than smooth level MYI, and it may reveal a higher elevation when covered by a very thick snow cover. This may also explain discrepancies.*

**A22:** Agree. In the revision Line 316-318, we have added the explanation for the discrepancies:

These discrepancies may arise due to the local cases where rough FYI exhibits greater roughness than smooth level MYI. FYI may also show higher elevations when covered by very thick snow.

**Q23:** *Line 262: "sea ice . . . exhibits. . . highest elevation". You should again clearly state that the values of elevation are total freeboard, i.e. also include the snow layer.*

**A23:** In the revision, we have replaced "elevation" with "total freeboard" throughout the manuscript.

**Q24:** *Line 265: Figs. 6g-l in the paper by Wang et al (2020) are indeed well suited for comparing with your results. The values they show (Fig. 6l for 2017), however, have only a narrow peak at 1.5 to 2.5 m, otherwise values are lower. Figs 6g-l should also be mentioned with regard to your Figs 13-15, considering window sizes for averaging the elevation.*

**A24:** Upon thorough examination of Wang et al.'s work (2020), we were unable to find the resolution (window size) information they used for plotting Fig. 6l. However, based on their mentioning of a $20\,\mathrm{km}$ width track, we estimate that each dot corresponds to a window size larger than $20\,\mathrm{km}$, significantly exceeding the $500 \times 500\,\mathrm{m}$ window size we utilized. Unfortunately, Wang et al. did not provide the processed data used in Fig. 6l in the supplements, preventing a quantitative comparison with our results. Therefore, we conducted a visual comparison between Fig. 6l in Wang et al. (2020) with the four segments (W2-U, W2-L, W3-U, and W3-L) in our study.

We have included Fig. 6l (as shown in Fig. 3 below), and we define the two tracks in Wang's result as Track-W and Track-E. The region with latitude $< 70°\mathrm{S}$ $(> 70°\mathrm{S})$ is referred to as the northern (southern) track.

[Figure]

Fig. 3: The total freeboard from the paper by Wang et al., 2020 (Fig.6l). We labeled the two tracks Track-W and Track-E. The region with latitude $< 70°\mathrm{S}$ $(> 70°\mathrm{S})$ is referred to as the northern (southern) track.

The Northern and Southern Track-W segments are partially overlaid with W2-U and W2-L, respectively. In our study, the total freeboard of W2-U and W2-L reaches a mean value of $\sim 1\,\mathrm{m}$ and 75% percentile value of $\sim 1.5\,\mathrm{m}$ within the first $100\,\mathrm{km}$, which agrees with the red dot in Fig.6l Track-W. Then, the total freeboard goes down to a mean value as $\sim 0.7\,\mathrm{m}$ and 75% percentile value of $\sim 0.75\,\mathrm{m}$ from $100 - 200\,\mathrm{m}$. Although there is a data gap in the Fig.6l (Wang's paper), we can see that color of the dots changes from red to yellow, which is consistent with the decreasing trend of the total freeboard within $300\,\mathrm{km}$ from our results. For W2-L, the total freeboard from results are around at the mean values of $0.5\,\mathrm{m}$, agree with a mix of green and yellow dots $(0.4 - 0.9\,\mathrm{m})$ in the Southern Track-W in Fig.6l (Wang's paper).

Note that the OIB ATM data used for Wang's study was acquired on 14 and 22 November 2017, while the SAR images in our study were acquired on 30 and 25 October 2017 for W2-U and W2-L, respectively. The sea ice drifts and potential melting could induce slight differences between our results and Wang's results.

The W3-U and W3-L can be compared with Northern and Southern Track-E, respectively. From Fig. 6l, a mix of green and yellow dots in the northern Track-E represent the total freeboard $0.5 - 1.2\,\mathrm{m}$, which agrees well with our result for W3-U, see the first row in Fig.9 (in the manuscript). At around $70°\mathrm{S}$, the dots transit to a mix of cyan and blue colors, representing the total freeboard of $0.2 - 0.7\,\mathrm{m}$, which is consistent with the W3-L in Fig.14 (in the manuscript). The slight difference can be attributed to the temporal difference of SAR images used in our study. Specifically, the image for W3-L was acquired on October 26, 2017, while the Track-E image was acquired on November 22, 2017.

In the revision, We have added the above analyses in a new Section 5.2.

**Q25:** *Line 283: ice type "MTI"? I think it should be MYI.*

**A25:** Corrected.

**Q26:** *Line 294-295: Sentence: "The variation of the roughness along the R1 segment also highlights the importance of combining topographic mapping with ice category mapping to comprehensively characterize sea ice features." Since there is no direct relationship between ice type and topography data, it is not per se "important" to combine both, but can be useful in certain cases, e.g. for operational ice charting. Since the edges of ice floes with open water between the floes contribute to the ice roughness, ice concentration may also be a useful parameter to be combined with ice topography in some cases.*

**A26:** We agree with the reviewer. We have modified the statement in the revision Line 353-356:

The variation of the roughness along the R1 segment suggests that ice topography provides add-on information that can be useful to be integrated into the operational ice charting. Furthermore, Since the edges of ice floes with open water between the floes can also contribute to the ice roughness, combining ice topography with ice concentration can help characterize the sea ice more comprehensively.

**Q27:** *A point of my interest: For radar applications, it would also be useful to show how large deviations between DMS and SAR values (= your penetration depth) can be, and where this occurs (spatial distribution along your tracks).*

**A27:** Thanks for this super interesting point. Considering the studied area is mainly covered by snow, we believe the penetration largely depends on the local snow conditions and snow spatial distributions. Unfortunately, our current dataset does not have in-situ measurements of snow properties, so we cannot investigate the deviation with the snow parameters. In the future, it would be interesting to propose/conduct fieldwork that includes TanDEM-X acquisitions, lidar measurement, and in-situ snow measurements and analyze the relation between the radar freeboard and total freeboard at varying snow depths.

We have added texts in the Section Conclusion Line 485-488:

The spatial distribution of penetration depth (total freeboard minus radar freeboard) can be an interesting topic for future research. In snow-covered sea ice, penetration is significantly influenced by local snow conditions. Hence, conducting a coordinated campaign encompassing TanDEM-X acquisitions, lidar measurements, and in-situ snow assessments holds great promise for analyzing the relation between radar freeboard and total freeboard across different snow conditions.

Again, we sincerely thank the editor and reviewers for helping us improve the manuscript.

---

## Author Response (AR2)

**Response to the comments of Reviewer**

First, we express our gratitude to the anonymous reviewers for their valuable suggestions. Herein, we provide a point-by-point response to address the additional comments provided by the second reviewer.

**Q1:** *lines 2-3: I don't think that the replacement of the first sentence is an improvement. The freeboard as such is one parameter from which other parameters (thickness, surface roughness) can be deduced, but it does not reflect sea ice dynamics directly. I suggest: "The total freeboard, which is the ice layer above water level and includes snow thickness, is needed to retrieve ice thickness and ice surface topography.*

**A1:** We have replaced the sentence with the suggested one.

**Q2:** *line 21: "The sea ice surface topography plays a crucial role for understanding sea ice dynamics and interactions within the air-ocean-ice system. It determines the spatial...*

**A2:** Done.

**Q3:** *line 93: ...with only seconds varying between them...? I think you mean: ...image pairs from the same orbit that are acquired with only a few seconds difference.*

**A3:** Yes, we have modified the sentence as suggested.

**Q4:** *section 2.4: I would prefer if one ice chart is shown as an example.*

**A4:** In the revision, a new figure (Fig.3) has been added to visualize the Ice Chart as an example.

**Q5:** *line 169: not "it's", better "it is"*

**A5:** Done.

**Q6:** *lines 215-216: Suggestion: According to Hallikainen and Winebrenner (1992) the penetration depth into multi-year ice varies between 0.3 and 1 m at X-band, depending on salinity and temperature.*

**A6:** Done.

**Q7:** *line 254-255: The parameter m refers to... Question: Is it a really a layer-to-layer ratio? This name is misleading in view of equation 9 which tells us that it is the ratio between the backscattered intensities of the upper and lower interface of one layer which you explain following equation 9.*

**A7:** In the revision, we renamed the parameter m to "layer-to-layer scattering ratio" to avoid confusion.

**Q8:** *Coming back to my concern regarding freeboard values $> 3m$: It would be helpful to provide the information about spatial resolution. in Fig. 7 and Fig.12. If it is 10 m then a local freeboard of 3 m is realistic in areas of heavy deformation and snow load. If the resolution is $> 100$ m such a value is not realistic.*

**A8:** We agree. In the revision, we have added the spatial resolution to the captions of both Fig. 6 and Fig. 8.

Yes, the spatial resolution in Fig. 7 (now Fig. 8 in the revision) is $10 \times 10$ m, therefore it is realistic that the values can be above 3m due to the heavy deformation. For Fig. 12 (now Fig. 13 in the revision) from Wang's results, the Icebridge ATM L1B elevation data with a footprint of 1m were used. However, Wang did not mention the spatial resolution for plotting the red dot representing 1.5-2.5m in the figure.

**Q9:** *line 395: "Furthermore, since... "*

**A9:** Done.

Again, we sincerely thank the editor Vishnu and the reviewers for helping us improve the manuscript.